# Myosin IIA and formin dependent mechanosensitivity of filopodia adhesion

N.O. Alieva[1,8], A.K. Efremov[1,2,8], S. Hu[1], D. Oh[1], Z. Chen[1,3], M. Natarajan[1], H.T. Ong[1], A. Jégou [4], G. Romet-Lemonne[4], J.T. Groves[1,3], M.P. Sheetz[1,5], J. Yan[1,2,6] & A.D. Bershadsky [1,7]

Filopodia, dynamic membrane protrusions driven by polymerization of an actin filament core, can adhere to the extracellular matrix and experience both external and cell-generated pulling forces. The role of such forces in filopodia adhesion is however insufficiently understood. Here, we study filopodia induced by overexpression of myosin X, typical for cancer cells. The lifetime of such filopodia positively correlates with the presence of myosin IIA filaments at the filopodia bases. Application of pulling forces to the filopodia tips through attached fibronectin-coated laser-trapped beads results in sustained growth of the filopodia. Pharmacological inhibition or knockdown of myosin IIA abolishes the filopodia adhesion to the beads. Formin inhibitor SMIFH2, which causes detachment of actin filaments from formin molecules, produces similar effect. Thus, centripetal force generated by myosin IIA filaments at the base of filopodium and transmitted to the tip through actin core in a formin-dependent fashion is required for filopodia adhesion.

[1] Mechanobiology Institute, National University of Singapore, T-lab, 5A Engineering Drive 1, Singapore 117411, Singapore. [2] Center for BioImaging Sciences, National University of Singapore, 14 Science Drive 4, Singapore 117557, Singapore. [3] Department of Chemistry, University of California, Berkeley, CA 94720, USA. [4] Institut Jacques Monod, CNRS, Université de Paris, 15 rue Helene Brion, F-75013 Paris, France. [5] Department of Biological Sciences, Columbia University, New York, NY 10027, USA. [6] Department of Physics, National University of Singapore, Singapore 117542, Singapore. [7] Weizmann Institute of Science, Herzl St 234, Rehovot 7610001, Israel. [8] These authors contributed equally: N.O. Alieva, A.K. Efremov. Correspondence and requests for materials should be addressed to A.D.B. (email: alexander.bershadsky@weizmann.ac.il)

Filopodia are ubiquitous cell extensions involved in cell motility, exploration of the microenvironment, and adhesion[1,2]. These finger-like membrane protrusions help cells to determine the direction of movement[3], establish contacts with other cells[4], and capture inert particles or living objects (bacteria), which cells subsequently engulf[5–7]. Filopodia are involved in numerous processes of embryonic development, as well as in cell migration in adult organisms. Moreover, augmented filopodia activity is a hallmark of tumor cells, which use them in the processes of invasion and metastasis[1].

The main element of filopodia is the actin core, which consists of parallel actin filaments with barbed ends oriented towards the filopodium tip, and pointed ends toward the cell body[1,2]. Actin filaments are connected to each other by several types of cross-linking proteins[8,9]. The filopodia grow via actin polymerization at the tip, in a process driven by formin family proteins such as mDia2[10,11], FMNL2 and 3[12,13], as well as by actin elongation protein Ena/VASP[14,15]. In addition to proteins that cross-link and polymerize actin, filopodia also contain actin-based molecular motors. In particular, myosin X, localized to the tips of the filopodia is known to be required for filopodia growth, and its overexpression promotes filopodia formation[16–18].

Adhesion of the filopodia to the extracellular matrix (ECM) is mediated by the integrin family of receptors (e.g., $\alpha_v\beta_3$)[19], which are localized to the tip area. In addition to integrins, filopodia tips have been shown to contain other proteins involved in integrin-mediated adhesion, such as talin and RIAM[19]. The hallmark of integrin-mediated adhesions of focal adhesion type is their mechanosensitivity[20,21]. They grow in response to pulling forces applied to them, either by the actomyosin cytoskeleton, or exogenously by micromanipulations, and may play a role in matrix rigidity sensing[22]. Filopodia also may participate in matrix rigidity sensing. For example, it was demonstrated that cell durotaxis, a preferential cell movement along a gradient of substrate rigidity, is mediated by filopodia[23]. However, force dependence of filopodia adhesion has not yet been explored.

In the present study, we find that lifetime of filopodia induced by myosin X depends on forces generated by myosin IIA localized to filopodia bases. To further investigate the force dependence of filopodia growth and adhesion, we monitored filopodia attached to beads coated with matrix protein fibronectin under conditions of pulling with a constant rate. We demonstrate that adhesion of filopodia to fibronectin strongly depends on myosin IIA activity. Moreover, formin family protein function at the filopodia tips is also requires for filopodia adhesions, most probably through a role in the transmission of force through the actin core, from the filopodium base to the filopodium tip. Thus, filopodia are elementary units demonstrating adhesion-dependent mechanosensitivity.

## Results

**Myosin X-induced filopodia are associated with myosin IIA.** Transfection of HeLa-JW cells with either GFP-myosin X or mApple-myosin X promoted filopodia formation in agreement with previous studies[16]. Myosin X was concentrated at the filopodia tips, forming characteristic patches also called puncta or comet tails (Fig. 1). Here, we focused on filopodia originating from cell edges and extending along the fibronectin-coated substrate. These filopodia demonstrated periods of persistent growth, with an average velocity of $67 \pm 6$ nm/s (mean ± s.e.m., $n = 89$) interrupted by pauses and periods of shrinking with an average velocity of $28 \pm 3$ nm/s (mean ± s.e.m., $n = 100$), consistently with previously published results[24]. In addition to myosin X, the expression of fluorescent fusion protein constructs of other characteristic proteins such as mDia2, VASP, and talin revealed the preferential localization of these proteins to the filopodia tips

(Fig. 1a, b) in agreement with localization of endogenous proteins[25–27].

Expression of GFP-labeled myosin light chain in HeLa-JW cells showed that myosin II does not localize to the filopodia tips or shafts, but is often seen at the proximal ends of the filopodia (Fig. 1c, d, Supplementary Movie 1). Structured illumination microscopy (SIM) visualizes individual bipolar myosin II filaments as pairs of fluorescent dots (corresponding to myosin II head regions of the filaments); the filament length was estimated to be 300 nm[28]. One or few myosin II filaments were usually located at the filopodium base (Fig. 1d).

Cos-7 cells provide a convenient model for the studies of myosin IIA localization and function, since these cells express myosin IIB and IIC, but not myosin IIA[29,30]. Upon transfection with myosin X, the wild-type Cos-7 cells form filopodia with a short life-time that apparently do not adhere to the fibronectin-coated substrate (Fig. 1f, Supplementary Movie 2). Co-transfection of these cells with myosin IIA-GFP heavy chain (GFP was localized at the N-terminus adjacent to myosin IIA head) resulted in formation of numerous bipolar myosin IIA filaments visualized as doublets of fluorescent dots. These doublets were often localized to the bases of filopodia (Fig. 1e, g and Fig. 2a, b, Supplementary Movie 3). The life imaging of 102 filopodia from 3 cells (including the cell shown in Supplementary Movie 6) revealed (Fig. 2d) that 54 of them had one or more myosin IIA doublets at their bases during the whole duration of the movie (240 s) as in Fig. 2b. Fourty-five filopodia had no myosin IIA as in Fig. 2c, and in the remaining three filopodia the myosin IIA filaments appeared during part of the observation period.

Myosin IIA mutant N93K, which has reduced myosin ATPase and motor activity, but preserves the ability to bind to actin filaments[31,32], was still localized to the bases of myosin X-induced filopodia, similar to wild-type myosin IIA (Fig. 1h, Supplementary Movie 4). Out of 56, 52 filopodia contained N93K-positive filaments at their bases during observation period (240 s) as shown in Supplementary Movie 4. Unlike myosin IIA, Emerald-myosin IIB was weakly localized to the filopodia bases in these cells (Fig. 1i, Supplementary Movie 5). Only 19 of 90 assessed filopodia contained myosin IIB at their bases during period of observation (240 s).

**Myosin IIA augments filopodia lifetime in Cos-7 cells.** Comparison between dynamics of filopodia associated and non-associated with myosin IIA filaments showed that the presence of myosin IIA at the filopodia bases positively correlates with the duration of lifetime of filopodia (Fig. 2d). Among filopodia having myosin IIA at their bases, the majority (45 of 54) existed for more than 240 s, while among filopodia nonassociated with myosin IIA only 4 survived for this period. Lifetime of other filopodia nonassociated with myosin IIA was only $102 \pm 2$ s (mean ± s.e.m., $n = 45$). In the Fig. 2d, the bars representing the lifetimes of filopodia associated with myosin IIA filaments during entire period of observations, part of this period, and nonassociated, were marked by magenta, yellow, and cyan colors, respectively. The difference between lifetimes of myosin IIA-associated and nonassociated filopodia for three assessed cells (pooled data, Fig. 2d) were significant according to nonparametric Wilcoxon rank-sum test ($p < 0.001$). Thus, these data show that the presence of one or few myosin filaments at the filopodium base locally augments the lifetime of this filopodium.

We then compared the lifetimes of filopodia in Cos-7 cells transfected with different constructs of myosin X and myosin II (Fig. 2e, f). In these experiments we did not distinguish between filopodia associated and nonassociated with myosin II, but

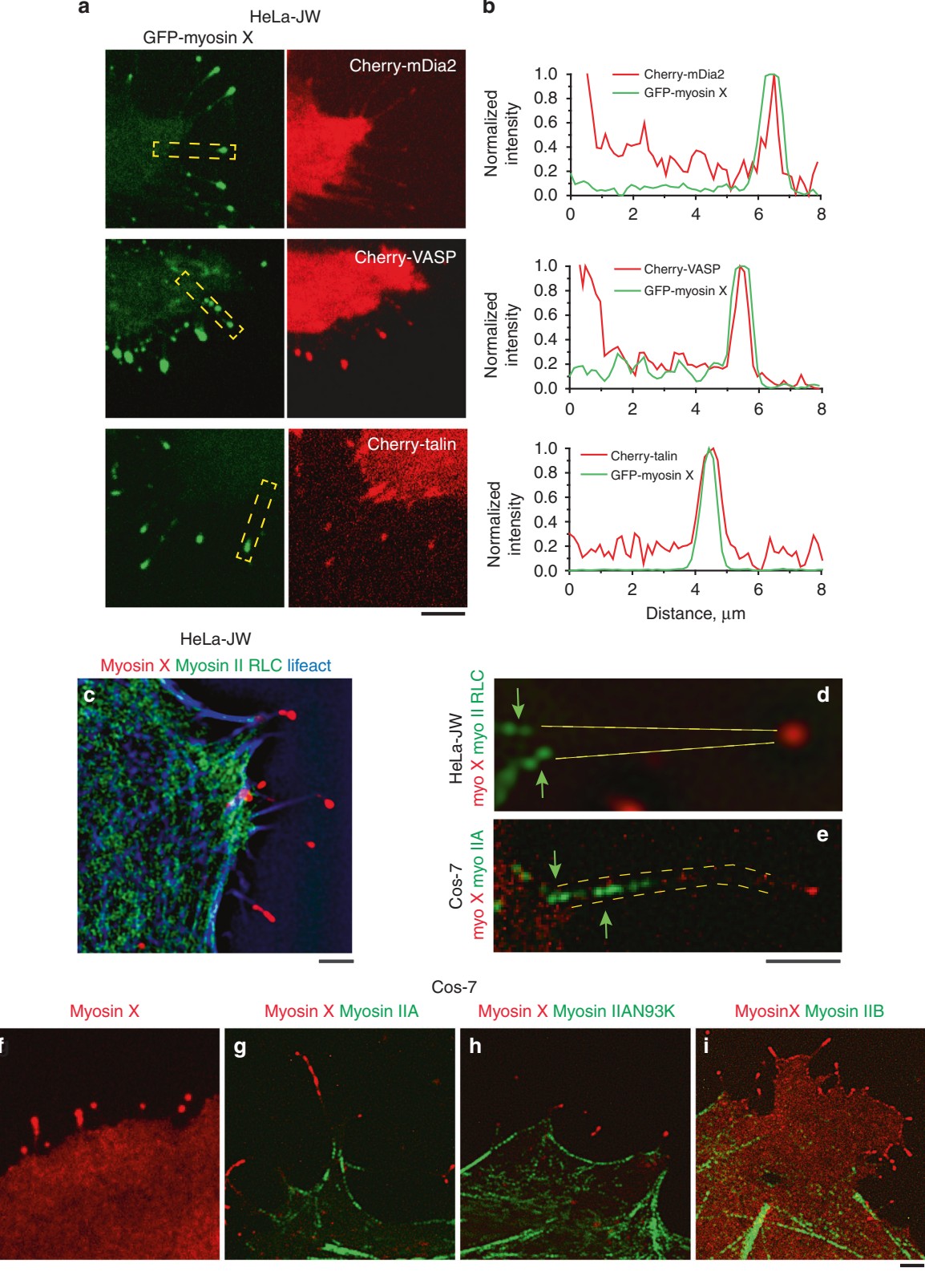

compared the survival curves in the cohorts of filopodia from cells treated as indicated (Fig. 2e) and calculated the corresponding lifetimes (Fig. 2f). This analysis revealed that average lifetime of filopodia in cells expressing myosin IIA and myosin X was significantly higher than in cells expressing myosin X only (Fig. 2f). Unlike the wild type, mutant myosin IIA N93K only slightly enhanced the lifetime of filopodia (Fig. 2e, f). Thus, motor

activity of myosin IIA is critically important for its effect on filopodia lifetime.

In agreement with previous studies[33], formation of filopodia in Cos-7 cells can be also induced by expression of truncated myosin X lacking FERM domain (Supplementary Movie 7). Co-expression of myosin IIA significantly enhanced the lifetime of the filopodia induced by this truncated myosin X (Supplementary

**Fig. 1** Molecular composition of filopodia induced by myosin X expression. **a** Images of filopodia in HeLa-JW cells co-expressing GFP-myosin X with mCherry fusion constructs of mDia2, VASP, and talin, respectively. **b** Line scans of the fluorescence intensities through the filopodia indicated by dashed boxes in images in (**a**). Intensities of the myosin X, mDia2, VASP, and formin were normalized to their maximal values at the filopodia tips. **c** Visualization of mApple-myosin X (red), myosin II RLC-GFP (green), and actin (mTagBFP-Lifeact, blue) in HeLa-JW cell. **d-e** Zoomed images of bipolar myosin IIA filaments at the bases of filopodia. **d** Myosin IIA and myosin X labeled as indicated in (**c**). **e** Cos-7 cell expressing mApple-myosin X (red) and myosin IIA-GFP heavy chain (green). Arrows indicate the doublets of fluorescent spots corresponding to heads of myosin II mini-filaments. **f-i** Images of Cos-7 cells expressing various constructs of myosin X (red) and myosin II (green): **f** GFP-myosin X, **g** mApple-myosin X and myosin IIA-GFP heavy chain, **h** mApple-myosin X and GFP-myosin IIA N93K, **i** mApple-myosin X and Emerald-myosin IIB heavy chain. Note, that both myosin IIA wild-type- and N93K-containing filaments, but not myosin IIB, can be seen at the bases of filopodia. See also Supplementary Movies 1–5, which correspond to images (**c**, **f-i**), respectively. Images were collected using spinning disk confocal microscopy (SDCM) (**a**, **f**) and structural illumination microscopy (SIM) (**c-e** and **g-i**). Scale bars, 5 μm (**a**), 2 μm (**c-i**)

Movie 8) showing that binding of myosin X to integrin via FERM domain[18] is not required for myosin IIA-driven enhancement of filopodia lifetime.

Finally, expression of Emerald-myosin IIB heavy chain in Cos-7 cells did not affect the filopodia lifetime (Fig. 2e, f) consistent with the lack of localization of the myosin IIB to the filopodia bases in these cells (Fig. 1i, Supplementary Movie 5).

**Pulling force induces growth of matrix-attached filopodia.** The lifetime of filopodia attached to the ECM is a complex parameter, which depends on filopodia adhesion to the matrix and protrusion activity of adherent filopodia. To observe the dynamics of filopodia adhesion and protrusion under controlled experimental conditions, we monitored the growth of filopodia that were adhered to fibronectin-coated beads trapped by optical tweezers (Fig. 3a, see Methods for more details). These experiments were performed on HeLa-JW cells, which generated distinct filopodia extending from stable cell edges, more suitable for optical tweezers manipulation than filopodia of Cos-7 cells. Two micrometre diameter fibronectin-coated polystyrene beads were placed onto filopodia tips by the optical tweezers. About 20–30 s was sufficient for the initial attachment of the bead to the filopodium, since after that time the bead remained attached to the filopodium tip even after the optical trap was switched off. After this attachment test, the movement of microscope piezo stage in the direction from the tip to the base of filopodium was initiated (Fig. 3b, Supplementary Movies 9–11). Immediately after attachment of the bead to the filopodium tip, a portion of myosin X demonstrated transient retrograde movement with an approximate velocity of 31 ± 5 nm/s (mean ± s.e.m., $n = 42$, 6 cells) (Fig. 3c, Fig. 6b). The original amount of myosin X at the tip was fully restored after several minutes (Fig. 3c, top panel, Supplementary Movie 10).

The pulling force exerted by filopodium on the bead was monitored by measuring the bead displacement from the center of the trap ($\Delta X$). In order to preserve the structural integrity of the filopodia, the velocity of the stage movement was set to approximately 10–20 nm/s, which is slower than the average velocity of spontaneous filopodia growth in these cells. With this setup we observed sustained filopodia growth for more than 10 min, during which time the F-Tractin-tdTomato labeled actin core remained intact (Figs. 3b, 4a, Supplementary Movie 10). During the first 3 min after stage movement commenced, the exerted force approached 3–5 pN, then dropped to the 1.5–2 pN for a further 1–3 min, after which it rapidly increased again (Fig. 3c, middle and bottom panels). In a typical experiment, we detected 2–4 such peaks with a mean peak force value of 3 pN (Fig. 3c inset) alternating with the 1–3 min periods of lower force (1.5–2 pN).

Filopodia growth in these experiments continued until one of three type of events occurred: (i) withdrawal of the bead from the trap by filopodium, (ii) detachment of filopodium from the bead detected by returning of the bead to the center of the trap, and

(iii) formation of a membrane tether lacking F-actin between the bead and filopodium tip (Fig. 4a–d and Supplementary Movies 10–16). The fractions of these outcomes for each type of treatment are represented on pie charts (Fig. 4e) by red, blue, and white, respectively.

Pulling-induced filopodia growth was depended on integrin-mediated adhesion of filopodia tips to fibronectin-coated beads. When the beads were coated with concanavalin A instead of fibronectin, application of force never induced the growth of filopodia actin cores. Instead, an immediate detachment of filopodia tips from the beads, withdrawal of the beads from the trap, or, in majority of cases, formation of membrane tethers (Fig. 4e, Supplementary Movie 12) occurred.

**Role of myosin IIA in filopodia adhesion.** We further studied how the presence and activity of myosin IIA affects force-induced filopodia growth and adhesion. The function of myosin II was suppressed in HeLa-JW cells by three different methods: through the inhibition of ROCK by Y-27632, by siRNA-mediated knockdown of myosin IIA heavy chain (siMIIA HC), and through the inhibition of myosin II ATPase activity by light-insensitive S-nitro-blebbistatin.

Inhibition of ROCK blocks myosin II regulatory light chain (RLC) phosphorylation, which interferes with myosin II filament assembly[28,34–36]. HeLa-JW cells treated with 30 μM of ROCK inhibitor Y-27632 lost their myosin II filaments in less than half an hour. siMIIA HC knockdown also resulted in a loss of most of the myosin II filaments (Supplementary Fig. 1). Inhibition of myosin II ATPase activity by S-nitro-blebbistatin did not disrupt myosin II filaments[28], although this treatment did result in profound changes to the organization of the actomyosin cytoskeleton, including a loss of stress fibers. Myosin X-positive comet tails persisted at the tips of filopodia in HeLa-JW cells irrespective of inhibition or lack of myosin IIA.

While the morphological integrity of filopodia was preserved in myosin II inhibited or depleted cells, adhesion of filopodia to the ECM was significantly impaired. First, in all cells with suppressed myosin II function, the prevalent outcome of pulling experiments (more than 70%) was detachment of the tip of filopodium from the bead (Fig. 4b–e). The time intervals between the start of pulling and bead detachment were, on average, shorter in myosin II impaired cells than in control ones (Fig. 4b–e). Furthermore, while fibronectin-coated bead pulling experiments revealed that the maximal forces exerted on the bead approached ~5 pN in filopodia from control cells (Figs. 3c, 4a, f), in cells with impaired myosin II activity these forces were in a range of 2 pN (Fig. 4b–f, Supplementary Movies 13 and 14). We also examined the immediate effect of Y-27632 during the pulling-induced sustained growth of filopodia. Shortly after the drug was added to the experimental chamber, filopodia detached from the bead after a transient increase of the force (Supplementary Fig. 2a, b, Supplementary Movie 15). Altogether, these results suggest that

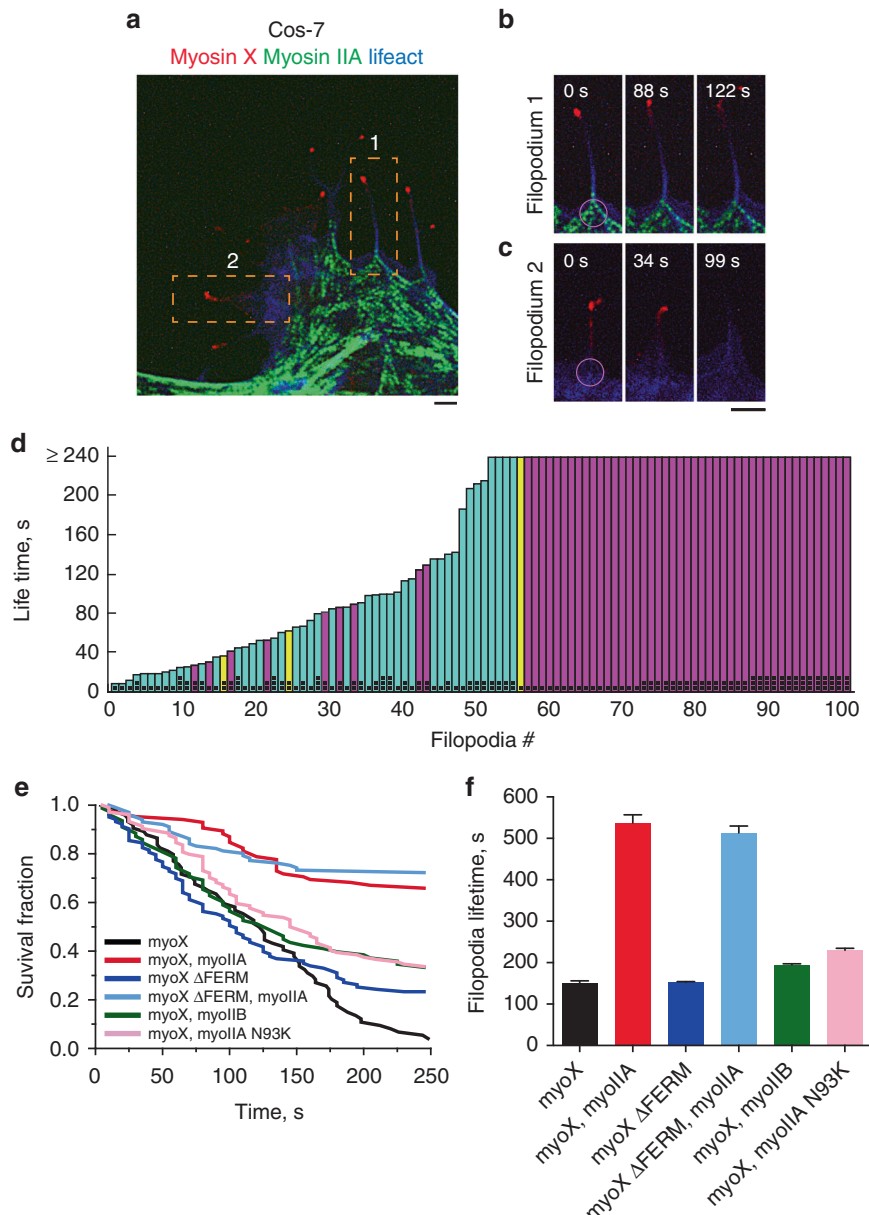

**Fig. 2** Effects of myosin IIA on lifetime of myosin X-induced filopodia. **a** A region of Cos-7 cell with myosin X positive filopodia either associated or non-associated with myosin IIA filaments. The cell expressed mApple-myosin X (red), myosin IIA-GFP heavy chain (green), and mTagBFP-Lifeact (blue). Dashed boxes outline filopodia associated (1) and nonassociated (2) with myosin IIA filaments. See also Supplementary Movie 6. **b**, **c** High magnification of frames showing filopodium 1 (**b**) and 2 (**c**) boxed in (**a**) at different time points. The purple 1.5 μm diameter circles at the left-most frames illustrate how the presence of myosin IIA filaments at the filopodia bases was assessed (see Methods). Images were collected using SIM. Scale bars 2 μm. **d** Lifetime values of filopodia associated and nonassociated with myosin IIA filaments. Pooled data showing 102 filopodia from 3 cells (including the cell shown in (**a**)). Each bar corresponded to individual filopodium; the bars labeled by one, two, or three black dots represent filopodia taken from first (shown in (**a**) and (**b**)), second and third cell, respectively. The bars are arranged in ascending order according to filopodia lifetime. The filopodia, which survived longer than the observation period of 240 s, are shown as bars with the uniform maximal height (240 s). Color code: filopodia nonassociated with myosin IIA filaments—cyan, filopodia associated with myosin IIA filaments part time—yellow, filopodia associated with myosin IIA filaments during entire period of observations—magenta. **e** Dynamics of survival fractions of the filopodia cohorts in Cos-7 cells transfected with the indicated constructs. Number of assessed filopodia (*n*) and cells for each type of transfection were: myo X (*n* = 72, 3 cells), myoX, myoIIA (*n* = 86, 3 cells), myoX ΔFERM (*n* = 104, 3 cells), myoX ΔFERM, myoIIA (*n* = 100, 3 cells, myoX, myoIIB (*n* = 93, 3 cells) and myoX, myoIIA N93K (*n* = 104, 4 cells). **f** Average lifetimes of filopodia (mean ± s.d.) calculated by fitting graphs shown in (**e**) to exponential decay function

filopodia are unable to establish a proper adhesion contact with the fibronectin matrix in the presence of inhibitors of myosin II mechanochemical activity or in the absence of myosin IIA filaments.

In addition to the studies of filopodia growth in response to pulling forces, we examined the effects of myosin II inhibition on the dynamics of free, unconstrained filopodia (Fig. 5). We found

that knockdown of myosin IIA or cell treatment with Y-27632 or S-nitro-blebbistatin significantly reduced elongation and retraction rates of filopodia, growing along the fibronectin-coated substrate. Of note, filopodia that originated from the upper cell surface and did not attach to the substrate were apparently unaffected by treatments impairing myosin II activity.

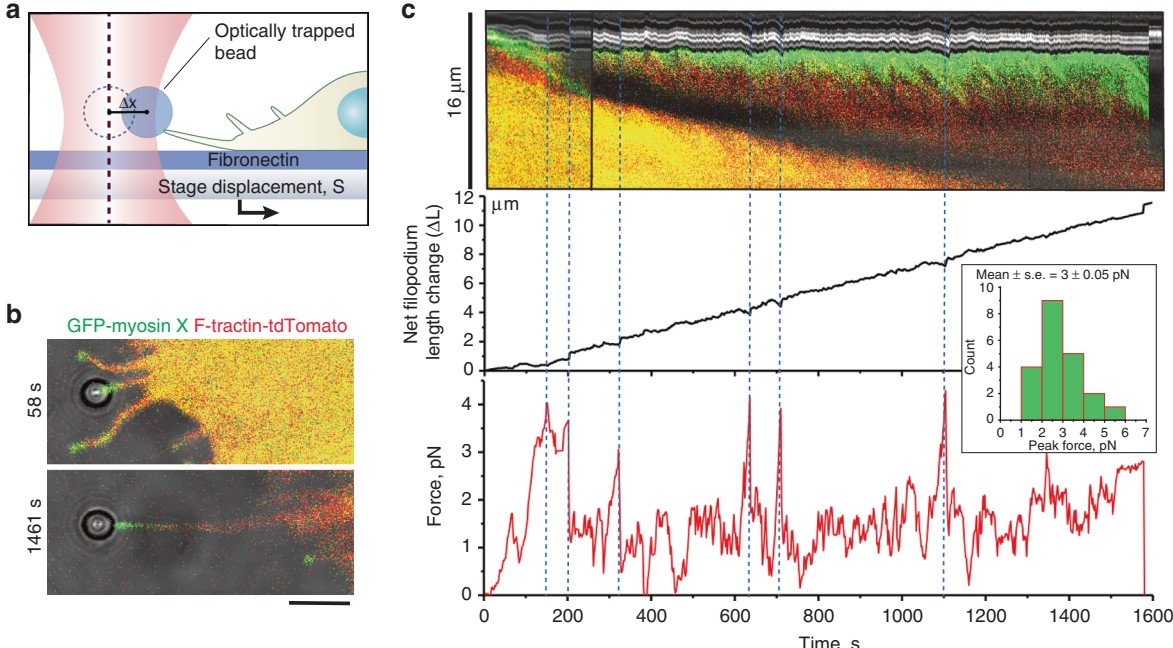

**Fig. 3** Dynamics of pulling-induced filopodia growth. **a** Experimental setup used to observe force-induced filopodia growth. Optical tweezers were used to trap fibronectin-coated microbeads attached to filopodia tips of HeLa-JW cells. **b** Confocal images of a typical cell expressing GFP-myosin X and F-Tractin-tdTomato with an attached bead, taken immediately after starting of stage movement (top) and in the course of sustained growth (bottom). Note that both myosin X and actin remain at the filopodium tip during growth. See also Supplementary Movies 9 and 10. Scale bar, 5 μm. **c** Top panel: A kymograph showing the dynamics of myosin X and actin in the filopodium shown in (**b**). This kymograph is composed of two parts following the change of the filopodium direction at 256 s from the beginning of observation. Middle panel: The length of new pulling-induced segment of filopodia is plotted versus time. Bead position in the coordinate system associated with moving microscope stage is indicated. The origin of the coordinate system corresponds to the bead position in the center of the laser trap at the initial time point. The net filopodium length change ($\Delta L$) was calculated as $\Delta L = S - \Delta X$, where $S$ is the microscope piezo stage displacement from its initial position, while $\Delta X$ is the deviation of the bead from the center of the optical trap. Bottom panel: Forces experienced by the bead. Note the discrete peak force values corresponding to the moments of filopodia growth cessation (seen in the middle panel) as marked with dotted lines. Inset: The distribution of peak force values, based on the pooled measurements of 21 peaks from 6 beads. Graphs were obtained by Origin software package

**Actin-formin interaction is required for filopodia adhesion.** As mentioned before, in myosin X-induced filopodia of HeLa-JW cells, the formin mDia2 is localized to the filopodia tips, and overlaps with myosin X patches (Fig. 1a, b, top panels). Small molecular inhibitor of formin homology domain 2 (SMIFH2)[37] was used to investigate the role of formins in attachment of filopodia to fibronectin-coated beads. We found that in SMIFH2-treated cells (40 μM, 1 h), filopodia behavior mimics that observed in myosin II inhibited/depleted cells. Majority of filopodia from SMIFH2 treated cells also detached from the beads rather than withdrew them from the trap (Fig. 4d, e, Supplementary Movie 16). The duration of contacts between filopodia and beads was significantly shorter (Fig. 4d, e), and the maximal force exerted by filopodia to the bead was significantly weaker than in control cells (Fig. 4f). Similarly, the growth of unconstrained filopodia was inhibited in cells treated with SMIFH2 as compared to control cells (Fig. 5).

While the number of filopodia visualized by actin labeling in myosin X expressing cells treated with SMIFH2 remained the same as in control cells ($0.36 \pm 0.01$ (mean ± s.e.m., $n = 34$ cells) and $0.39 \pm 0.02$ (mean ± s.e.m., $n = 31$ cells) per micrometre of cell boundary, respectively, the difference was insignificant according to unpaired two-tailed $t$ test with Welch's correction, $p = 0.17$), only about 25% of these filopodia preserved myosin X comet tails at their tips 1–2 h following SMIFH2 addition (Supplementary Fig. 3). We found that SMIFH2 induced disintegration of the comet tails into myosin X patches, which rapidly moved centripetally toward the cell body (Fig. 6a top and

middle panel, 6b, Supplementary Movies 17 and 18). Such movement was sometimes observed in filopodia of control cells (Fig. 3c), where it proceeds with the same velocity as retrograde movement of actin visualized by photoactivated PAGFP-β-actin construct (Supplementary Fig. 4, Supplementary Movie 19). In filopodia of cells treated with SMIFH2 the retrograde movement of myosin X patches was much more prominent, and led to the gradual disappearance of myosin X from the filopodia tips (Fig. 6a middle). Patches of VASP co-localized with the patches of myosin X and moved centripetally together with them in SMIFH2-treated cells (Supplementary Fig. 5, Supplementary Movie 20).

The velocity of retrograde movement of myosin X patches in filopodia of cells treated with SMIFH2 was $84 \pm 22$ nm/s (mean ± s.e.m., $n = 45$) (Fig. 6b) versus $31 \pm 5$ nm/s (mean ± s.e.m., $n = 42$) in control filopodia pulled by the beads (Figs. 3c, 6b). This rapid movement of myosin X patches in filopodia of SMIFH2-treated cells might be a result of the detachment of myosin X-bearing actin filaments from the filopodia tips. Once free, their subsequent retrograde movement is driven by myosin II located at the bases of the filopodia. Indeed, incubation of SMIFH2-treated cells with Y-27632 efficiently stopped the retrograde movement of the myosin X-positive patches, reducing their average velocity to $0.66 \pm 0.14$ nm/s (mean ± s.e.m., $n = 17$), see Figs. 6a (bottom panel), 6b and Supplementary Movies 17, 18, and 21.

We also performed an experiment with addition of SMIFH2 to the filopodium attached to the bead, during the force-induced

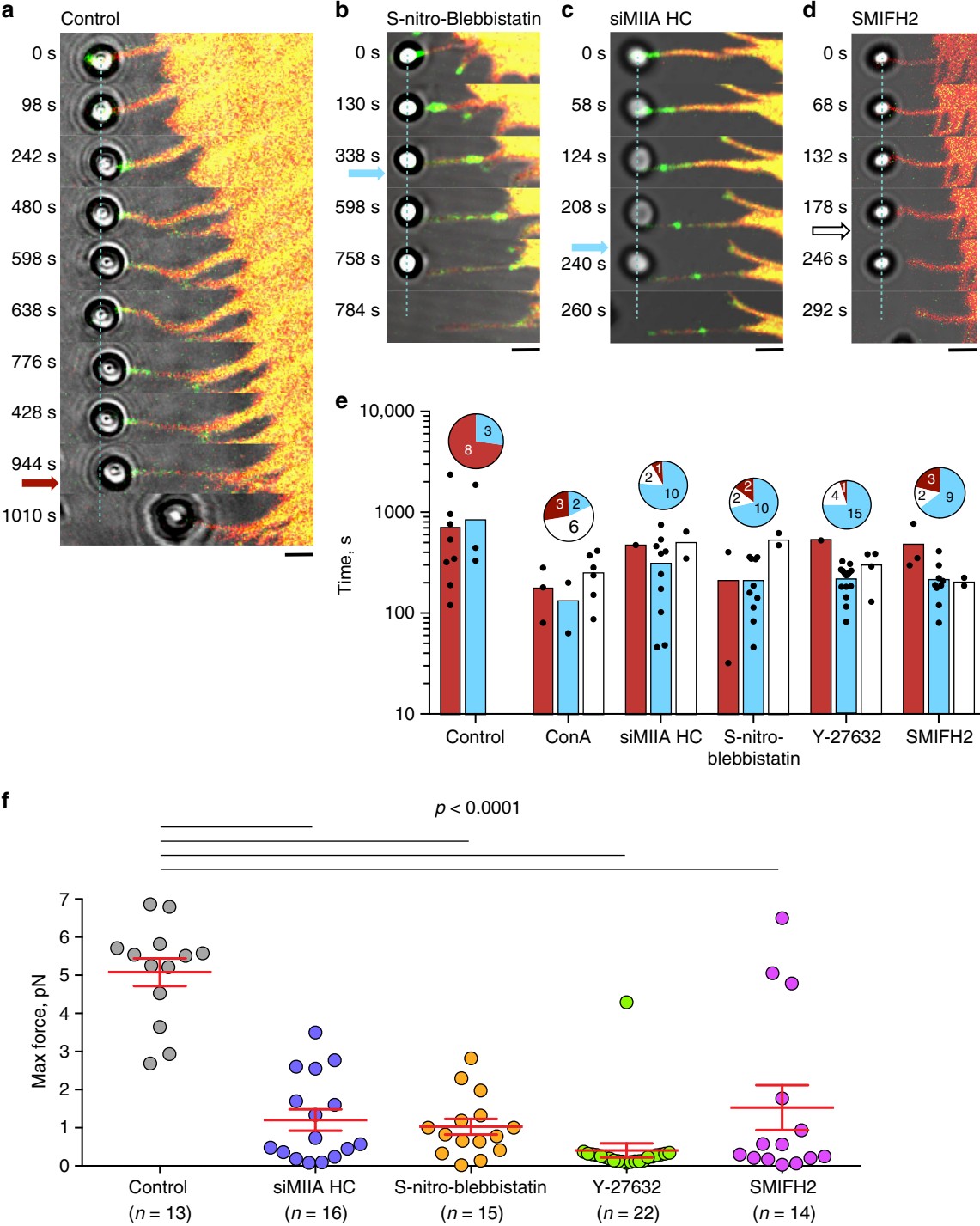

sustained growth of filopodia. Within about 10 s after addition of the drug, myosin X patch disintegration, disappearance of actin from the tip and appearance of membrane tether instead were observed. The force generated by a filopodium dropped to a value of zero in ~3 min (Supplementary Fig. 2c, d, Supplementary Movie 22). The fibronectin-coated bead, however, remained associated with the filopodium tip via the membrane tether (Supplementary Movie 22), and did not detach even upon switching the trap off. This suggests that SMIFH2 treatment disrupted the link between actin core and the integrin adhesion receptors, which still remain associated with the membrane tether at the filopodium tip.

To prove that SMIFH2 treatment can indeed detach actin filaments from formin located at the filopodia tips, we performed

experiments where the actin filaments were growing from immobilized formin mDia1 construct containing FH1, FH2, and DAD domains of formin under fluid shear conditions[38] in the absence or presence of SMIFH2. Treatment with 100 μM SMIFH2 did not significantly affect the rate of actin filament elongation (35 ± 6 vs. 32 ± 8 subunits/s, mean ± s.d., $n = 40$ filaments, $p = 0.01$ according to unpaired two-tailed $t$ test with Welch's correction, without and with SMIFH2, respectively) in agreement with the original study (see Fig. 3 in Rizvi et al.[37], in which it was shown that 100 μM SMIFH2 reduced the fraction of formin-associated filaments but not their elongation rate). At the same time, addition of 100 μM SMIFH2 resulted in a rapid decrease in the fraction of filaments remaining associated with

**Fig. 4** Inhibition of myosin II or formin reduces filopodia adhesion. **a–d** Growth of filopodia attached to laser-trapped beads upon stage movement in HeLa-JW cells treated as indicated (see Supplementary Movies 11, 13, 14, and 16). Cells were transfected with GFP-myosin X (green) and F-Tractin-tdTomato (red). Position of the center of laser trap is indicated by blue dashed line. **a** At 944 s filopodia started to retract and at 960 s (dark red arrow) pulled the bead out of the trap. **b–f** Filopodia in cells treated with: 20 µM of S-nitro-blebbistatin (**b**); myosin IIA siRNA (**c**); 40 µM of SMIFH2 (**d**). The detachment of the filopodium from the bead, detected by returning of the bead to the center of the trap, occurred at 404 s (**b**), 230 s (**c**) after starting the stage movement (blue arrows). Upon switching off the trap (at 760, 256 s, respectively) the detached beads disappeared from the field of view (**b**, **c**). In (**d**) the return of bead to the center of trap at 200 s (white arrow) was accompanied by formation of membrane tether and the bead remained within the field of view after switching off the trap at 284 s (Supplementary Movie 16). **e** Statistics of the outcomes of the pulling experiments. Color pie charts represent the fractions of outcomes for each treatment. Red—bead withdrawal from the trap, blue—filopodia detachment from the bead, white—formation of membrane tether. The numbers of assessed filopodia indicated in each slice. Each dot on the graphs underneath the pie charts represents time interval between start of pulling and corresponding outcome for individual filopodium in cell treated as indicated; color-coded bars represent the mean values. The filopodia were attached to fibronectin-coated or ConA-coated beads as indicated. Concentration of Y-27632 was 30 µM. **f** Peak values of the forces exerted by filopodia on the fibronectin-coated beads during the stage movement. Mean values (horizontal lines) and s.e.m. (error bars) are shown. Numbers of measured filopodia (n) are indicated. p Values calculated according to unpaired two-tailed t test with Welch's correction were all less than 0.0001

immobilized formins under conditions of mild shear flow (Fig. 7a, Supplementary Movies 23 and 24). Fitting the curve of the fraction of the actin filaments that remain associated with immobilized formin (survival fraction) during a period of 400 s (Fig. 7b) with single exponential decay fraction revealed that SMIFH2 treatment increased the $k_{off}$ from $(9.3 \pm 1.5) \times 10^{-05} \text{ s}^{-1}$ (mean ± s.d., $n = 40$) for control to $(1.0 \pm 0.16) \times 10^{-03} \text{ s}^{-1}$ (mean ± s.d., $n = 40$) for SMIFH2-treated filaments. In addition, to quantitatively estimate the difference between survival curves for control conditions and SMIFH2 treatment, the nonparametric logrank test[39] was used, which revealed the highly significant difference ($p = 0.0009$). Thus, SMIFH2 treatment disrupted physical contacts between formin molecules and the actin filaments.

Altogether, our in vitro and in vivo data suggest that SMIFH2-induced rapid centripetal movement of myosin X occurs due to detachment of actin filaments with associated myosin X molecules from the filopodia tips, and is driven by myosin II-mediated pulling of these filaments towards the base of filopodium. This, in turn, suggests that in nontreated cells formin participates in transduction of the force generated by myosin II at the filopodium base to the filopodium tip.

**Regulation of filopodia adhesion by the substrate rigidity**. To check whether filopodia adhesion mechanosensitivity can be used by cell for its orientation during migration, we compared the behavior of unconstrained filopodia on a rigid substrate with that on fluid supported lipid bilayer (SLB) where the traction forces cannot develop[40]. To this end, we created a composite substrate, on which a rigid surface was covered by orderly patterned small islands ($D = 3$ µm) of SLB. Both rigid and fluid areas were coated with the integrin ligand, RGD peptide, at the same density (Supplementary Fig. 6). We found that filopodia encountered with the SLB islands could not attach properly and as a result spent a significantly shorter time over the SLB substrate than over a rigid area of the same geometry (Fig. 8a, b, Supplementary Movie 25). Thus, we demonstrate that proper adhesion of filopodia can be prevented not only by inhibition of myosin II or formin, but also by micro-environmental conditions under which filopodia tips do not develop traction force.

**Other types of filopodia**. All experiments described in Results were performed on filopodia induced by overexpressing myosin X in HeLa-JW or Cos-7 cells. The results related to filopodia of other types (induced by constitutively active Cdc42 or mDia2 formin) are presented in Supplementary Information.

## Discussion

In the present study, we have demonstrated that adhesion of myosin X-induced filopodia to the extracellular matrix depends on the forces generated by associated myosin IIA filaments. Our interest in filopodia induced by myosin X is in part justified by the fact that this protein is overexpressed in many types of cancer cells and adhesion of myosin X containing filopodia to the matrix may play an important role in the cancer cells invasion and metastasis[41–43].

SIM revealed the presence of individual myosin IIA filaments at the bases of myosin X-induced filopodia in HeLa-JW and Cos-7 cells. In particular, experiments with Cos-7 cells, which do not contain endogenous myosin IIA, demonstrated that filaments containing exogenous myosin IIA, but not IIB, are associated with filopodia.

The dependence of filopodia adhesion and growth on myosin IIA-generated forces was revealed in two types of experimental systems. First, filopodia of Cos-7 cells with exogenous myosin IIA filaments at their bases have a longer lifetime than filopodia lacking myosin IIA filaments. We compared lifetime durations of myosin IIA associated and nonassociated filopodia in the same cell and found that the effect of myosin IIA filaments on filopodia is local and not related to overall activation of cell contractility upon myosin IIA overexpression. Myosin IIB overexpression does not increase the filopodia lifetime and this myosin does not localize to filopodia bases. At the same time, myosin IIA mutant with compromised motor activity (myosin IIA N93K)[31] localizes to filopodia bases, but only slightly enhances the filopodia lifetime. Thus, the local motor function of myosin IIA is required for augmentation of filopodia lifetime.

This increase of the filopodia lifetime can be explained by stabilization of adhesion of the filopodia to the substrate and promotion of growth of adherent filopodia. Therefore, our second experimental system was designed to follow the pulling force-induced growth of filopodia that were adhered to fibronectin-coated beads trapped by optical tweezers. This system permitted us to investigate key molecular players in force dependent filopodia adhesion and growth. In control HeLa-JW cells, filopodia attached to optically trapped fibronectin-coated beads underwent growth upon slow movement of the microscope stage and developed periodic 3–5 pN forces. Such forces can be in principle developed by individual myosin IIA filaments (Supplementary Note 2). The average duration of growth was about 13 min, after which majority of filopodia withdrew the bead from the optical trap, thereby preserving the adhesion integrity. Myosin IIA-depletion or cell treatment with inhibitors of myosin II assembly (Y-27632) or motor activity (blebbistatin) led to significant decrease of the forces exerted by filopodia and shortening of the duration of filopodia growth upon pulling.

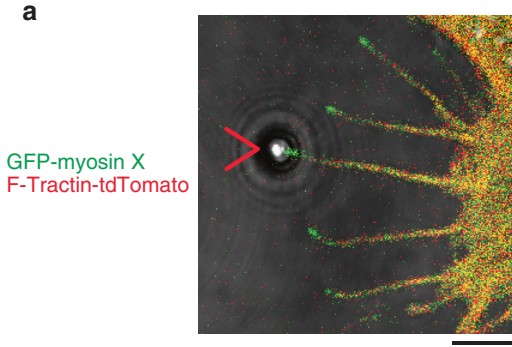

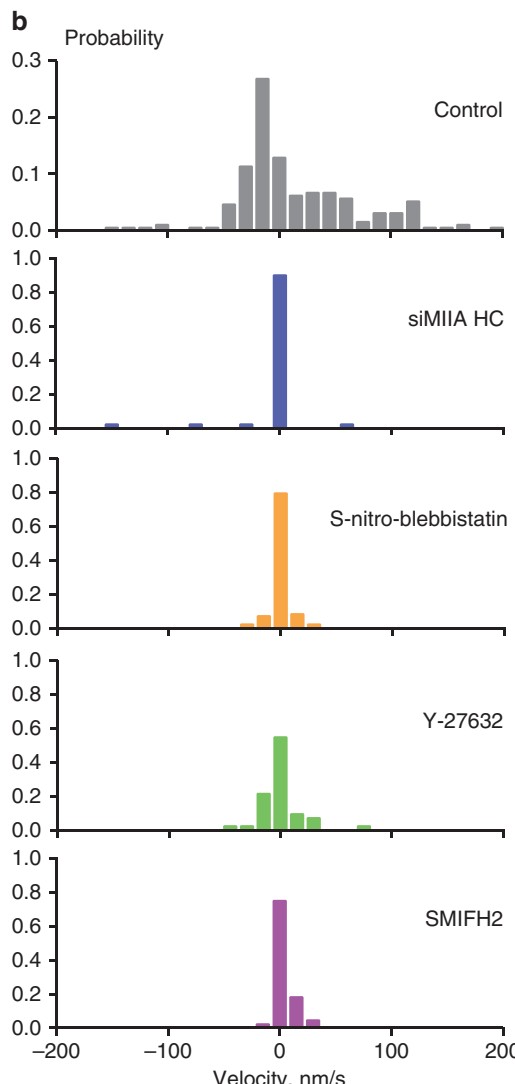

**Fig. 5** Dynamics of unconstrained filopodia Inhibition of myosin II or formins interfered with growth of unconstrained filopodia. **a** A cell expressing GFP-myosin X and F-Tractin-tdTomato representative of those used in experiments assessing filopodia growth. The filopodium attached to the laser trapped fibronectin-coated bead is indicated by the red arrowhead. Such filopodia were excluded from the score. Scale bar 2 μm. **b** Histograms showing the probability distribution of growth/retraction velocities (nm/s) of unconstrained filopodia in HeLa-JW cells, measured in the same experiments as those assessing growth of filopodia attached to the fibronectin-coated beads. Positive values of the velocities correspond to growth, and negative—to retraction of filopodia. Probabilities represent the mean fractions of time, during which the velocity value belonged to a given interval with a width 15 nm/s. The numbers of assessed filopodia (with number of cells in parenthesis) were 24 (5), 24 (4), 43 (6), 24 (5), and 42 (8) for (from top to bottom) control (gray), myosin IIA HC siRNA knockdown (blue), S-nitro-blebbistatin (orange), Y-27632 (green), and SMIFH2 (purple) treatments, respectively

Since myosin II is located at the bases of filopodia, a question requiring further clarification is how the pulling force is transmitted to the filopodia tips. Our data suggest that this force is transmitted by the filaments of the actin core attached to formin molecules at the filopodia tips. We have shown that formin inhibition by SMIFH2 suppresses filopodia adhesion to the beads in the same manner as inhibition of myosin II. SMIFH2 treatment efficiently detaches actin filaments from formin molecules in vitro and most probably detaches actin core filaments from the tips of filopodia, which triggers rapid retrograde myosin II dependent movement of myosin X and VASP patches. Altogether, our experiments suggest that myosin II inhibition, or

inhibition of the formin-mediated association between actin filaments and the filopodia tips, result in filopodia that are unable to form stable adhesions with fibronectin-coated beads.

Formins could function in force/myosin IIA dependent filopodia adhesion and growth not only as linkers between actin filaments and filopodia tips. Recent studies demonstrate that formin-driven actin polymerization can be enhanced by pulling forces[38,44–46]. Thus, myosin II-generated force transmitted via actin core to formins at the filopodium tip can in principle stimulate actin polymerization, promoting filopodia growth and reinforcing adhesion. Such a mechanism is not addressed in the present study and deserves further investigation.

Force-driven growth of filopodia attached to the bead is an integrin-dependent process and was not observed in experiments with integrin-independent adhesion of filopodia to beads coated by concanavalin A. A major linker between integrin and actin filaments, talin, has been detected at the filopodia tips in this and other publications[19,27]. Previously, it was established that force-driven unfolding of talin facilitates interaction of talin with another adhesion complex component, vinculin, resulting in reinforcement of the association between talin and actin filaments[47–50]. Since vinculin enrichment was detected in shafts but not tips of filopodia[51], the question whether this mechanism is applicable to filopodia adhesion reinforcement requires additional studies. A protein, RIAM, that competes with vinculin for talin-binding and localizes to filopodia tips[19] is mainly involved in activation of β2 integrin[52]. Another mechanosensory protein involved in integrin adhesion, p130cas[53], could also play a role in integrin signaling in filopodia[54]. In some types of filopodia the attachment of filopodia tips to the substrate was shown to generate a signal (possibly force-dependent), which, instead of further elongation of filopodia, triggers the formation of associated lamellipodia[55,56].

Force dependence of filopodia adhesion and dynamics was documented here mainly for filopodia induced by overexpression of myosin X. Our pilot experiments revealed that filopodia induced by constitutively active mDia2 formin or Cdc42 demonstrate different morphology and dynamics (see Supplementary Note 1, Supplementary Fig. 7, Supplementary Movies 26 and 27). It was suggested that myosin X is required for the delivery of integrins to the filopodia tips[18]. However, association of integrin with myosin X FERM domain was dispensable for the myosin IIA-driven augmentation of the filopodia lifetime in Cos-7 cells. This suggests that the link between integrin and myosin X is not critically important for force-dependent filopodia adhesion stabilization. An interesting possibility might be a potential

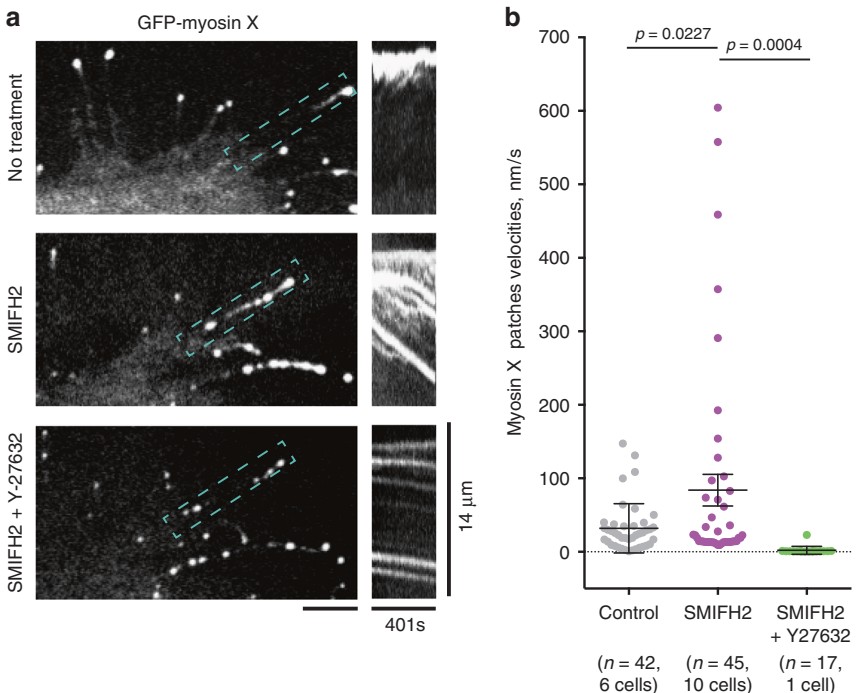

**Fig. 6** Formin inhibitor SMIFH2 promotes intrafilopodial centripetal movement. **a** Filopodium of HeLa-JW cell is shown before SMIFH2 treatment (top panel), 15 min following the addition of 20 μM SMIFH2 (middle panel) and 15 min after subsequent addition of 30 μM Y-27632 (bottom panel). Myosin X patches are shown in the left images (see also Supplementary Movies 17, 18, and 21), and kymographs representing the movement of the patches along the boxed filopodium—in the images on the right. Note that treatment with SMIFH2 resulted in formation of numerous myosin X patches moving from the tip to the base of filopodium. Addition of ROCK inhibitor Y-27632 stopped this movement. Images for analysis were obtained with SDCM. Scale bars, 5 μm. **b** Graph displaying the myosin X patches velocities in filopodia of control cells (left, gray dots), of cells treated with SMIFH2 for 15 min (middle, purple dots), and in SMIFH2-treated cells 15 min following the addition of Y-27632 (right, green dots). Each dot corresponds to individual myosin X patch; numbers of analyzed patches (n) and cells, as well as significance of the differences (p values) are indicated. The p values were calculated using unpaired two-tailed t test with Welch's correction

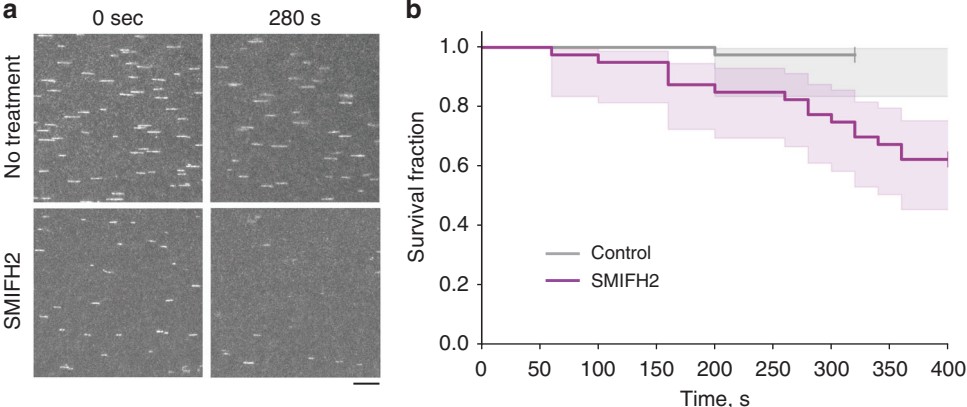

**Fig. 7** SMIFH2 enhances the detachment of actin filaments from mDia1 formin in vitro. **a** The constitutively active mDia1 formin construct (FH1FH2DAD) was anchored to the glass surface of a microfluidic chamber by one of their FH2 domains using an anti-His antibody (as in ref. [38], see Methods). Actin filaments were grown first in the presence of Alexa488-labeled actin to form fluorescent segments at their tips. Once the formin had nucleated a filament, it was exposed, from time zero onward, to 1 μM unlabeled actin and 4 μM profilin, in the absence or presence of 100 μM SMIFH2. Images show frames corresponding to the beginning (time zero, left column) and 280 s (right column) following addition of unlabeled actin into the flow chamber in the absence (top row) and presence (bottom row) of SMIFH2 inhibitor in solution. The representative fragments of epifluorescent image field of view of the labeled segments of actin filaments are seen. The number of filaments decreased with time due to their detachment from immobilized formin (see Supplementary Movies 23 and 24). Scale bar, 10 μm. **b** The time at which each filament detached was recorded and the survival fraction of the filaments at each time point was calculated. See Methods for more details. Graphs of the survival fraction curves with 95% confidence interval shaded area around the actual data are presented. To estimate the significance of difference between the two survival fraction curves, the logrank test[39] was used. Calculated p value (Python software) was equal to 0.0009 (***)

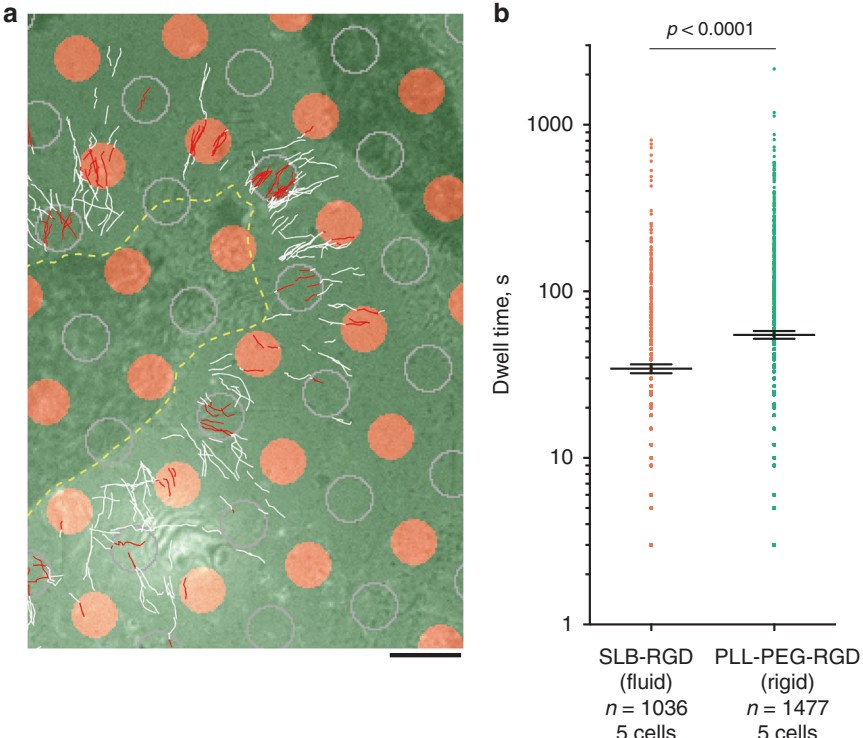

**Fig. 8** Attachment of filopodia to RGD-coated rigid and fluid substrate. **a** HeLa-JW cells expressing GFP-myosin X were plated on micropatterned coverslips covered with circular islands ($D = 3\,\mu m$) of SLB conjugated to RGD (orange circles), organized into a square lattice. The glass between the islands was covered with poly-L-lysine-PEG (PLL-PEG) conjugated to RGD at the same density (Supplementary Fig. 6). Trajectories of GFP-positive filopodia tips acquired during a 14–36 min time interval are shown. The cell border is shown by a yellow dashed line. For comparison of the trajectories on rigid and fluid substrates, the circles of similar diameter were drawn by computer in the centers of the square lattice formed by SLB islands (outlined by gray contours). The segments of the trajectories located inside either the SLB islands or the drawn circles on the rigid substrate are shown in red, and the remaining parts of the trajectories are shown in white. See also Supplementary Movie 25. Scale bar, 5 µm. **b** Dwelling time that myosin X-positive filopodia tips spent inside fluid (orange) or rigid (green) circles defined above. Pooled data for 5 cells; $n$—number of filopodia tip trajectories analyzed. Aligned dot plots and mean ± s.e.m. values are shown. The $p$ value was calculated using unpaired two-tailed $t$ test with Welch's correction

collaboration between myosin X and actin polymerization machinery including VASP[26] and, perhaps, formins, but corresponding mechanisms remain to be elucidated.

Our finding that filopodia adhesion is force-dependent explains how filopodia could respond differently to substrates of varying stiffness. On a stiff substrate, the force generated by myosin II and applied to the adhesion complex will develop faster than on a compliant substrate[57]. Accordingly, filopodia adhesion should be more efficient on stiff substrates than on compliant substrates. Indeed, we showed that the contacts of filopodia with RGD ligands associated with fluid membrane bilayer were less stable than with the areas of rigid substrate covered with RGD of the same density. These considerations can explain involvement of filopodia in the phenomenon of durotaxis[23], a preferential cell movement towards stiffer substrates[58]. This may provide a mechanism to rectify directional cell migration.

Orientation based on filopodia adhesion is characteristic for several cell types, in particular for nerve cells. Application of external force can regulate the direction of growth cone advancement[59]. The growth cones of most neurites produce numerous filopodia, and the adhesion of these filopodia can determine the direction of neurite growth[60,61]. The results from these experiments can now be explained by preferential adhesion of filopodia, which experience larger force. Interestingly, the filopodia-mediated traction force in growth cones is myosin IIB-dependent[62], suggesting that preferential use of one or another myosin II isoforms for filopodia mediated mechanosensing could be cell type specific.

The mechanosensitivity of filopodia adhesion provides a mechanism of cell orientation that complements that mediated by focal adhesions. Focal adhesions are formed by cells attached to rigid two-dimensional substrates, whereas filopodia adhesion can be formed by cells embedded in three-dimensional fibrillar ECM network. Further investigation of filopodia mechanosensitivity could shed a new light on a variety of processes related to tissue morphogenesis.

## Methods

**Cell culture, transfection, and constructs**. Hela-JW, a subline of a HeLa cervical carcinoma cell line derived in the laboratory of J. Willams (Carnegie-Mellon University, USA) on the basis of better attachment to plastic dishes[63], was obtained from the laboratory of B. Geiger[64]. Cos-7 (African green monkey fibroblast-like cell line) was obtained from ATCC (ATCC® CRL-1651 ). Both cell lines were grown in Dulbecco's Modified Eagle Medium (DMEM) supplemented with 10% fetal bovine serum (FBS), 100 U ml$^{-1}$ penicillin/streptomycin, 2 mM glutamine, and 1 mM sodium pyruvate at 5% $CO_2$ at 37 °C. HeLa-JW cells were transfected with DNA plasmids by electroporation (two pulses of 1005 V for 35 ms) using the Neon transfection system (Thermofisher), while Cos-7 cells were transfected by jet-PRIME transfection reagent (Polyplus) according to the manufacturer's protocol. Both cell lines were tested for mycoplasma contamination. No cell lines used in this study were found in the database of commonly misidentified cell lines that is maintained by ICLAC and NCBI Biosample. We did not attempt to authenticate them. Expression vectors encoding the fluorescent fusion proteins used in this study are described in the Supplementary Note 3.

**Live cell imaging and confocal microscopy**. Following electroporation or chemical-based transfection, cells were seeded at a density of $2 \times 10^4$ cells ml$^{-1}$ in 2 ml onto 35 mm glass based dishes with a bottom base cover glass #1 either 12 or 27 mm in diameter (Iwaki, Japan) coated with 10 µg ml$^{-1}$ fibronectin

(Calbiochem) for 20 min. Cells were imaged in Leibovitz's L-15 medium without Phenol Red containing 10% FBS. Samples were mounted in a humidified cell culture chambers and maintained at 37 °C with 5% $CO_2$.

Snapshots or time-lapse images were acquired with a spinning-disc confocal microscopy system (PerkinElmer Ultraview VoX) based on an Olympus IX81 inverted microscope, equipped with a 100× oil immersion objective (1.40 NA, UPlanSApo), an EMCCD camera (C9100-13, Hamamatsu Photonics), and Volocity control software (PerkinElmer).

Photoactivation with instant imaging was performed by activation of a defined region inside filopodia of live cells using a blue diode laser (405 nm, 100 mW) on a CSU-W1 spinning disk confocal system attached to a Nikon Eclipse Ti-E inverted microscope with Perfect Focus System, controlled by MetaMorph software (Molecular device), supplemented with a 100× oil 1.45 NA CFI Plan Apo Lambda oil immersion objective and sCMOS camera (Prime 95B, Photometrics).

For SIM, two types of equipment were used: (1) Live SR (Roper Scientific) module on a Nikon Eclipse Ti-E inverted microscope (specifications of setup described above), (2) Nikon N-SIM microscope, based on a Nikon Ti-E inverted microscope with Perfect Focus System controlled by Nikon NIS-Elements AR software supplemented with a 100× oil immersion objective (1.40 NA, CFI Plan-ApochromatVC) and EMCCD camera (Andor Ixon DU-897).

All SIM images with obtained with system (1) except for images on Fig. 1c, d, which were obtained with set up (2).

**Local presence of myosin IIA filaments at filopodia bases**. The presence of myosin IIA filaments at filopodia bases was determined manually by examining of all filopodia in time-lapse movies of Cos-7 cells transfected with mApple-myosin X and Myosin IIA-GFP constructs. The base of filopodium were defined as a space occupied by the circle with the diameter of 1.5 µm, which is internally tangent to the plasma membrane at the area of negative curvature manifesting the filopodia orifice. The myosin IIA filaments located inside this circle and above it (inside the filopodia) were scored as filopodia-associated (see Fig. 2b, c, left panels).

**Lifetime analysis**. The average lifetime of filopodia, $\lambda$, was calculated by fitting the time-dependent filopodia survival fraction to the exponential decay function, $e^{-t/\lambda}$, where $t$ is the time. The fitting was done by using the Levenberg–Marquardt algorithm built in Origin software package. Standard deviation (s.d.) of the average filopodia lifetime was estimated based on the standard error computed by the fitting algorithm.

**Transfection of siRNA and immunoblotting**. Cells were seeded into a 35 mm dish on day 0 and transfected with 50 nM of myosin IIA HC MYH9 ON-TARGET plus SMART pool siRNA (L-007668-00-0005, Dharmacon) using Screenfect^TM^A (WAKO, Japan) on day 1. Control cells were transfected with scrambled control ON-TARGET plus Non-targeting pool siRNA (D-001810-10, Dharmacon). Transfection of plasmid GFP-myosin X and F-Tractin-tdTomato was performed in the evening of day 1 using Jet Prime transfection reagent (Polyplus) and cells were imaged on day 2. For assessment of myosin IIA heavy chain expression, transfected cells were lysed in RIPA buffer on day 2 (exactly 24 h following siRNA transfection) and analyzed by Western blotting with primary rabbit antibodies to the myosin IIA tail domain (M8064, Sigma-Aldrich, dilution 1:1000); staining of α-tubulin with mouse monoclonal DM1A antibody (T6199, Sigma-Aldrich, dilution 1:5000) was used as a loading control. HRP-conjugated anti-rabbit IgG (Bio-Rad, 1706515, dilution 5000) and anti-mouse IgG (A4416, Sigma-Aldrich, dilution 1:10000) were used as secondary antibodies, respectively.

**Immunofluorescence antibody staining**. Cells were pre-fixed by addition of warm 20% PFA (Tousimis) into medium (to make a final 2% solution) and subsequent incubation for 15 min at room temperature. This was followed by fixation and permeabilization by 3.7% PFA, 0.2% glutaraldehyde, and 0.25% Triton X-100 in PBS for 15 min. The fixed cells were then washed two times with PBS and blocked with 5% bovine serum albumin (BSA) for 30 min. To assess endogenous level of myosin IIA cells were incubated with anti-myosin IIA tail domain (M8064, Sigma-Aldrich, dilution 1:800) 1 h at 4 °C, and 405 Alexa-Fluor-conjugated secondary antibodies (A31556, Molecular Probes, dilution 1:200) for 1 h at room temperature.

**Drug treatment**. For formin drug inhibition studies, cells were incubated with 20 or 40 µM SMIFH2 (4401, TOCRIS, UK) in serum containing DMEM for 1–2 h at 5% $CO_2$, 37 °C. In in vitro experiments, SMIFH2 (340316-62-3, Sigma-Aldrich) was used at a concentration of 100 µM (see below). For myosin II inhibition studies, 30 µM or 50 µM Rho-kinase (ROCK) inhibitor Y-27632 (Y0503, Sigma-Aldrich), or 20 µM S-nitro-blebbistatin (13013-10, Cayman Chemicals) was added for 10–20 min before the experiments or directly during observation. All inhibitors remained in the medium during the entire observation period.

**Optical tweezers and data acquisition**. All experiments involving filopodia pulling were carried out on a Nikon A1R confocal microscope adapted for the use of laser tweezers. Totally, 2.19 µm diameter polystyrene beads (PC05N, Bangs Laboratories) were coated with fibronectin (341635, Calbiochem) according to a

previous protocol[65]. For bead trapping we used an infrared laser ($\lambda = 1064$ nm, power 0.5–1 W, YLM-5-LP-SC Ytterbium Fiber Laser, IPG photonics).

To determine the forces, $F$, applied to the bead by the optical trap, we measured the displacement of the bead from the trap center, $\Delta X$, and with knowledge of the stiffness of the trap, $K$, the force was calculated as $F = K\Delta X$. The trap stiffness was calibrated using the equipartition method[66] by tracking the fluctuations of a bead trapped by optical tweezers, using an Andor Neo sCMOS camera, at 100 fps. The displacement of the beads from the center of the optical trap in confocal microscopy observations was monitored using a piA640–210gm camera (Basler) at 0.5–1 fps and Metamorph software for tracking. The smallest detectable bead displacement was ~5 nm, corresponding to the smallest force measured of ~0.04 pN. The moment of detachment of filopodia from the bead was detected by the bead returning to the center of the optical trap. To confirm the detachment, we then switched the trap off, which resulted in release of the bead and its disappearance from the field of view unless it remained connected to filopodia via the membrane tethers.

For laser trap experiments, HeLa-JW cells transfected by electroporation with GFP-myosin X and F-Tractin-tdTomato, were seeded at a density of $2 \times 10^4$ cells ml$^{-1}$ in 750 µl onto chambered #1 borosilicate cover glasses (155383, Lab-Tek) coated with fibronectin (341635, Calbiochem) by incubation in 1 µg ml$^{-1}$ solution in PBS for 20 min at 37.0 °C. A reduced concentration of fibronectin (compared to that used for regular cell observations) was used to prevent the beads sticking to the cover slip. Chambers with cells were mounted on a P-545.3R7 stage equipped with E-545 Plnano piezo controller (Physik Instrumente), which was moved in order to generate the pulling force between filopodia and trapped beads. The velocity of the stage movement was maintained in a range of 10–20 nm/s by PIMikroMove 2.15.0.0 software. The specimen was incubated in a custom-built microscope hood at 37.0 °C, 5% $CO_2$ humidified environment. Simultaneously with pulling, the cells were imaged using lasers $\lambda = 488$ and 561 nm for excitation of GFP and tdTomato, respectively. Collected experimental data were processed by particle tracking algorithms of the Metamorph software.

**In vitro assay for formin-dependent actin polymerization**. We used a microfluidics based assay to assess formin processivity in the course of formin-driven actin polymerization[38]. Formin construct Snap-mDia1(FH1FH2DAD)-6×HisTag was specifically anchored to the bottom surface of a microchamber, using a biotinylated pentaHis antibody (Qiagen) and streptavidin to bind to a biotin-BSA-functionalized glass surface which was further passivated with BSA. The microfluidics PDMS chamber height was 60 µm. Immobilized formin constructs were allowed to nucleate actin filaments by exposing them for 30 s to 2 µM 20% Alexa488-labeled Lys$^{328}$ actin[67] and 0.4 µM profilin in a buffer containing 10 mM Tris-HCl (Euromedex) pH 7.8, 1 mM MgCl2 (Merck), 200 µM ATP (Roche), 50 mM KCl (VWR Chemicals) and supplemented with 5 mM DTT (Euromedex), 1 mM DABCO (1,4-diazabicyclo[2.2.2]octane, Sigma-Aldrich) to reduce photobleaching. Recording of actin filament elongation was started after the buffer was changed to contain 1 µM unlabeled actin and 4 µM profilin, with or without 100 µM SMIFH2 (Sigma-Aldrich). The microfluidics flow was kept low enough to ensure that the viscous drag had no impact on filament detachment from formin. Images were acquired every 10 s on a Nikon Ti-E microscope using a 60× objective and a Hamamatsu Orca Flash 4.0 V2 + sCMOS camera. Actin was purified from rabbit muscle[68]. Recombinant Profilin I and Snap-mDia1(FH1FH2DAD)-6×His-Tag were expressed in *E. coli* and purified[69]. Formin elongation rates were obtained by measuring the slope of kymograph from individual filament traces[38]. Survival fraction curves representing the fractions of actin filaments remaining associated with immobilized formins were calculated for cohorts of 40 filaments for control and SMIFH2 treatment. The $k_{off}$ rates obtained by exponential decay fitting, and the significance of the difference between survival fraction curves by logrank test were calculated using the Python software.

**Unconstrained filopodia growth study**. The same image sequences, which were used in experiments with optical trap, were then analyzed to track the tips of filopodia, which were neighboring to filopodia analyzed in pulling experiments and hence not attached to the beads. To track filopodia tips, the Imaris 8.3.1 spots tracking procedure was used. The trajectories of unconstrained filopodia tips were manually segmented into periods of linear growth, shrinking, and pausing. Elongation and shrinking rates were obtained using the linear regression fitting function of Origin software package.

**Analysis of filopodia adhesion to rigid and fluid substrates**. The cleaned glass coverslips were coated with PLL-g-PEG-biotin (Susos) for 2 h followed by UV etching using 3 µm circular shape arrays of chromium (Cr) photomask[70], which resulted in removal of PLL-g-PEG-biotin polymers on the etched circular regions. After multiple rinses with UHQ water, phospholipid vesicles were introduced to the surface for 5 min, allowing the self-assembly of lipid bilayer on the etched glass surfaces. Phospholipid vesicles were synthesized using the existing protocol[70]. Specifically, 98% of DOPC (1,2-dioleoyl-sn-glycero-3-phosphocholine) was mixed with 2% of biotin-DOGC (1,2-dioleoylsn-glycero-3-[(N-(5-amino-1-carboxy-ypentyl)iminodiacetic acid)-succinyl]) biotin lipids. The lipid solution was further diluted with tris buffered saline (Sigma-Aldrich) at a ratio of 1:1 before

introduction. The glass coverslips were assembled into the donut shape chamber at the UHQ water reservoir. The hybrid substrate was then incubated with 0.1% BSA (Sigma Aldrich) for 2 h to reduce nonspecific protein absorption. Next, 1 µg/ml of Dylight-405 NeutrAvidin (Thermo Fisher) was introduced to the chamber for 1 h, and then incubated with 1 µg/ml of RGD-biotin (Peptides International) for 1 h. Multiple rinses with UHQ were applied after each step. A fluidity test of RGD molecules at the surface of supported lipid bilayer (SLB) was performed by fluorescence recovery after photobleaching. The concentration of RGD on both SLB and PLL-g-PEG polymer was similar, based on the estimated fluorescence of Dylight-405 NeutrAvidin (Supplementary Fig. 6).

HeLa-JW cells expressing a GFP-myosin X were introduced to the chamber, allowing cells to adhere to and spread on the RGD-coated hybrid substrate. Cells were visualized using a charge coupled EMCCD camera (Photometrics) coupled with total internal reflection microscopy with 100× objective (1.5 NA, Nikon). The chamber was maintained at 37 °C during observation. The time-lapse video was recorded at varying intervals over a range of 3–5 s.

The GFP-Myosin X cluster at the filopodia tip was tracked by using a cross-correlation single-particle tracking method[71]. Obtained trajectories were analyzed using the inpolygon matlab function to find trajectory segments that crossed SLB circular islands and computer drawn islands in the center of the SLB pattern. The time intervals between the initial and end points of the segments were then used to estimate the average filopodia tip dwell time on the SLB and computer drawn circular islands (Fig. 8). In this analysis, we used only trajectories that spanned more than five frames.

**Graphs and statistical analysis**. Prism (GraphPad 6.0 Software) was used for graphing and statistical analysis unless it specified differently elsewhere.

**Reporting summary**. Further information on research design is available in the Nature Research Reporting Summary linked to this article.

## Data availibility
The authors declare that all data supporting the findings of this study are available within the article and its Supplementary Information or from the authors upon reasonable request.

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

## Acknowledgements

Encouraging and productive discussions with Drs. D. Bray (University of Cambridge, UK), M.M. Kozlov (Tel Aviv University, Israel) and E.J. Manser (IMCB, A*STAR, Singapore) are much appreciated. We are grateful to Drs. T. Kachanawong (MBI, Singapore), R. Cheney (University of North Carolina Medical School, USA), M.J. Schell (Uniformed Services University, Bethesda, Maryland, USA), W. Bement (University of Wisconsin, Madison, USA), V. Verkhusha (Albert Einstein College of Medicine, New York, NY, USA), W. Wolf and R. Chisholm (Northwestern University, Chicago, IL, USA), R. Adelstein, (NHLBI, NIH, USA), Vicente-Manzanares (Universidad Autónoma de Madrid, Madrin, Spain), S. Narumiya (Faculty of Medicine, Kyoto University, Japan) and J. Brugge (Harvard University, Boston, MA, USA) for providing genetic constructs, Dr. D. Kovar (University of Chicago, IL, USA) for a sample of SMIFH2 inhibitor. We thank the Protein Cloning and Expression Core facility of the MBI for help with sub-cloning of mCherry-mDia2, mApple-myosin X and mDia2 ΔDAD-GFP. We also thank Dr. F. Margadant and Lau Wai Han (MBI Microscopy Core facility) and Dr. V. Vyasnoff (MBI, Singapore) for their kind help with the optical tweezers setup. We thank Drs. S. Wolf and A. Wang (Science Communication MBI, Singapore) for excellent editorial help. This research is supported by the National Research Foundation, Prime Minister's Office, Singapore and the Ministry of Education under the Research Centres of Excellence program through the Mechanobiology Institute, Singapore (Ref no. R-714-006-006-271) (A.D.B.) and Singapore Ministry of Education Academic Research Fund Tier 3 MOE Grant No. MOE2016-T3-1-002 (A.D.B.). A.D.B. also acknowledges support from Maimonides Israeli-France grant (Israeli Ministry of Science Technology and Space) and EU Marie Skłodowska-Curie Network InCeM (Project ID: 642866) at the Weizmann Institute of Science. Y.J. acknowledge support from grant MOE2012-T3-1-001.

## Author contributions

A.D.B. conceived the research. N.O.A. and A.K.E. performed majority of experiments and analyzed the data, N.O.A., A.K.E. and A.D.B. wrote the paper. S.H. performed the early experiment on N-SIM system. D.O. and Z.C. performed the experiments on SLB with subsequent data analysis, M.N. provided the technical assistance, H.T.O. provided the expertize in image analysis, A.J. and G.R.L. conducted the in vitro experiments and provided the subsequent data analysis, J.T.G., M.P.S, J.Y. and A.D.B. supervised the research.

## Additional information

**Competing interests:** The authors declare no competing interests.

