## [Peer Review File · Nature Communications]

Reviewers' comments:

Reviewer #1 (Remarks to the Author):

This study investigates roles of formins and myosin II isoforms for adhesions at filopodial tips. Authors show that myosin enhances adhesion strength and that formins appear to attach actin filaments to the membrane at the filopodial tip. Although both findings are fairly predictable, this work provides experimental validation to these predictions, which is important. However, there are some serious concerns that weaken the significance of this study.

First, the experimental system is rather artificial. Authors investigate exclusively myosin X-induced filopodia. Although ectopic expression helps to make filopodia visible, it is unreasonable to use only one marker. VASP, mDia, talin or even a plasma membrane reporter could be used as a complementary approach to support the conclusions.

Second, in most experiments filopodia are affected by external force instead of expressing their normal behavior. External force can enhance adhesion by itself regardless of (or together with) myosin, which complicates the interpretation of experiments. Although some experiments here are done on free filopodia, they are not particularly conclusive in answering the main question.

Finally, the presented evidence is often unconvincing with many disparities between the data and the text. The examples are given below. Some of them might be minor, but their sheer abundance greatly undermines quality of the work.

Specific comments:

(1) p. 4: The statement that the actin core in filopodia remains intact (Fig. 1B and 3A) is poorly supported by the data. The kymograph in Figure 1C shows the opposite – the red signal is interrupted by gaps within the filopodium.

(2) p. 4: "When the beads were coated with concanavalin A instead of fibronectin, application of force usually resulted in the formation of membrane tethers, rather than growth of the filopodia (movie S3)." Movie S3 quickly goes out of focus making it unclear whether the red color (is it actin probe?) is present or not at the bead. Since actin is poorly visible in fibronectin beads, there is no much difference between the two conditions.

(3) p. 4: The statement in the text ("In a typical experiment, we detected ~ 5 such peaks") conflicts with the statement in figure 1 legend ("21 peaks from 8 beads"), which suggests 2.6 peaks per case.

(4) p. 4: "...the retrograde movement of myosin X patches colocalized with the patches of VASP (fig. S1B middle panel & fig. S6)" and a similar statement on p. 7. These figures do not show any movement, only colocalization.

(5) p. 4: Movie S4 is uninformative, because nothing can be seen in the actin panel.

(6) p. 5: The text states that in COS7 cells "numerous bipolar myosin IIA filaments frequently localized to the bases of filopodia labeled by myosin X (Fig. 2B right, 2C right, movie S5C). Almost every filopodium in such cells displayed one or few myosin IIA filaments at its base during the period of observation." Figure 2B (right) shows one, or at most two, myosin filaments that could be considered to localize to filopodial bases, whereas most of myosin IIA filaments are far away from all other filopodia. Quantification is needed to support the above statement, especially about "almost every filopodium...".

(7) p. 5: "by decreasing the rate of filopodia withdrawal (Fig. 2G)." Figure 2G shows not the rate of filopodia withdrawal but the filopodial lifetime in a different form than in figure 2F.

(8) p. 6: "While treatment with either S-nitroblebbistatin or Y27632 did not change the average filopodia length, myosin IIA knockdown resulted in moderate but significant shortening of filopodia, (Fig. 2D)". The inconsistency between the myosin IIA knockdown and effects of blebbistatin and Y-27632 on filopodial lengths, as well as a really minor effect of IIA knockdown, are a concern. In any case, what is important about filopodia length changes? It is hard to interpret them. The shortening of filopodia after IIA knockdown, if it is true, is counter intuitive, because myosin-driven retrograde flow should shorten filopodia, so filopodia should lengthen after myosin inhibition. On the other hand,

lamellipodia become more active after inhibition of myosin II and would engulf filopodia indirectly making them apparently shorter. However, in such case, Y-27632 and blebbistatin should have the same effect, as myosin knockdown.

(9) p. 6: "While in control HeLa-JW cells application of pulling force via fibronectin-coated bead induced sustained growth of attached filopodia accompanied by the development of up to ~ 5 pN force, in the cells with impaired myosin II activity the filopodia detached earlier, after developing rather small forces (Fig. 3B-C, E, movies S6 B-C)." There are several concerns with this single sentence.

- It is not clear how quantification of forces was performed. There is no relevant information in Methods. Do authors take into consideration only one maximal peak per bead? This makes it difficult to compare with the average (~3 pN) values shown in fig. 1C.

- If this is how the quantification was done, then the above statement conflicts with Fig. S4, which shows a max peak of ~10 pN for the Y-27632-treated cell, which is greater, not smaller than for control cells.

- Figure S4 also conflicts with figure 3E, which shows a bunch of zeros and one 5 pN value for Y27632, while the ~10 pN data point from fig. S4 is not there.

- Timing of detachment should be quantified for all conditions used in the study. Exact criteria of how the moment of detachment is determined should be given.

(10) In movies 6b and 6c, the filopodia appear attached to the bead for almost entire movie, but the figure legend says that filopodia detach somewhere in the middle of the sequence. At the end of the movie, the bead disappears. Why, if the bead is in a trap?

(11) p. 6: "We also examined the immediate effect of Y27632 during the force-induced sustained growth of filopodia. After the drug was added, filopodia detached from the bead (fig. S4, movie S7)." After addition of the drug, the filopodium becomes invisible. Does it lack actin or become bleached?

(12) p. 7: "in cells treated with SMIFH2... practically none of these filopodia had myosin X comet tails at their tips (Fig. 3D and fig. S5)" This statement conflicts with data in Fig. 4 (and corresponding movie 8) and fig. S6, which show many more filopodia with myosin X than without. This result should be supported by quantification.

(13) p. 7: "incubation of SMIFH2 treated cells with Y27632 efficiently stopped the retrograde movement of the myosin X positive patches". Needs quantification.

(14) p. 7: "Following treatment with 100 μ M SMIFH2, a rapid decrease in the fraction of filaments remaining associated with immobilized formin under conditions of mild shear flow ("survival fraction") was observed (fig. S7)." Figure S7 shows only a graph and no other evidence. Raw data should be shown too.

(15) p. 8: The experiment in Fig. 5 might answer the question of how exactly myosin inhibition affects the filopodial length (shown in Fig. 2D), if analyzed differently. The current quantification does not distinguish between retraction and protrusion, polymerization and retrograde flow. It also does not take into account behavior of the adjacent cell edge. Therefore, the provided numbers do not tell much about the mechanism of induced filopodia pausing. Even though this experiment is done on filopodia not affected by external force, it is not informative about roles of contraction in filopodial adhesion.

(16) p. 9: "we interpret the acceleration of myosin X/VASP movement upon SMIFH2 treatment as an evidence of actin filament detachment from formins at the filopodia tips." Strictly speaking, it can be due to acceleration of actin polymerization at the tip. Is detachment seen directly by the actin probe?

(17) p. 10: "The myosin II-driven force is transmitted to the filopodium tip via actin core associated with formin molecules at the tip." Correlation of the presence of myosin filaments at the filopodium base with the force pulses would be a nice supporting evidence for the model.

(18) p. 11: "Thus, force developed during growth cessation may trigger the subsequent filopodia growth". It is not clear on what data this conclusion is based. Since much of the subsequent discussion is built on this idea, better justification for it is needed.

Minor comments:

1. Abstract: "Strikingly, pharmacological inhibition or knockdown of myosin IIA, which localized to the base of filopodia, resulted in weakening of filopodia adherence strength." In fact, this is a totally predictable result. An opposite one would be striking.
2. p. 3: I do not recommend using the term "dynamic instability" for filopodia dynamics, as it implies that during retraction filopodia depolymerize from the tip, which is hardly the case.
3. p. 3: Colocalization of mDia2, VASP, and talin at the tips of myosin X-induced filopodia is shown only by co-expression. Are endogenous proteins also localize there?
4. The order of supplemental figures is scrambled.
5. Since different microscopy modes are used in the study, figure legends should indicate them (SIM or confocal or else).
6. p. 5: The reference to "Fig. 2E right" is irrelevant, because there is only one 2E panel.
7. Figure S3: Why does actin have a punctate appearance?
8. In movie 6a, the filopodium kinks. How does it do so, if the cell is pulled away?
9. p. 7: "...their mean length decreased only slightly (Figs. 3D and 4A, fig. S5)" No length data are shown in fig. 3D.

Reviewer #1 (Remarks to the Author):

This study investigates roles of formins and myosin II isoforms for adhesions at filopodial tips. Authors show that myosin enhances adhesion strength and that formins appear to attach actin filaments to the membrane at the filopodial tip. Although both findings are fairly predictable, this work provides experimental validation to these predictions, which is important.

We respectfully disagree that the results of our study are “fairly predictable”. The sense of predictability of our results may emerge from the perception that the filopodia tips and the focal adhesions are analogous. Indeed, years ago we have demonstrated the mechanosensitivity of focal adhesions (Riveline et al., 2001), and these results have been confirmed and extended by many researchers since then. However, it should be noted that cell filopodia are not identical to either nascent or mature focal adhesions. While similarly to focal adhesions, filopodia tips are enriched in integrin and talin proteins, vinculin has never been observed to localize specifically to the filopodia tips either by us or by other research groups (Hu et al., 2014). Therefore, the mechanism of mechanosensitivity based on force-dependent interactions between vinculin and talin that is often referenced in the literature (Gauthier and Roca-Cusachs, 2018; Yan et al., 2015) cannot be applied to filopodia adhesion. On the other hand, it is well-known that filopodia tips contain myosin X proteins, which are most probably absent in focal adhesions. Thus, the protein composition between focal adhesion and filopodia is not analogous. Furthermore, our utilization of the optical tweezers experimental setup permitted us to directly assess the strength of filopodia adhesion to extracellular matrix, via measurements of the filopodia detachment force from microbeads coated with fibronectin. As such measurements have never been published previously for filopodia or for focal adhesion complexes, we believe that the experimental results reported in our study will be of great interest for researchers working in the mechanobiology field. Importantly, filopodia are perhaps the most general type of cell adhesion protrusions used by living cells, which are much more ubiquitous than focal adhesions. In particular, filopodia participate in cell migration through the extracellular 3D environment, while focal adhesions are formed mainly by cells attached to the rigid planar substrate. Therefore, our discovery of filopodia adhesion mechanosensitivity and elucidation of the role of myosin IIA and formins in this process is a novel and important step in understanding of the mechanisms of cell directional migration.

However, there are some serious concerns that weaken the significance of this study.

First, the experimental system is rather artificial. Authors investigate exclusively myosin X-induced filopodia. Although ectopic expression helps to make filopodia visible, it is unreasonable to use only one marker. VASP, mDia, talin or even a plasma membrane reporter could be used as a

complementary approach to support the conclusions.

For the experiments with HeLa and Cos-7 cells, which normally do not produce many filopodia, we used fluorescently-tagged myosin X not only for marking the filopodia tips but also as a powerful stimulator of filopodia formation. Since myosin X is a ubiquitous component of the majority of filopodia types and naturally overexpressed in many cancer cells such an experimental system is thought to represent the behavior of sufficiently broad class of filopodia and has been used in many recent studies (Jacquemet et al., 2016). We also investigated the force driven elongation of filopodia induced by other means, namely by constitutively active mDia2 and Cdc42 or overexpressed VASP. In all these cases the behavior of filopodia was more complex than that of filopodia induced by overexpression of myosin X and its detailed investigation is beyond the scope of the present study.

Second, in most experiments filopodia are affected by external force instead of expressing their normal behavior. External force can enhance adhesion by itself regardless of (or together with) myosin, which complicates the interpretation of experiments. Although some experiments here are done on free filopodia, they are not particularly conclusive in answering the main question.

It was the aim of our study to investigate the mechanosensitivity of filopodia adhesions. This aim cannot be approached without experiments in which a mechanical pulling force is applied to filopodia-matrix adhesions. The fact that filopodia can by itself produce a pulling force has already been documented in previous studies (Bornschiogl et al., 2013). This means that adhesions between filopodia tips and extracellular matrix systematically experience such forces. Moreover, there is ample evidence (Watanabe et al., 2010) that the growth of adherent filopodia consists of series of episodes whereby pulling force is applied to filopodia tips. To study how such forces affect adhesion, we combined experiments where controlled external force was applied to filopodia tips together with the examination of the growth of unconstrained adherent filopodia. Combining these two types of experiments is necessary for understanding the adhesion mechanosensitivity.

Finally, the presented evidence is often unconvincing with many disparities between the data and the text. The examples are given below. Some of them might be minor, but their sheer abundance greatly undermines quality of the work.

We are grateful to this reviewer for their careful editing and have attempted to answer all these examples.

Specific comments:

(1) p. 4: The statement that the actin core in filopodia remains intact (Fig. 1B and 3A) is poorly supported by the data. The kymograph in Figure 1C shows the opposite – the red signal is interrupted by gaps within the filopodium.

In our movies the “gaps” in actin fluorescence labeling is a result of reduced brightness (partially because of photobleaching). It does not mean that the continuity of actin filaments is interrupted. In the revised version, we add an image of filopodia fixed after the experiment with pulling, in which the integrity of actin core is clearly seen (movie S2, new numeration). We have also changed the color balance in the kymograph in Fig. 1C to make the actin signal more prominent.

(2) p. 4: “When the beads were coated with concanavalin A instead of fibronectin, application of force usually resulted in the formation of membrane tethers, rather than growth of the filopodia (movie S3).” Movie S3 quickly goes out of focus making it unclear whether the red color (is it actin probe?) is present or not at the bead. Since actin is poorly visible in fibronectin beads, there is no much difference between the two conditions.

We have now included another movie S4 (new numeration), which shows the process of tether formation more clearly. We have also changed the description of the results of experiments with Con A coated beads emphasizing that pulling with such beads never induced the growth of filopodia actin cores.

(3) p. 4: The statement in the text (“In a typical experiment, we detected ~ 5 such peaks”) conflicts with the statement in figure 1 legend (“21 peaks from 8 beads”), which suggests 2.6 peaks per case.

We are grateful to the reviewer for attracting our attention to this discrepancy. We have now checked the numbers of peaks registered in all our experiments and found that this number varied from 2-4 peaks per experiment. We have changed the text accordingly. In particular, after inspection of the movies mentioned in the Fig. 1 legend we found that the 21 peaks were observed in 6 independent experiments with beads.

(4) p. 4: “...the retrograde movement of myosin X patches colocalized with the patches of VASP (fig. S1B middle panel & fig. S6)” and a similar statement on p. 7. These figures do not show any movement, only colocalization.

We have now included a supplementary movie S6 (new numeration) in which retrograde movement of colocalized VASP and myosin X is shown.

(5) p. 4: Movie S4 is uninformative, because nothing can be seen in the actin panel.

Experiments with local photoactivation of actin in filopodia are extremely difficult due to small amount of labeled actin, which results in a weak signal. We, however, succeeded in visualization of this weak signal showing retrograde actin movement. In the present version we highlighted the actin signal to make it easily noticeable. Retrograde movement of actin shown in movie S4 (movie S7 in new numeration) is clearly seen also in corresponding

kymograph on fig. S1C.

(6) p. 5: The text states that in COS7 cells “numerous bipolar myosin IIA filaments frequently localized to the bases of filopodia labeled by myosin X (Fig. 2B right, 2C right, movie S5C). Almost every filopodium in such cells displayed one or few myosin IIA filaments at its base during the period of observation.” Figure 2B (right) shows one, or at most two, myosin filaments that could be considered to localize to filopodial bases, whereas most of myosin IIA filaments are far away from all other filopodia. Quantification is needed to support the above statement, especially about “almost every filopodium...”.

The Fig. 2B shows one frame from the movie S5C (movie S8C in new numeration). In this frame some filopodia indeed do not contain filaments of myosin IIA, however, this localization can be seen on other frames in the same movie. We have now analyzed the histories of > 20 filopodia from this movie and found that ~ 70% of the monitored filopodia had one or more myosin IIA doublets at their bases during the movie duration (see revised version of results, page 7).

(7) p. 5: “by decreasing the rate of filopodia withdrawal (Fig. 2G).” Figure 2G shows not the rate of filopodia withdrawal but the filopodial lifetime in a different form than in figure 2F.

Thank you for this correction, we have changed the text accordingly.

(8) p. 6: “While treatment with either S-nitroblebbistatin or Y27632 did not change the average filopodia length, myosin IIA knockdown resulted in moderate but significant shortening of filopodia, (Fig. 2D)”. The inconsistency between the myosin IIA knockdown and effects of blebbistatin and Y-27632 on filopodial lengths, as well as a really minor effect of IIA knockdown, are a concern. In any case, what is important about filopodia length changes? It is hard to interpret them. The shortening of filopodia after IIA knockdown, if it is true, is counter intuitive, because myosin-driven retrograde flow should shorten filopodia, so filopodia should lengthen after myosin inhibition. On the other hand, lamellipodia become more active after inhibition of myosin II and would engulf filopodia indirectly making them apparently shorter. However, in such case, Y-27632 and blebbistatin should have the same effect, as myosin knockdown.

We agree with this reviewer that filopodia length is not the best parameter for the characterization of the filopodia growth. It is rather the filopodia pulling experiments and the measurements of filopodia lifetime, which revealed the role of myosin IIA in filopodia adhesion and growth. The measurements of filopodia length in our study simply show that experimental treatments presented in (fig. S2, new) did not result in drastic changes in filopodia length. Therefore, in the revised version of the paper we put the data on filopodia length in the supplementary material section. Nevertheless, it is worth noting

that the simple idea mentioned by the reviewer that inhibition of myosin dependent retrograde flow should promote filopodia growth (first proposed by (Lin et al., 1996) is not apparently consistent with our data. We never observed any enhancement of filopodia growth upon myosin II inhibition. On the contrary, in our case, the tension developed by myosin IIA seems to be required for filopodia growth (Fig. 5).

(9) p. 6: “While in control HeLa-JW cells application of pulling force via fibronectin-coated bead induced sustained growth of attached filopodia accompanied by the development of up to ~ 5 pN force, in the cells with impaired myosin II activity the filopodia detached earlier, after developing rather small forces (Fig. 3B-C, E, movies S6 B-C).” There are several concerns with this single sentence.

- It is not clear how quantification of forces was performed. There is no relevant information in Methods.

We have now extended the description of the procedure of force measurements in Methods. The forces were calculated on the basis of bead displacements from the optical trap center using the trap stiffness value determined in independent experiments.

Do authors take into consideration only one maximal peak per bead? This make it difficult to compare with the average (~ 3 pN) values shown in fig. 1C.

Depending on the experimental question, we either presented the complete time course of force changes (as in the main graph in Fig. 1C or supplementary fig. S4), or plotted the distribution of force peak values (as in the inset to Fig. 1C), or, finally, showed the maximal peak value over the entire duration of the experiment (Fig. 3E).

- If this is how the quantification was done, then the above statement conflicts with Fig. S4, which shows a max peak of ~ 10 pN for the Y-27632-treated cell, which is greater, not smaller than for control cells.

The experiment shown in fig. S4 represent the forces applied to the bead by filopodium slightly before and immediately after addition of Y27632 inhibitor. It therefore differs from the experiments summarized in Fig. 3E, in which the cells were pretreated by different inhibitors for more than 10 minutes. The development of a high force of 10pN detected in this experiment may reflect a transient cell reaction to the drug application. For us it is more important that after addition of the inhibitor, the force shortly dropped to zero suggesting the detachment of the filopodium from the bead (which was confirmed by the bead slipping away after release from the trap).

- Figure S4 also conflicts with figure 3E, which shows a bunch of zeros and one 5 pN value for Y27632, while the ~ 10 pN data point from fig. S4 is not there.

We did not include the results of experiment presented in fig. S4 into the score shown in Fig. 3E. As mentioned in the previous answer, in experiment 3E only cells pretreated by inhibitors were used while experiment S4 shows the immediate response of the cell to addition of Y27632 inhibitor.

- Timing of detachment should be quantified for all conditions used in the study. Exact criteria of how the moment of detachment is determined should be given.

The moment of detachment of filopodia from the bead was detected by return of the bead to the center of optical trap (see fig. S4A and B, for example). To confirm the detachment, we then switched the trap off, which resulted in release of the bead and its disappearance from the field of view unless it remained connected to filopodia via the membrane tethers (movie S9B-D, S10 and S13 in new numeration). The timing of these events is indicated in Fig. 3 and fig. S4.

(10) In movies 6b and 6c, the filopodia appear attached to the bead for almost entire movie, but the figure legend says that filopodia detach somewhere in the middle of the sequence. At the end of the movie, the bead disappears. Why, if the bead is in a trap?

We apologize for some confusion. The moment of filopodia detachment from the bead was detected by abrupt change of the bead position corresponding to its return to the center of the trap. This moment was determined using Metamorph tracking software, which permits registration of the bead position with 5 nm precision. In some cases this moment can be easily determined by visual inspection of the movie. For example in the movie 6b (movie 9B in new numeration) the image of the bead abruptly changed at 6 min 44 s, which corresponded to the moment of detachment, as correctly indicated in the legend to this movie. After detachment, we continued filming for six more minutes and then switched off the trap at 12 min 40 s releasing the bead to demonstrate that it is indeed not attached to filopodia anymore (the bead “disappeared” only when the trap was off). In Fig. 3B this moment was indicated. The same is true for the movie S6C (movie S9C in new numeration), where exact moment of detachment is correctly indicated in the legend to the movie and the moment of the trap switching off is indicated in Fig. 3C. To avoid confusion we removed misleading yellow arrows from Fig. 3 and indicated in the legend the real times of bead detachment (similar to those indicated in the legend to movies 6b, c and d). We described the method of detection of bead detachment in the Methods.

(11) p. 6: “We also examined the immediate effect of Y27632 during the force-induced sustained growth of filopodia. After the drug was added, filopodia detached from the bead (fig. S4, movie S7).” After addition of the drug, the filopodium becomes invisible. Does it lack actin or become bleached?

In this movie (movie S10 in new numeration) the intensity of actin labeling

was relatively low and the apparent “disappearance” of actin was a result of photobleaching. We have now indicated this in the legend.

(12) p. 7: “in cells treated with SMIFH2... practically none of these filopodia had myosin X comet tails at their tips (Fig. 3D and fig. S5)” This statement conflicts with data in Fig. 4 (and corresponding movie 8) and fig. S6, which show many more filopodia with myosin X than without. This result should be supported by quantification.

There is no contradiction between data presented in Fig. 3D/S5 and Fig. 4/movie S8 (movie S11 in new numeration). In the Fig. 3D and S5 the cells were treated with SMIFH2 for a long time (1 and 2 hours, respectively). After such treatment the myosin X indeed appeared to be removed from the filopodia tip. As the reviewer requested, we have now quantified the percentage of myosin X containing filopodia after long-term incubation with SMIFH2 and indicated it in the text (page 10). 25% (2 experiments, 14 cells). In Fig. 4 and movie S8 (movie S11 in new numeration) we investigated the process of disintegration of the myosin X comet tail upon inhibition of formins. In such experiments we began observations 15 min following drug addition when the majority of filopodia still contained myosin X.

(13) p. 7: “incubation of SMIFH2 treated cells with Y27632 efficiently stopped the retrograde movement of the myosin X positive patches”. Needs quantification.

We have measured the velocities of myosin X patches in cells treated with SMIFH2 and Y27632 and included these data into the modified version of the paper.

(14) p. 7: “Following treatment with 100 μ M SMIFH2, a rapid decrease in the fraction of filaments remaining associated with immobilized formin under conditions of mild shear flow (“survival fraction”) was observed (fig. S7).” Figure S7 shows only a graph and no other evidence. Raw data should be shown too.

We have included in the revised version the frames corresponding to the beginning (time zero) and 300 sec time of the movies showing formin dependent actin filament growth with and without addition of the formin inhibitor SMIFH2 (fig. S7A). The primary data of this type were used for computation of the values of filaments “survival fraction” shown in the graph S7B.

(15) p. 8: The experiment in Fig. 5 might answer the question of how exactly myosin inhibition affects the filopodial length (shown in Fig. 2D), if analyzed differently. The current quantification does not distinguish between retraction and protrusion, polymerization and retrograde flow. It also does not take into account behavior of the adjacent cell edge. Therefore, the provided numbers do not tell much about the mechanism of induced filopodia pausing. Even

though this experiment is done on filopodia not affected by external force, it is not informative about roles of contraction in filopodial adhesion.

In our answer to comment #8, we agreed that filopodia length is a secondary parameter, which is not crucially important for elucidation of the roles of myosin IIA and formins in filopodia growth. Therefore, in the revised version we used the filopodia lifetime as a more meaningful parameter, which obviously depends on myosin IIA activity (Fig. 2I and J). In addition, it appears that inhibition of either myosin IIA or formin(s) almost entirely blocks filopodia growth, as shown in Fig. 5. These findings provide us with basic information about filopodia dynamics and permit us to hypothesize that growth of filopodia depend on a signal generated by appropriate adhesion of their tips to the extracellular matrix. More detailed description of the dynamics of the unconstrained filopodia will require simultaneous visualization of filopodia adhesion to the substrate and traction forces developed by filopodia, which will require substantial modification of the present experimental setup. We are planning to implement this program in our future studies.

(16) p. 9: “we interpret the acceleration of myosin X/VASP movement upon SMIFH2 treatment as an evidence of actin filament detachment from formins at the filopodia tips.” Strictly speaking, it can be due to acceleration of actin polymerization at the tip. Is detachment seen directly by the actin probe?

The detachment of actin filaments from formin upon treatment with SMIFH2 was clearly demonstrated in *in vitro* experiments (fig. S7). We believe that *in vivo* the inhibitor produces similar effect. To the best of our knowledge there are no data in the literature, which describe even a transient increase of actin polymerization in cells treated with SMIFH2.

(17) p. 10: “The myosin II-driven force is transmitted to the filopodium tip via actin core associated with formin molecules at the tip.” Correlation of the presence of myosin filaments at the filopodium base with the force pulses would be a nice supporting evidence for the model.

We believe that our experiments with knockdown of myosin IIA and inhibition of myosin II by blebbistatin and Y-27632 convincingly demonstrated the role of myosin II filaments in generation of forces applied to adherent filopodia tips. We agree with this reviewer that simultaneous visualization of myosin IIA filaments and measurement of traction forces could make this statement even more convincing. This however will require the significant improvement of our experimental setup, which we believe is beyond the scope of this study.

(18) p. 11: “Thus, force developed during growth cessation may trigger the subsequent filopodia growth”. It is not clear on what data this conclusion is based. Since much of the subsequent discussion is built on this idea, better justification for it is needed.

This idea emerged from previous experiments by (Watanabe et al., 2010),

who demonstrated the cyclic growth of filopodia attached to the substrate, and our own results showing cyclic development of pulling forces in the growing filopodia. Notably the periods of force development preceded the periods of filopodia elongation. In the new version of the discussion we have described this mechanism in more detail.

Minor comments:

1. Abstract: “Strikingly, pharmacological inhibition or knockdown of myosin IIA, which localized to the base of filopodia, resulted in weakening of filopodia adherence strength.” In fact, this is a totally predictable result. An opposite one would be striking.

This result is indeed predictable if you assume that filopodia behave similarly to the focal adhesions which demonstrate force dependent growth (Riveline et al., 2001). This idea was indeed motivation for our study of filopodia adhesion mechanosensitivity. However, as we extensively discussed above in the answer to the first statement of this reviewer, filopodia and focal adhesions cannot be regarded as completely analogous, and therefore this idea cannot be taken for granted. Moreover, not all types of filopodia demonstrated force dependent growth. In particular, unlike myosin X-induced filopodia, the filopodia induced by constitutively active Cdc42 did not grow upon force application. Investigation of the variability in the force response between different filopodia types is beyond the scope of present study. However this example clearly shows that the force dependence of filopodia adhesion is not the obvious phenomenon. Finally, since filopodia mediated adhesion is very important in many types of cellular activity (see above), the investigation of its force dependence is important irrespective of their perceived similarity to focal adhesions. Above all, we believe that such studies are important since they permit to understand the directionality of cell migration.

2. p. 3: I do not recommend using the term “dynamic instability” for filopodia dynamics, as it implies that during retraction filopodia depolymerize from the tip, which is hardly the case.

We agree and have changed the text accordingly.

3. p. 3: Colocalization of mDia2, VASP, and talin at the tips of myosin X-induced filopodia is shown only by co-expression. Are endogenous proteins also localize there?

Co-localization of endogenous mDia2, VASP, talin and myosin X in filopodia tip was documented in a series of previous studies done by different groups. We have included this information in our paper: mDia2 (Schirenbeck et al., 2005), VASP (Bear and Gertler, 2009), talin (Sydor et al., 1996), myosin X (Zhang et al., 2004).

4. The order of supplemental figures is scrambled.

We corrected this.

5. Since different microscopy modes are used in the study, figure legends should indicate them (SIM or confocal or else).

We checked the legends and corrected them.

6. p. 5: The reference to “Fig. 2E right” is irrelevant, because there is only one 2E panel.

This figure is now changed.

7. Figure S3: Why does actin have a punctate appearance?

The myosin IIA knockdown cell expressing Ftractin shown in fig. S3B (lower right panel) was fixed after pulling experiment. The actin cytoskeleton in cells lacking myosin IIA undergo strong reorganization, and formation of transient actin patches cannot be excluded. We cannot also entirely exclude that the Ftractin-positive patches seen in this figure are just fixation artifacts. We show this image only to demonstrate that this cell, which we used for pulling experiments, did not contain myosin IIA. The actin labeling was used here just for cell identification.

8. In movie 6a, the filopodium kinks. How does it do so, if the cell is pulled away?

Filopodia buckling is a well known phenomenon (Leijnse et al., 2015). The pulling forces we apply may not be sufficient to straighten such buckled filopodia. In addition, in movie 6A (movie S9A in new numeration) formation of kink could be related to formation of a transient adhesion between filopodia shaft and the substrate.

9. p. 7: “...their mean length decreased only slightly (Figs. 3D and 4A, fig. S5)”
No length data are shown in fig. 3D.

In this sentence we referred not only to the quantitative data (Figure 4A), but also to representative images of SMIFH2 treated filopodia (Figure 3D and S5). We have removed now the references to these images.

Reviewer #2 (Remarks to the Author):

The filopodia of motile cells are involved in sensing the chemical environment but also might have mechanosensing properties. Alieva and colleagues studied the mechanoresponse of filopodia and studied the molecular mechanism underlying mechanosensing.

To address this topic, the authors employed an optical-tweezer setup to analyze filopodia that were induced by myosin-X overexpression. By comparison of Cos7 cells (no endogenous Myo IIA) to Cos7 cells with

transient MyoIIA expression and by comparing HeLa cells with knockdown of MyoIIA the authors show a dependence of filopodia length and force on MyoIIA. Chemical inhibition of contractility, in contrast, only influences force but not filopodia length. In additional experiments, the authors demonstrate that inhibition of formins (located at the tip of filopodia) resembles the effects of reduced MyoIIA expression. In addition, formin inhibition prevents MyoX localization at the filopodia tip by an unknown mechanism. Finally, the authors analyze filopodia that are not attached to a bead in an optical tweezer. Under these conditions, MyoIIA knockdown, formin inhibition, and contractility inhibition all interfere with the growth rate of filopodia.

Based on these results, Alieva and colleagues propose a mechanosensitive mechanism where filopodia growth is caused by force-transmission from MyoIIA at the filopodia base to formins at the filopodia tip. This interpretation, is supported by publications showing that actin polymerization from isolated formins is increased under mechanical load.

Unfortunately, the main findings in this manuscript are not fully consistent with this interpretation:

- One major concern is that filopodia length is not influenced by contractility inhibitors (blebbistatin or Y27632, Fig. 2D). It was shown by other groups that mechanosensing and mechanoreponse of focal adhesions also rely on actin integrity (work from the Gardel group) and that the crosslinking function of MyoIIA might be as relevant as its motor function in this context (Choi et al., 2008, NCB). This could be an alternative explanation given the limited effect of the chemical inhibitors.

We are very grateful to this reviewer for raising this important question. In the paper by Choi et al the role of the crosslinking function of myosin IIA in the maturation of focal adhesions was suggested based on experiments in which a mutant of myosin IIA with compromised motor function (myosin IIAN93K) was still able to rescue the focal adhesion maturation in myosin IIA knockout cells. Following the same logic, we have now studied whether myosin IIAN93K (kindly provided to us by Dr. Vicente-Manzanares) could increase the filopodia lifetime in Cos-7 cells (lacking myosin IIA) similarly to the wild type myosin IIA. This experiment (Fig. 2H, I and J, new), however, clearly showed that myosin IIAN93K only slightly increased the lifetime of filopodia in Cos-7 cells as compared to wild type myosin IIA. This new result strongly suggests that the motor function of myosin IIA, not crosslinking, is critically important for stabilization of filopodia adhesions.

Concerning the results of filopodia length measurements mentioned by this reviewer we now agree (following the discussion with reviewer #1) that filopodia length is not the best parameter for studying the myosin II dependent regulation of filopodia adhesions. The effect of myosin II inhibition on filopodia adhesion is better manifested by the decrease of the filopodia detachment force from fibronectin-coated beads (Fig. 3E) and by decrease of filopodia lifetime (Fig. 2I and J, new). In this sense the mechanosensitivity of filopodia adhesion differs from that of the focal adhesion complexes, which responds to force by increasing in size.

- The localization of MyoIIA should be of major importance based on the author's interpretation. In "Fig. 2B right" it appears that MyoIIA is not located at the base of filopodia at all. This is in contrast to "Fig. 2C right" although both pictures show the same condition. If MyoIIA localization had a direct effect on filopodia length, I would expect a drastic effect of a prominent localization within a filopodium as seen in "Fig 2C right". However, the effect is rather modest (Fig. 2D).

This point was also brought up by reviewer #1, question 6. As we answered previously, the localization of myosin filaments at the base of filopodia are transient and not every filopodia at any given moment has the myosin filament at the base. In particular, the original Fig. 2B showed one frame from the movie S5C (movie S8C in new numeration). In this frame some filopodia indeed did not contain filaments of myosin IIA, however these can be seen on the other frames of the same movie. We have now analyzed the histories of > 20 filopodia from this movie and found that ~ 70% of the monitored filopodia had one or more myosin IIA doublets at their bases during the movie duration (see revised version of the results, page 7). Concerning the "modest" effect of the presence of myosin filament, the reviewer refers to our measurement of filopodia length (original Fig. 2D), which, as we mentioned above, is not the best parameter for characterizing filopodia adhesion and growth. The difference between filopodia that contained or lacked myosin IIA was better revealed by measurements of filopodia forces developed upon pulling (Fig. 3), filopodia lifetime (new Fig. 2I and J) and filopodia growth rate (Fig. 5).

- The effects of formin inhibition are interesting. However, the effects on MyoX strongly speak in favor of an alternative interpretation. As the authors mention, integrin-mediated adhesion is necessary for filopodia growth and stabilization. Based on the observation that SMIFH2 reduces the amount of MyoX at filopodia tips (Fig. 4), it is likely that the amount of integrins is reduced as well. Accordingly, SMIFH2 treated filopodia fail to stabilize because of missing integrin-mediated adhesion. This hypothesis has to be tested to justify publication in Nature Communications. Mechanosensing of integrin-mediated adhesion is well established. The novelty of the data would be reduced if this was the explanation for the observed effects.

Indeed, the detachment of actin filaments from the formin at the filopodia tips by SMIFH2 can bring about retrograde movement of not only myosin X but also other associated proteins, including perhaps integrin. In principle, this could explain the reduced adhesiveness of filopodia after prolonged incubation with SMIFH2. To check whether removal of integrin from filopodia tips is the primary mechanism behind the loss of adhesion we performed an experiment with addition of SMIFH2 to the filopodium already attached to the bead. In this experiment myosin II retrograde movement, cessation of the polymerization of the actin core, and the drop of the pulling force generated by filopodium were very prominent but the fibronectin-coated bead remained associated with the filopodium tip via the membrane tether (new fig. S4C,

movie S13 in new numeration). This result strongly suggests that the primary effect of SMIFH2 is disruption of the link between actin core and adhesion receptors while depletion of the receptors is a secondary effect. Moreover, we know that the decrease of adhesion of the filopodia tips to the matrix upon inhibition of myosin II occurs when myosin X is still located at the filopodia tips.

Of note, we examined the lifetime of filopodia induced by truncated myosin X lacking the integrin binding FERM domain. In Cos-7 cells (lacking myosin IIA), this myosin X mutant induced short living filopodia similar to the full-length myosin X. Ectopic expression of myosin IIA increased the lifetime of these filopodia to the same extent as filopodia induced by full-length myosin X (new Fig. 2E, F, I and J). This suggests that association of myosin X with integrin via FERM is not required for the myosin IIA-driven augmentation of filopodia lifetime and probably for the force-dependent filopodia adhesion reinforcement in general.

- Moreover, blebbistatin treatment and MyoIIA knockdown reduce the MyoX level at filopodia tips similar to SMIFH2 (Fig. 3B, C). Again, the question arises whether the observed effects are caused by reduced forces or by reduced stabilization via adhesions.

In general, myosin IIA knockdown and treatment of cells with myosin II inhibitors reduces the level of myosin X at the filopodia tips to a lesser degree than SMIFH2. The myosin X comet tail is clearly seen in the frames showing cells, which were already pretreated with 40 μ M of blebbistatin for 1 hour or transfected with myosin IIA siRNA for 48 hours. As we explained in the answer to the previous question, even in cells treated with SMIFH2 the removal of myosin X and most probably integrin is a secondary effect that follows the uncoupling of actin filaments from the filopodia tips. The same is true for the myosin IIA knockdown cells and cells treated with myosin II inhibitors. Indeed, in the majority of our experiments, labeled myosin X still remained at the filopodia tips when adhesion and force generation by filopodia were already impaired (see in addition Fig. 5 inset). If, as this reviewer suggests, myosin X is an indicator of the presence of integrin, our results (including experiments in which we monitored the effect of addition of Y-27632 on filopodia already attached to the beads) show that inhibition of adhesion and force generation by filopodia preceded the removal of integrin from the filopodia tips.

- The proposed mechanism suggests that filopodia under mechanical load grow by formin-mediated actin polymerization. However, the opposite effect is published (Guillou, 2008, Journal Exp Cell Research): filopodia stop growing as soon as their tip binds to the substrate. Accordingly, the authors should clarify how their model relates to such a situation. The setup shown in Sup.Fig 9 appears ideally suited to answer this question.

We are grateful to this reviewer for attracting our attention to this interesting

study. The filopodia dynamics observed in our system is similar to that described by several previous authors, in particularly (Watanabe et al., 2010).

The question as to why filopodia attachment to the substrate in the experimental system described by (Guillou et al., 2008) cells triggers formation of lamellipodia rather than further elongation of filopodia requires extensive study which is beyond the scope of the present paper. We however included a reference to the Guillou paper into our Discussion.

In summary, the aim of the paper is of general interest for the readers of Nature Communications. The experimental setup is well suited to study this topic, and the quality of the available data speaks in favor of the author's approach. However, the proposed mechanism is not fully supported by the data. Therefore, I suggest additional experiments as outlined above before the manuscript is ready for publication in Nature Communications.

We would like to thank this reviewer for their positive comments and suggestion for publication in Nature Communications. We also agree that additional experiments and changes outlined above improved the quality of the new version manuscript.

Minor points:

- The authors could improve data presentation concerning the aim of the paper. A proper comparison of mechanically manipulated filopodia (Fig. 1,3) to "control filopodia" (Fig. 5) would be helpful.

We believe that our observations of dynamics of unconstrained filopodia are consistent with the results on the growth of filopodia upon application of pulling forces. We discussed this similarity in more details in the revised version of this paper.

- The alteration between "length of filopodia" and "rate of length change" makes it difficult to compare the effects of mechanical manipulation.

In new version we moved the data about filopodia length to the supplementary figures and renamed the parameter of "length of filopodia change" into "rate of filopodia growth" to enhance clarity of reading.

- Also, the potential of the optical tweezer setup is not fully exploited. What happens when the bead in contact with the filopodium is not moved? What happens when the bead is moved away from the cell instead of towards the cell? These experiments would help to understand a proposed filopodia-mediated mechanoresponse.

We are grateful to this reviewer for this suggestion. Filopodia grow through cycles of protrusion-pause-retraction. Our aim was to study the mechnosensing properties of filopodia under stretching forces, in order to mimic the process of filopodia growth under the pulling force generated by the contractile activity of living cells. It will be interesting to study filopodia pausing

and contraction as well. We plan to do these experiments in the future.

All changes, which were made to address reviewers' questions and concerns, are highlighted below in new version of the manuscript.

- Bear, J.E., and Gertler, F.B. (2009). Ena/VASP: towards resolving a pointed controversy at the barbed end. *J Cell Sci* 122, 1947-1953.
- Bornschlogl, T., Romero, S., Vestergaard, C.L., Joanny, J.F., Van Nhieu, G.T., and Bassereau, P. (2013). Filopodial retraction force is generated by cortical actin dynamics and controlled by reversible tethering at the tip. *Proc Natl Acad Sci U S A* 110, 18928-18933.
- Gauthier, N.C., and Roca-Cusachs, P. (2018). Mechanosensing at integrin-mediated cell-matrix adhesions: from molecular to integrated mechanisms. *Curr Opin Cell Biol* 50, 20-26.
- Guillou, H., Depraz-Depland, A., Planus, E., Vianay, B., Chaussy, J., Grichine, A., Albiges-Rizo, C., and Block, M.R. (2008). Lamellipodia nucleation by filopodia depends on integrin occupancy and downstream Rac1 signaling. *Exp Cell Res* 314, 478-488.
- Hu, W., Wehrle-Haller, B., and Vogel, V. (2014). Maturation of filopodia shaft adhesions is upregulated by local cycles of. *PLoS One* 9, 0107097.
- Jacquemet, G., Baghirov, H., Georgiadou, M., Sihto, H., Peuhu, E., Cettour-Janet, P., He, T., Perala, M., Kronqvist, P., Joensuu, H., *et al.* (2016). L-type calcium channels regulate filopodia stability and cancer cell invasion downstream of integrin signalling. *Nat Commun* 7, 13297.
- Leijnse, N., Oddershede, L.B., and Bendix, P.M. (2015). Helical buckling of actin inside filopodia generates traction. *Proc Natl Acad Sci U S A* 112, 136-141.
- Lin, C.H., Espreafico, E.M., Mooseker, M.S., and Forscher, P. (1996). Myosin drives retrograde F-actin flow in neuronal growth cones. *Neuron* 16, 769-782.
- Riveline, D., Zamir, E., Balaban, N.Q., Schwarz, U.S., Ishizaki, T., Narumiya, S., Kam, Z., Geiger, B., and Bershadsky, A.D. (2001). Focal contacts as mechanosensors: externally applied local mechanical force induces growth of focal contacts by an mDia1-dependent and ROCK-independent mechanism. *J Cell Biol* 153, 1175-1186.
- Schirenbeck, A., Bretschneider, T., Arasada, R., Schleicher, M., and Faix, J. (2005). The Diaphanous-related formin dDia2 is required for the formation and maintenance of filopodia. *Nat Cell Biol* 7, 619-625.
- Sydor, A.M., Su, A.L., Wang, F.S., Xu, A., and Jay, D.G. (1996). Talin and vinculin play distinct roles in filopodial motility in the neuronal growth cone. *J Cell Biol* 134, 1197-1207.
- Watanabe, T.M., Tokuo, H., Gonda, K., Higuchi, H., and Ikebe, M. (2010). Myosin-X induces filopodia by multiple elongation mechanism. *J Biol Chem* 285, 19605-19614.
- Yan, J., Yao, M., Goult, B.T., and Sheetz, M.P. (2015). Talin Dependent Mechanosensitivity of Cell Focal Adhesions. *Cell Mol Bioeng* 8, 151-159.
- Zhang, H., Berg, J.S., Li, Z., Wang, Y., Lang, P., Sousa, A.D., Bhaskar, A., Cheney, R.E., and Stromblad, S. (2004). Myosin-X provides a motor-based link between integrins and the cytoskeleton. *Nat Cell Biol* 6, 523-531.

Force dependence of filopodia adhesion: involvement of myosin II and formins

N.O. Alieva¹†, A.K. Efremov^{1,2}†, S. Hu¹, D. Oh¹, Z. Chen¹, M. Natarajan¹,
H.T. Ong¹, A. Jégou³, G. Romet-Lemonne³, J.T. Groves^{1,4}, M.P. Sheetz^{1,5}, J. Yan^{1,2,6},
A.D. Bershadsky^{1,7*}

Affiliations:

¹Mechanobiology Institute, National University of Singapore, T-lab, 5A Engineering Drive 1, Singapore 117411, Singapore.

²Center for BioImaging Sciences, National University of Singapore, 14 Science Drive 4, Singapore 117557, Singapore.

³Institut Jacques Monod, 15 rue Helene Brion 75205 Paris cedex 13, France.

⁴Department of Chemistry, University of California, Berkeley, CA, 94720, USA.

⁵Department of Biological Sciences, Columbia University, New York, New York 10027, USA.

⁶Department of Physics, National University of Singapore, Singapore 117542, Singapore

⁷Weizmann Institute of Science, Herzl St 234, Rehovot, 7610001, Israel

*Correspondence to: alexander.bershadsky@weizmann.ac.il.

† Equal contribution.

Abstract: Filopodia are dynamic membrane protrusions driven by polymerization of an actin filament core, mediated by formin molecules at the filopodia tips. Filopodia can adhere to the extracellular matrix and experience both external and cell generated

pulling forces. The role of such forces in filopodia adhesion is however insufficiently understood. Here, we induced sustained growth of filopodia by applying pulling force to their tips via attached fibronectin-coated beads trapped by optical tweezers. Strikingly, pharmacological inhibition or knockdown of myosin IIA, which localized to the base of filopodia, resulted in weakening of filopodia adherence strength. Inhibition of formins, which caused detachment of actin filaments from formin molecules, produced similar effect. Thus, myosin IIA-generated centripetal force transmitted to the filopodia tips through interactions between formins and actin filaments are required for filopodia adhesion. Force-dependent adhesion led to preferential attachment of filopodia to rigid versus fluid substrates, which may underlie cell orientation and polarization.

INTRODUCTION

Filopodia are ubiquitous cell extensions involved in cell motility, exploration of the microenvironment and adhesion^{1,2}. These finger-like membrane protrusions help cells to determine the direction of movement³, establish contacts with other cells^{4,5} and capture inert particles or living objects (bacteria), which cells subsequently engulf⁶⁻⁹. Filopodia are involved in numerous processes of embryonic development, as well as in cell migration in adult organisms. Moreover, augmented filopodia activity is a hallmark of tumor cells, which use them in the processes of invasion and metastasis¹.

The main element of filopodia is the actin core, which consists of parallel actin filaments with barbed ends oriented towards the filopodium tip, and pointed ends toward the cell body^{1,2,10}. Actin filaments are connected to each other by several types of crosslinking proteins¹¹⁻¹⁴. The filopodia grow via actin polymerization at the

tip, in a process driven by formin family proteins such as mDia2^{15-17, 18}, FMNL2 & 3¹⁹⁻²¹, as well as by actin elongation protein Ena/VASP^{15, 22, 23, 24}. In addition to proteins that crosslink and polymerize actin, filopodia also contain actin based molecular motors, such as myosin X, localized to the tips of the filopodia²⁵. Although the function of myosin X is unclear, it is known to be required for filopodia growth, and its overexpression promotes filopodia formation^{26, 27}.

Adhesion of the filopodia to the extracellular matrix (ECM) is mediated by the integrin family of receptors (e.g. $\alpha_v\beta_3$)^{25, 28}, which are localized to the tip area. One possible function of myosin X is the delivery of integrins to this location²⁵. In addition to integrins, filopodia tips have been shown to contain other proteins involved in integrin mediated adhesion, such as talin²⁹ and RIAM³⁰. Several studies suggest that typical cell matrix adhesions, known as focal adhesions, could in some cases originate from filopodia^{31, 32}. Thus, filopodia could be considered as primary minimal cell matrix adhesion structures.

The hallmark of integrin mediated adhesions of focal adhesion type is their mechanosensitivity³³⁻³⁵. They grow in response to pulling forces applied to them, either by the actomyosin cytoskeleton, or exogenously by micromanipulations, and may play a role in matrix rigidity sensing. Indeed, correlation between focal adhesion size and matrix rigidity is well-documented³⁶⁻³⁸. Filopodia also may participate in matrix rigidity sensing. For example, it was demonstrated that cell durotaxis, a preferential cell movement along a gradient of substrate rigidity is mediated by filopodia³⁹. However, force dependence of filopodia adhesion has not yet been explored.

In the present study, we monitored filopodia adhesion and growth under conditions of pulling with a constant rate. We have demonstrated that adhesion of filopodia to the ECM strongly depends on myosin II activity and found myosin II filaments localized to the base regions of filopodia. Moreover, formin family protein activity at the filopodia tips is also required for filopodia adhesions, most probably through a role in the transmission of force through the actin core, from the filopodium base to the filopodium tip. Thus, filopodia are elementary units demonstrating adhesion-dependent mechanosensitivity.

RESULTS

Dynamics of filopodia induced by expression of myosin X in HeLa-JW cells

Transfection of HeLa-JW cells with either GFP-myosin X or mApple-myosin X resulted in a strong enhancement of filopodia formation in agreement with previous studies⁴⁰. During filopodia movement, myosin X was concentrated at the filopodia tips, forming characteristic patches sometimes also called “puncta” or “comet tails” (fig. S1A, movie S1). Here, we focused on filopodia originating from stable cell edges and extending along the fibronectin-coated substrate. These filopodia demonstrated periods of persistent growth, with an average velocity of 67 ± 6 nm/s (mean \pm SEM, n = 89) interrupted by pauses and periods of shrinking with an average velocity of 28 ± 3 nm/s (mean \pm SEM, n = 100). This behavior is consistent with previously published results⁴¹. In addition to myosin X, the filopodia tips were also enriched in several other proteins such as mDia2, VASP and talin (fig. S1B and D).

To observe the dynamics of filopodia adhesion and protrusion under controlled experimental conditions, we monitored the growth of filopodia that were adhered to fibronectin-coated beads trapped by optical tweezers (see Methods for more details). First, 2 μ m diameter fibronectin-coated polystyrene beads were placed onto filopodia tips by the optical tweezers. After 20-30 s, which is required for the initial attachment of the bead to the filopodium, the movement of microscope piezo stage in the direction from the tip to base of filopodium was initiated (Fig. 1, movies S2, S3, S9A). The force exerted by filopodium on the bead was monitored by measuring the bead displacement from the center of the trap (Δx). In order to preserve the structural integrity of the filopodia, the velocity of the stage movement was set to approximately 10-20nm/s, which is slower than the average velocity of spontaneous filopodia growth. With this setup we observed sustained filopodia growth for more than 10 mins, during which time the tdTomato-Ftractin labelled actin core remained intact (Fig. 1B and 3A, movie S2). Pulling-induced filopodia growth was depended on integrin-mediated adhesion of filopodia tips to fibronectin-coated beads. When the beads were coated with concanavalin A instead of fibronectin, application of force never induced the growth of filopodia actin cores. Instead, pulling via concanavalin A-coated bead resulted either in detachment of filopodia tips from the beads (18%), or withdrawal of the bead from the trap (27%), or, in majority of cases (55%), in the formation of membrane tethers, n = 11 (movie S4).

During the first 3 mins after stage movement commenced, the exerted force approached the maximal value of 3-5 pN. However, it then dropped to the 1.5-2pN range, and remained at this level for a further 1-3 mins, after which it rapidly increased again (Fig. 1C). In a typical experiment, we detected 2-4 such peaks with a

mean peak force value of 3pN alternating with the 1-3 min periods of lower force (1.5-2pN).

The pattern of force dependent filopodia elongation described above was typical for myosin X-induced filopodia, but not for filopodia induced by constitutively active Cdc42 Q61L. Cdc42-induced filopodia attached to laser-trapped fibronectin-coated beads did not grow upon stage movement and eventually pulled the beads out of the trap (movie S5). In this study, we exclusively focused on the filopodia-induced by myosin X overexpression.

Immediately after attachment of the bead to the filopodium tip, the myosin X patch, or a significant portion that pinched off the main myosin X mass, started to move centripetally with an approximate velocity of 31 ± 5 nm/s (mean \pm SEM, $n = 42$). Co-expression of myosin X with VASP in HeLa-JW cells revealed that the retrogradly moving myosin X patches were always colocalized with the patches of VASP (fig. S1B middle panel). Moreover, centripetal movement of myosin X patches in cells transfected with photoactivatable β -actin and myosin X proceeded with the same velocity as the movement of a photoactivated actin spots in the same filopodia (fig. S1C, movie S7). However, despite the retrograde movement of a large part of myosin X, it did not entirely disappear from the filopodium tip and the original amount was fully restored after several minutes (Fig. 1C, kymograph), even though detachment and subsequent centripetal movement of myosin X portions from the filopodium tip were occasionally observed throughout the entire period of force-induced filopodium growth (movie S3).

Involvement of myosin II in filopodia dynamics

Expression of GFP labeled myosin light chain in HeLa-JW cells showed that myosin II does not localize to the filopodia tips or shafts, but is often located at the proximal ends of the filopodia (Fig. 2A and B top, movie S8A). Structure illumination microscopy (SIM) revealed few myosin II mini-filaments on either side of the filopodium base.

Localization of exogenous myosin IIA at the filopodia bases was especially prominent in Cos-7 cell model. It should be noted that Cos-7 cells express myosin IIB and IIC, but do not contain myosin IIA^{42,43}. Upon transfection with myosin X, the wild type Cos-7 cells form filopodia with a short life-time that apparently do not adhere to the fibronectin-coated substrate (Fig. 2C, movie S8B). Co-transfection of these cells with GFP-myosin IIA heavy chain resulted in formation of numerous bipolar myosin IIA filaments frequently localized to the bases of filopodia (Fig. 2B bottom, 2D, movie S8C). Majority of filopodia in such cells were associated with myosin IIA filaments during the period of observation. **Indeed, the analysis of the histories of 25 filopodia from the movie S8C revealed that 17 of them (68%) had one or more myosin IIA doublets at their bases during the movie duration (4 min 5 s). Myosin X-containing filopodia in Cos-7 cells co-transfected with myosin IIA were somewhat longer than those in wild type Cos-7 lacking myosin IIA (fig. S2). Moreover, expression of myosin IIA significantly increased the lifetime of myosin X-induced filopodia (Fig. 2J) as it can be inferred from the analysis of survival fraction of the filopodia cohort (Fig. 2I).**

In agreement with previous studies, formation of filopodia in Cos-7 cells can be also induced by expression of truncated myosin X lacking FERM domain (Fig. 2E, movie S8D)⁴⁴. Co-expression of myosin IIA significantly enhanced the lifetime of the

filopodia induced by this truncated myosin X (Fig. 2I and J, movie S8E) showing that binding of myosin X to integrin via FERM domain²⁵ is not required for myosin IIA-driven enhancement of filopodia lifetime. In contrast, expression of GFP-myosin IIB heavy chain in Cos-7 cells did not affect the filopodia lifetime (Fig 2I, J). Consistently, myosin IIB was not localized to the filopodia base in these cells (Fig. 2G, movie S8F).

Mutant myosin IIA N93K has reduced myosin ATPase and motor activity, but preserves the ability to bind to actin filaments^{45, 46}. GFP fusion construct of this mutant forms filaments localized to the bases of myosin X-induced filopodia in Cos-7 cells, similarly to wild type myosin IIA (Fig. 2H). However, unlike the wild type, the mutant myosin IIA N93K only slightly enhanced the lifetime of filopodia (Fig. 2I, J). Thus, motor activity of myosin IIA is critically important for the effect of myosin IIA on filopodia life time.

We further studied how the presence and activity of myosin IIA affects unconstrained and force-induced filopodia growth. The function of myosin II was suppressed in HeLa-JW cells in three separate experiments: through the inhibition of ROCK by Y27632, by siRNA mediated knockdown of myosin IIA heavy chain (MYH9), and through the inhibition of myosin II ATPase activity by light-insensitive S-nitro-blebbistatin. Inhibition of ROCK blocks myosin II regulatory light chain (RLC) phosphorylation, which interferes with myosin II filament assembly⁴⁷⁻⁵⁰. As a result, HeLa-JW cells treated with 30 μ M of Y27632 essentially lose their myosin II filaments in less than half an hour. siRNA knockdown of MYH9 also resulted in a loss of most of the myosin II filaments (fig. S3). Inhibition of myosin II ATPase activity by S-nitro-blebbistatin did not disrupt myosin II filaments⁵⁰, although this treatment did result in profound changes to the organization of the actomyosin

cytoskeleton, including a loss of stress fibers. While treatment with either S-nitroblebbistatin or Y27632 did not change the average filopodia length, myosin IIA knockdown resulted in moderate but statistically significant shortening of filopodia, (fig. S2), which is consistent with the results on filopodia length in wild type and myosin IIA expressing Cos-7 cells mentioned above (fig. S2). Myosin X-positive comet tails persisted at the tips of filopodia in both HeLa-JW and Cos-7 cells irrespectively to inhibition or lack of myosin IIA.

Despite the morphological integrity of filopodia being preserved in myosin II inhibited or depleted cells, adhesion of filopodia to the ECM was significantly impaired. While in control HeLa-JW cells application of pulling force via fibronectin-coated bead induced sustained growth of attached filopodia accompanied by the development of up to ~ 5 pN force, in the cells with impaired myosin II activity the filopodia detached earlier, after developing rather small forces (Fig. 3B-C, E, movies S9 B-C). This suggests filopodia are unable to establish a proper adhesion contact in the absence of active myosin IIA. We also examined the immediate effect of Y27632 during the force-induced sustained growth of filopodia. Shortly after the drug was added to the experimental chamber, filopodia indeed detached from the bead (fig. S4, movie S10).

Interaction between actin filaments and formins is required for filopodia adhesion and myosin X localization

In myosin X-induced filopodia, the formin mDia2 is localized to the filopodia tips, and overlaps with myosin X patches (fig. S1B, top panels). Small molecular inhibitor of formin homology domain 2 (SMIFH2)⁵¹ was used to investigate the role of

formins in attachment of filopodia to fibronectin-coated beads. We found that in SMIFH2 (40 μ M, 1hour) treated cells, adhesion of filopodia to the beads was impaired in a similar way to the adhesion of filopodia in myosin II inhibited/depleted cells. The duration of contact between the filopodia and bead was significantly shorter, and the maximal force exerted by filopodia to the bead was significantly weaker than in control cells (Fig. 3D and E, movie S9D).

While the number of filopodia in cells treated with SMIFH2 remained the same as in control cells and their mean length decreased only slightly (Fig. 4A), only about 25% of these filopodia preserved myosin X comet tails at their tips 1-2 hours following SMIFH2 addition (fig. S5; filopodia in 14 cells were scored in 2 independent experiments) despite originally being induced by over-expression of myosin X. We found that SMIFH2 induced disintegration of the comet tails into myosin X patches, which rapidly moved centripetally towards the cell body (Fig. 4B and C top, movie S11 and S12A). Although such movement was occasionally observed in control cells (above), it was much more prominent in cells treated with SMIFH2, and led to the gradual disappearance of myosin X from the filopodia tips. Of note, the movement of myosin X patches in SMIFH2 treated cells occurred together with the movement of its partner VASP⁵², another protein associated with barbed ends of actin filaments (fig. S6, movie S6).

The velocity of retrograde movement of myosin X patches in filopodia of cells treated with SMIFH2 was 84 ± 22 nm/s (mean \pm SEM, n = 45) vs 31 ± 5 nm/s in control cells (see the first section of Results). Such movement might be a result of the detachment of myosin X-bearing actin filaments from the filopodia tips. Once free, their

subsequent retrograde movement is driven by myosin II located at the bases of the filopodia. Indeed, incubation of SMIFH2 treated cells with Y27632 efficiently stopped the retrograde movement of the myosin X positive patches, reducing their average velocity to 0.66 ± 0.14 nm/s (mean \pm SEM, n = 17), see Fig. 4C and movies S12A-C.

We also performed an experiment with addition of SMIFH2 to the filopodium already attached to the bead. The drug was added during the force-induced sustained growth of filopodia. In about 10 seconds after addition of the drug, myosin X patch disintegration and cessation of the polymerization of the actin core were observed. The force generated by a filopodium dropped to a zero value in ~ 3 minutes (fig. S4C and D, movie S13). The fibronectin-coated bead, however, remained associated with the filopodium tip via the membrane tether (movie S13) and did not detach even upon switching the trap off. This suggests that addition of SMIFH2 disrupted the link between actin core and the integrin adhesion receptors, which still remain associated with the membrane tether at the filopodium tip.

To prove that SMIFH2 treatment can detach actin filaments from formin located at the filopodia tips, we performed in vitro experiments where the actin filaments were growing from immobilized formin mDia1 construct (FH1FH2DAD) in the absence or presence of SMIFH2. Following treatment with $100\mu\text{M}$ SMIFH2, a rapid decrease in the fraction of filaments remaining associated with immobilized formins under conditions of mild shear flow was observed (fig. S7). Thus, SMIFH2 treatment disrupted physical contacts between formin molecules and actin filaments. Therefore,

SMIFH2-induced rapid centripetal movement of myosin X is driven by myosin II mediated pulling of actin filaments detached from the filopodia tips.

Effect of myosin II and formin inhibition on the growth of unconstrained filopodia

In addition to the studies of filopodia growing in response to pulling forces, we examined the effects of myosin II and formin inhibition on the dynamics of free, unconstrained filopodia (Fig. 5 inset). We found that knockdown of myosin IIA and cell treatment with Y27632 or S-nitro-blebbistatin efficiently blocked growth and retraction of unconstrained filopodia, resulting in suppression of filopodia dynamics. In untreated myosin X-expressing cells, the fraction of filopodia in the “pause” state (with the growth rate between -15 and +15nm/s) was 13% (n = 194). At the same time, fractions of the “pausing” filopodia were 90% (n = 41), 80% (n = 83) and 55% (n = 42) for myosin IIA knockdown, S-nitro-blebbistatin-treated and Y27632-treated cells, respectively (Fig. 5). Similarly, the fraction of “pausing” filopodia in cells treated with the formin inhibitor SMIFH2 was 75% (n = 44) (Fig. 5)

In this study, we have shown that filopodia adhesion to the ECM is a force dependent process. This conclusion is based on experiments in which sustained growth of filopodia was maintained by the application of pulling force at the interface between a fibronectin-coated bead, and the tip of a filopodia. With this setup, inhibition of myosin II filament formation or myosin II ATPase activity resulted in suppression of filopodia adhesion to fibronectin-coated bead. It should be noted that our experiments were performed on filopodia induced by over-expressing myosin X and, therefore, our conclusions are, strictly speaking, only valid for this class of filopodia. However,

myosin X has been shown to be a universal component of filopodia ²⁶, so employment of such an experimental system does not restrict the generality of our finding.

Since myosin II is located at the bases of filopodia (Fig. 2), a question requiring further clarification is how the pulling force is transmitted to the filopodia tips. Our data are consistent with the idea that this force is transmitted by the filaments of the actin core attached to formin molecules and the filopodia tips. We have shown that formin inhibition by SMIFH2 suppresses filopodia adhesion to the beads in the same manner as inhibition of myosin II. Moreover, we demonstrated that SMIFH2 treatment led to a rapid, myosin II-dependent, movement of actin filament associated proteins, myosin X and VASP, from the filopodia tips towards the cell body. Since centripetal movements of myosin X patches proceed together with the movements of photoactivated actin spots, we interpret the acceleration of myosin X/VASP movement upon SMIFH2 treatment as an evidence of actin filament detachment from formins at the filopodia tips. Indeed, *in vitro* experiments demonstrated that addition of SMIFH2 to actin filaments growing from the immobilized formins under a condition of moderate flow results in the detachment of actin filaments from the formin molecules. Together, these experiments suggest that myosin II inhibition, or inhibition of the formin-mediated association between actin filaments and the filopodia tips, make filopodia unable to form stable adhesions with fibronectin-coated beads. This in turn prevents them from growing upon force application.

To check whether adhesion and growth of filopodia require the pulling force, we compared behavior of unconstrained filopodia on rigid substrate with that on fluid supported lipid bilayer (SLB) where the traction forces cannot develop ⁵³. To this end,

we created a composite substrate on which rigid surface was covered by orderly patterned small islands ($D = 3\mu\text{m}$) of SLB. Both rigid and fluid areas were coated with integrin ligand, RGD peptide, with the same density (fig. S8). We found that dynamics of filopodia extended over rigid regions of this substrate was similar to that of filopodia growing on rigid fibronectin-coated substrate used in rest of experiments. At the same time, filopodia that encountered the SLB islands could not attach properly and as a result spent over such substrate significantly shorter time than over rigid area of the same geometry (fig. S9A and B, movie S14). Accordingly, the average density of filopodia tips remaining inside the SLB islands during period of observation ($> 10\text{min}$) was lower than that on the rigid substrate (fig. S9C). Thus, not only inhibition of myosin II or formin, but also micro-environmental conditions under which filopodia tips do not develop traction force, prevent proper adhesion of filopodia.

DISCUSSION

In the present study, we have demonstrated that adhesion of myosin X-induced filopodia to extracellular matrix depends on the forces generated by associated myosin IIA filaments. Our interest to filopodia induced by myosin X is in part justified by the fact that this protein is overexpressed in many types of cancer cells and adhesion of myosin X containing filopodia to the matrix may play an important role in the cancer cells invasion and metastasis.⁵⁴

The structured illumination microscopy (SIM) revealed the presence of individual myosin IIA filaments at the base of filopodia. The force generated by one bipolar myosin IIA filament (consisting of about 30 individual myosin molecules⁵⁵, 15 at

each side) can be estimated based on the stall force for individual myosin IIA molecule (3.4pN according ⁵⁶) and duty ratio (5-11% according ⁵⁷) as 2.6-5.6pN. This value is consistent with pulling forces generated by individual filopodia as measured in our experiments. In the presence of formin inhibitor SMIFH2 detaching the filaments of the actin core from formin molecules at the filopodia tips, the filaments retract centripetally towards the cell body as can be inferred from the analysis of movements of myosin X and VASP clusters associated with these filaments. Inhibition of myosin IIA filament formation by Y-27632 suppressed such retrograde movement inside the filopodia. The forces generated by myosin IIA filaments appear to be sufficient to overcome the actin filament crosslinking inside the core and generate retrograde filament movement. Altogether, these data suggest that myosin IIA-generated forces are transmitted to filopodia tips via actin cores associated with formin molecules at the tips.

Dependence of filopodia adhesion on myosin IIA-generated forces was revealed in two types of experimental systems. First, we studied the effects of application of pulling force on the filopodia. In control HeLa-JW cells, filopodia attached to optically trapped fibronectin-coated beads were growing upon slow movement of the microscope stage and developed periodic 2-5pN forces. Myosin IIA-depletion or cell treatment with inhibitors of myosin II assembly (Y-27632) or motor activity (blebbistatin) led to significant decrease of the forces exerted by filopodia and subsequent detachment of filopodia from the trapped beads. Thus, myosin IIA-generated forces are required for the adhesion of filopodia to the beads and for force-induced filopodia growth.

Second, the function of myosin IIA in the regulation of filopodia lifetime in Cos-7 cells was studied. In Cos-7 cells lacking myosin IIA, myosin X overexpression induces only unstable short living filopodia. Their lifetime, however, can be significantly increased upon expression of exogenous myosin IIA. Such increase of the lifetime can be explained by stabilization of adhesion of these filopodia to the substrate. In contrast, myosin IIB overexpression does not increase the filopodia lifetime and does not localize to filopodia bases. Importantly, even though myosin IIA mutant with compromised motor activity (myosin IIA N93K) still localizes to filopodia bases, it only slightly enhances the filopodia lifetime. Thus, the motor function of myosin IIA rather than its actin crosslinking function⁴⁵ is important for stabilization of filopodia adhesion.

Force-driven growth of filopodia attached to the bead is an integrin-dependent process and was not observed in experiments with integrin-independent adhesion of filopodia to beads coated by concanavalin A. At the same time, association of integrin with myosin X FERM domain was dispensable for the myosin IIA-driven augmentation of the filopodia lifetime. This suggests that potential link between integrin and actin filaments via myosin X is not critically important for filopodia adhesion stabilization.

A major linker between integrin and actin filaments, talin, has been detected at the filopodia tips in this and other publications³⁰. Previously, it was established that force-driven unfolding of talin facilitates interaction of talin with another adhesion complex component, vinculin, resulting in reinforcement of the association between talin and actin filaments⁵⁸⁻⁶¹. The question whether this mechanism is applicable to filopodia adhesion reinforcement requires additional studies. While vinculin

enrichment was detected in shafts but not tips of filopodia⁶², a RIAM protein that competes with vinculin for talin-binding and localizes to filopodia tips^{30, 63} may be involved in force dependent filopodia adhesion. RIAM binds Ena/VASP and profilin⁶⁴ and could recruit these actin polymerization-promoting proteins to the filopodia tips.

Another mechanism of myosin II-dependence of filopodia adhesion and growth might involve formin-driven actin polymerization known to be a major factor in filopodia extension^{15-17, 19-21}. Recent studies demonstrate that formin-driven actin polymerization can be enhanced by pulling forces⁶⁵⁻⁶⁸. Thus, myosin II-generated force transmitted via actin core to formins at the filopodium tip can stimulate actin polymerization, promoting filopodia growth. Polymerization of actin could also be important for recruitment of new adhesion components to the filopodia tips and adhesion reinforcement⁶⁹.

Filopodia adhesion is tightly associated with filopodia growth and shrinking. In our experiments, inhibition of myosin II and formins not only suppressed filopodia adhesion but also resulted in reduction of motility of filopodia along the substrate. During pulling-induced growth of bead-attached filopodia, periods of filopodia elongation alternate with periods of growth cessation accompanied by increase of the pulling force. Notably the periods of force development preceded the periods of filopodia elongation. Thus, force developed during growth cessation may trigger the subsequent filopodia growth. Similarly, the growth of unconstrained filopodia along rigid substrate can proceed via periods of attachment, development of force, and consequent filopodia elongation⁴¹. Inhibition of force generation or transmission

suppresses such dynamics. Interestingly, in filopodia of other types the attachment of filopodia tips to the substrate instead of further elongation led to formation of associated lamellipodia⁷⁰. It is not yet known, whether this switch is also mediated by myosin II-generated force.

Our finding that filopodia adhesion and growth are both is force-dependent explains how filopodia could respond differently to substrates of varying stiffness. On a stiff substrate, the force generated by myosin II and applied to the adhesion complex will develop faster than on a compliant substrate⁷¹. Accordingly, filopodia adhesion should be more efficient on stiff substrates than on compliant substrates. Indeed, we showed that the contacts of filopodia with RGD ligands associated with fluid membrane bilayer were less stable than with the areas of rigid substrate covered with RGD of the same density. These considerations can explain involvement of filopodia in the phenomenon of durotaxis³⁹, a preferential cell movement towards stiffer substrates⁷². This may provide a mechanism to rectify directional cell migration.

Orientation based on filopodia adhesion is characteristic for several cell types, in particular for nerve cells. The growth cones of most neurites produce numerous filopodia, and the adhesion of these filopodia can determine the direction of neurite growth^{73,74}. Interestingly, the filopodia-mediated traction force in growth cones is myosin II-dependent⁷⁵ and application of external force can regulate the direction of growth cone advance⁷⁶. The results from these experiments can now be explained by preferential adhesion/growth of filopodia, which experience larger force. The mechanosensitivity of filopodia adhesion provides a mechanism of cell orientation that complements that mediated by focal adhesions. Focal adhesions are formed by

cells attached to rigid two-dimensional substrates, whereas filopodia adhesion can be formed by cells embedded in three-dimensional fibrillar ECM network. Thus, further investigation of filopodia mechanosensitivity could shed a new light on a variety of processes related to tissue morphogenesis.

METHODS

Cell culture and transfection

Hela-JW, a subline of a HeLa cervical carcinoma cell line derived in the laboratory of J. Willams (Carnegie-Mellon University, USA) on the basis of better attachment to plastic dishes⁷⁷, was obtained from the laboratory of B. Geiger⁷⁸. Cos-7 (African green monkey fibroblast-like cell line, was obtain from ATCC (ATCC[®] CRL-1651[™]). The cells from both cell lines were grown in DMEM- Dulbecco's Modified Eagle Medium supplemented with 10% fetal bovine serum (FBS), 100 U ml⁻¹ penicillin/streptomycin, 2 mM glutamine and 1 mM sodium pyruvate at 5% CO₂ at 37 °C. HeLa-JW cells were transfected with DNA plasmids by electroporation (two pulses of 1005 V for 35 ms) using Neon transfection system (Thermofisher), while Cos-7 cells – by jetPRIME transfection reagent (Polyplus) according to manufacturer protocol. Expression vectors encoding the following fluorescent fusion proteins were used: GFP-myosin X and **GFP-Myo10 ΔFERM**^{40, 44} (gift from R. Cheney, University of North Carolina Medical School, USA), reduced size myosin X construct (without 3'UTR) derived from GFP-myosin X was sub-cloned into mApple-C1 cloning vector backbone (M. Davidson collection, Florida State University, USA, kindly provided by P. Kanchanawong, MBI, NUS, Singapore), tdTomato–Ftractin⁷⁹ (gift from M. J. Schell, Uniformed Services University, Bethesda, Maryland, USA), mCherry-Utrophin^{80, 81} was kindly provided by W. Bement (University of Wisconsin,

Madison, USA), mTagBFP-Lifeact was obtained from M. Davidson collection, Florida State University, USA (Addgene), myosin regulatory light chain (RLC)-GFP⁸² (gift from W. Wolf and R. Chisholm, Northwestern University, Chicago, Illinois, USA), GFP-myosin IIA heavy chain was sub-cloned from pTRE-GFP-NMHCIIA (from R. Adelstein, NHLBI, NIH, USA) into pEGFP-C3 by MluI & HindIII cloning sites (cloned by M. Tamada, M. Sheetz laboratory), GFP-myosin IIA N93K⁴⁹ was kindly provided by Dr. Vicente-Manzanares (Universidad Autónoma de Madrid, Madrid, Spain), GFP-myosin IIB, mCherry-VASP and mCherry-talin (M. Davidson collection, Florida State University, kindly provided by P. Kanchanawong, MBI, NUS, Singapore), full-length mDia2 was sub-cloned from GFP-C1-mDia2⁸³ (gift from S. Narumiya, Faculty of Medicine, Kyoto University, Japan) into mCherry-C1 vector (Clontech), PAmCherry-b-actin was kindly provided by V. Verkhusha (Albert Einstein College of Medicine, NY, USA). All cell culture and transfection materials were obtained from Life Technologies.

Live cell imaging and confocal microscopy

Following electroporation, cells were seeded at a density of 2×10^4 cells ml⁻¹ in 2ml onto 35mm glass based dishes with 12 or 27 mm bottom base cover glass #1 in diameter (Iwaki, Japan) coated with 10 µg ml⁻¹ fibronectin (Calbiochem) for 20 min. Cells were imaged in Leibovitz's L-15 medium without Phenol Red containing 10% FBS at 5% CO₂ at 37 °C. Snapshot or time-lapse images were acquired with a spinning-disc confocal system (PerkinElmer Ultraview VoX) based on an Olympus IX81 inverted microscope, equipped with a 100× oil immersion objective (1.40 NA, UPlanSApo), an EMCCD camera (C9100-13, Hamamatsu Photonics), and Volocity control software (PerkinElmer).

We perform photoactivation experiments by activation of a defined region inside filopodia using blue diode laser (405 nm, 100mW). Photoactivation and livecell imaging were performed on CSU-W1 spinning disk confocal system on Nikon Eclipse Ti-E inverted microscope with Perfect Focus System, controlled by MetaMorph software (Molecular device) supplemented with a 100x oil 1.45 NA CFI Plan Apo Lambda oil immersion objective and sCMOS camera (Prime 95B, Photometrics).

For structured illumination microscopy (SIM), two types of equipment were used: (1) Live SR (Roper Scientific) module on Nikon Eclipse Ti-E inverted microscope (specifications of setup described above), (2) Nikon N-SIM microscope, based on a Nikon Ti-E inverted microscope with Perfect Focus System controlled by Nikon NIS-Elements AR software supplemented with a 100x oil immersion objective (1.40 NA, CFI Plan-ApochromatVC) and EMCCD camera (Andor Ixon DU-897). For life cell imaging the samples were mounted in a humidified cell culture chamber and maintained at 37 °C with 5% CO₂. All SIM images with obtained with system (1) except images on Fig. 2A and B (top panel), which were obtained with set up (2).

Transfection of siRNA and immunoblotting

Cells were seeded into a 35 mm dish on day 0 and transfected with 100µM of MYH9 ON-TARGET plus SMART pool siRNA (L-007668-00-0005, Dharmacon) using ScreenfectTMA (WAKO, Japan) on day 1. Control cells were transfected with scrambled control ON-TARGET plus Non-targeting pool siRNA (D-001810-10, Dharmacon). Transfection of plasmid GFP-myosin X and tdTomato-Ftractin was performed in the evening of the day 1 using Jet Prime transfection reagent (Polyplus)

and cells were imaged on day 2. For assessment of myosin IIA heavy chain expression, transfected cells were lysed in RIPA buffer on day 2 (exactly 24 hours following siRNA transfection) and analyzed by Western blotting with primary rabbit antibodies to the myosin IIA tail domain (M8064, Sigma-Aldrich, dilution 1:1000); staining of α -tubulin with monoclonal DM1A antibody (T6199, Sigma-Aldrich, dilution 1:5000) was used as a loading control. HRP-conjugated anti-rabbit IgG (Bio-Rad, 1706515, dilution 5000) and anti-mouse IgG (A4416, Sigma-Aldrich, dilution 1:10000) were used as secondary antibodies, respectively.

Immunofluorescence antibody staining

Anti-myosin IIA tail domain (M8064, Sigma-Aldrich, dilution 1:800), and 405 Alexa-Fluor-conjugated secondary antibodies (A31556, Molecular Probes, dilution 1:200). Cells were pre-fixed by addition of warm 20% PFA (Tousimis) into medium (to make 8% solution) and subsequent 15 min incubation at room temperature. This was followed by fixation and permeabilization by 3.7% PFA, 0.2% glutaraldehyde and 0.25% Triton X-100 in PBS for 15 min. The fixed cells were then washed two times with PBS and blocked with 5% bovine serum albumin (BSA) for 30 min. The cells were then stained with primary antibodies overnight at 4 °C, and incubated with secondary antibodies for 1h at room temperature.

Drug treatment

For formin drug inhibition studies, cells were incubated with 20 or 40 μ M SMIFH2 (4401, TOCRIS, UK) in serum containing DMEM for 1-2h at 5% CO₂, 37 °C. In *in vitro* experiments, SMIFH2 (340316-62-3, Sigma-Aldrich) was used at a concentration of 100 μ M (see below). For myosin II inhibition studies, 30 μ M or 50 μ M Rho-kinase (ROCK) inhibitor Y27632 (Y0503, Sigma-Aldrich) or 20 μ M S-nitro-

blebbistatin (13013-10, Cayman Chemicals) was added for 10-20 min before the experiments or directly during observations. All inhibitors remained in the medium during the entire period of observation.

Optical tweezers and data acquisition

All experiments involving filopodia pulling were carried out on a Nikon A1R confocal microscope adapted for the use of laser tweezers. 2.19 μm diameter polystyrene beads (PC05N, Bangs Laboratories) were coated with fibronectin (341635, Calbiochem) according to a previous protocol⁸⁴. For bead trapping we used an infrared laser ($\lambda = 1064\text{nm}$, power 0.5-1W, YLM-5-LP-SC Ytterbium Fiber Laser, IPG photonics).

To determine the forces, F , applied to the bead by the optical trap, we measured the displacement of the bead from the trap center, Δx , and then knowing the stiffness of the trap, k , the force was calculated as: $F = k\Delta x$. The trap stiffness was calibrated using the equipartition method⁸⁵ by tracking the fluctuations of a bead trapped by optical tweezers, using an Andor Neo sCMOS camera, at 100fps. The displacement of the beads from the center of the optical trap in confocal microscopy observations was monitored using piA640-210gm camera (Basler) at 0.5-1fps and Metamorph software for tracking. The smallest detectable bead displacement was $\sim 5\text{ nm}$, corresponding to the smallest force measured of $\sim 0.04\text{ pN}$. The moment of detachment of filopodia from the bead was detected by return of the bead to the center of optical trap. To confirm the detachment, we then switched the trap off, which resulted in release of the bead and its disappearance from the field of view unless it remained connected to filopodia via the membrane tethers.

For laser trap experiments, HeLa-JW cells transfected by electroporation with GFP-myosin X and tdTomato-Ftractin, were seeded at a density of 2×10^4 cells ml^{-1} in 750 μl onto chambered #1 borosilicate cover glasses (155383, Lab-Tek) coated with fibronectin (341635, Calbiochem) by incubation in 1 ng ml^{-1} solution in PBS for 20min at 37.0 °C. A reduced concentration of fibronectin (compared to that used for regular cell observations) was used to prevent the beads sticking to the cover slip. Chambers with cells were mounted on P-545.3R7 stage equipped with E-545 Plnano piezo controller (Physik Instrumente), which was moved in order to generate the pulling force between filopodia and trapped beads. The velocity of the stage movement was maintained in a range 10-20 nm/s by PIMikroMove 2.15.0.0 software. The specimen were incubated in a custom-built microscope hood at 37.0 °C, 5% CO_2 humidified environment. Simultaneously with pulling, the cells were imaged using lasers $\lambda = 488\text{nm}$ and 561nm for excitation of GFP and tdTomato, respectively. Collected experimental data were processed by particle tracking algorithms of the Metamorph software.

***In vitro* assay for formin processivity in the presence of SMIFH2**

We used a microfluidics based assay to assess formin processivity in the course of formin-driven actin polymerization. The method has been described in more detail elsewhere ⁶⁶. Briefly, formin construct Snap-mDia1(FH1FH2DAD)-6xHisTag was specifically anchored to the bottom surface of a microchamber, using a biotinylated pentaHis antibody (Qiagen) and streptavidin to bind to a biotin-BSA-functionalized glass surface which was further passivated with BSA. The microfluidics PDMS chamber height was 60 μm . Immobilized formin constructs were allowed to nucleate actin filaments by exposing them for 30s to a 2 μM solution of 20% Alexa488-labeled

at Lys³²⁸ actin⁸⁶ and 0.4 μ M profilin in a buffer containing 10 mM Tris-HCl (Euromedex) pH 7.8, 1 mM MgCl₂ (Merck), 200 μ M ATP (Roche), 50 mM KCl (VWR Chemicals) and supplemented with 5 mM DTT (Euromedex), 1 mM DABCO (1,4-diazabicyclo[2.2.2]octane) to reduce photobleaching. Recording of actin filament elongation was started after the buffer was changed to contain 1 μ M unlabeled actin and 4 μ M profilin, with or without 100 μ M SMIFH2 (Sigma-Aldrich). The microfluidics flow was kept low enough to ensure that the viscous drag had no impact on filament detachment from formin. Images were acquired every 10 s on a Nikon Ti-E microscope using a 60x objective and a Hamamatsu Orca Flash 4.0 V2+ sCMOS camera. Actin was purified from rabbit muscle⁸⁷. Recombinant Profilin I and Snap-mDia1(FH1FH2DAD)-6xHisTag were expressed in *E. Coli* and purified⁸⁸.

Filopodia density and length measurement; tracking filopodia tips

Cells labeled with GFP-myosin X, tdTomato-Ftractin or mCherry-Utrophin were imaged in 488nm (green) and 594nm (red) channels, simultaneously. Cell segmentation was performed using an in-house algorithm implemented as a Fiji macro. The segmented cell was analyzed using MATLAB software CellGeo⁸⁹ to estimate filopodia density and their length. First, each data set, which corresponded to 10 frames per movie taken at 1-2 fps at x100, was averaged over time to generate a smooth image for each channel. The averaged images from the green and red channels were then summed to produce an enhanced image for segmentation. To optimize the segmentation results, filopodia segmentation and cell body segmentation was conducted in two separate steps. For filopodia segmentation, a background subtraction procedure with a rolling ball of 20-pixel radius was applied first, followed by Triangle auto-thresholding (Fiji Auto Threshold v1.16.1). For cell body

segmentation, Gray Morphology open operator with a circle of 5-pixel radius was used (Fiji Morphology) to remove filopodia before applying Li auto-thresholding (Fiji Auto Threshold v1.16.1). The final segmentation result was obtained by combining the masks from the above two steps. After segmentation, the BisectoGraph module of CellGeo⁸⁹ was used to partition each cell into the cell body and individual filopodia, based on the parameters called critical radius (roughly the half of maximal filopodia width) and minimum filopodia length. In our analysis, a critical radius of 0.7 μm and a minimum filopodia length of 1.5 μm were used. Note that, our definition of filopodia length slightly differed from that used in original paper by⁸⁹. Namely, in our analysis the filopodia length is defined to be the distance from the filopodia tip to the cell body boundary. The MATLAB code for computing this length was kindly provided by D. Tsygankov (Georgia Institute of Technology, Atlanta, GA, USA). For filopodia density quantification, the perimeter of the cell body was measured and the number of filopodia per unit of length (μm^{-1}) was computed.

To track the filopodia tips, the Imaris 8.3.1 spots tracking procedure was used. The filopodia tips were identified using segmentation procedure described above followed by MATLAB binary morphology function `bwmorph` with 'skel' and 'endpoints' operations.

Unconstrained filopodia elongation and shrinking rates were obtained by linear regression fitting of the filopodia tips trajectories at the regions corresponding to the periods of filopodia persistent growth or shrinking, respectively.

Supported lipid bilayer micro patterns chamber and filopodia tips trajectory analysis on rigid and fluid substrates

The cleaned glass coverslips were coated with PLL-g-PEG-biotin (Susos) for two hours followed by UV etching using 3 μm circular shape arrays of chromium (Cr) photomask⁹⁰, which resulted in removal of PLL-g-PEG-biotin polymers on the etched circular regions. After multiple rinses with UHQ water, phospholipid vesicles were introduced to the surface for 5 min, allowing the self-assembly of lipid bilayer on the etched glass surfaces. Phospholipid vesicles were synthesized using the existing protocol⁹⁰. Specifically, 98 % of DOPC (1,2-dioleoyl-sn-glycero-3-phosphocholine) was mixed with 2 % of biotin-DOGC (1,2-dioleoyl-sn-glycero-3-[(N-(5-amino-1-carboxypentyl)iminodiacetic acid)-succinyl]) biotin lipids. The lipid solution was further diluted with Tris buffered saline (TBS, Sigma-Aldrich) at a ratio of 1:1 before introduction. The glass coverslips were assembled into the donut shape chamber at the UHQ water reservoir. The hybrid substrate was then incubated with 0.1 % bovine serum albumin (BSA, Sigma Aldrich) for two hours to reduce non-specific protein absorption. Next, 1 $\mu\text{g}/\text{ml}$ of Dylight-405 NeutrAvidin (Thermo Fisher) was introduced to the chamber for one hour, and then incubated with 1 $\mu\text{g}/\text{ml}$ of RGD-biotin (Peptides International) for one hour. Multiple rinses with UHQ were applied after each step. A fluidity test of RGD molecules at the surface of supported lipid bilayer (SLB) was performed by fluorescence recovery after photobleaching. The concentration of RGD on both SLB and PLL-g-PEG polymer was similar, based on the estimated fluorescence of Dylight-405 NeutrAvidin (fig. S8).

HeLa-JW cells expressing a myosinX-GFP chimera were introduced to the chamber, allowing cells to adhere to and spread on the RGD-coated hybrid substrate. Cells were visualized using a charge coupled EMCCD camera (Photometrics) coupled with total internal reflection microscopy (TIRF) with 100x objective (1.5 NA, Nikon). The

chamber was maintained at 37 °C during observation. The time-lapse video was recorded at varying intervals over a range of 3 – 5 s.

The GFP-Myosin X cluster at the filopodia tip was tracked by using a cross-correlation single particle tracking method⁹¹. Obtained trajectories were analyzed using the “inpolygon” matlab function to find trajectory segments that crossed SLB circular islands and computer drawn islands in the center of the SLB pattern. The time intervals between the initial and end points of the segments were then used to estimate the average filopodia tip dwell time on the SLB and computer drawn circular islands. In addition, the ratio between the number of filopodia tip trajectories remaining inside rigid and fluid circles relatively to the total number of trajectories in the circles during the period of observation was calculated to characterize the filopodia adhesion preferences to rigid and fluid substrates (fig. S9). In this analysis, we used only trajectories that spanned more than 5 frames.

Statistics and reproducibility

Prism (GraphPad 6.0 Software) was used for statistical analysis. Each exact n value is indicated in the corresponding figure or figure legend. The significance of the differences (*P* value) was calculated using the two-tailed unpaired Student's *t*-test.

Figure 1

Fig. 1. Dynamics of pulling-induced filopodia growth

(A) Experimental setup used to observe force-induced filopodia growth. Optical tweezers were used to trap fibronectin-coated microbeads attached to filopodia tips of HeLa-JW cells. (B) Confocal images of a typical cell expressing GFP-myosin X and tdTomato-F-actin with an attached bead, taken immediately after starting of stage movement (top) and in the course of sustained growth (bottom). Note that both myosin X and actin remain at the filopodium tip during growth. See also movies S2-3. Scale bar, 5 μm . (C) Top panel: A kymograph showing the dynamics of myosin X and actin in the filopodium shown in (B). This kymograph is composed of two parts following the change of the filopodium direction at 256 seconds from the beginning of observation. Middle panel: Filopodium growth in relation to the coordinate system of the microscope stage. The origin of the coordinate system corresponds to the bead position in the center of the laser trap at the initial time point. The coordinate of the bead is changing due to the uniform movement of the stage, and fluctuations of the bead position inside the trap. Bottom panel: Forces experienced by the bead. Note the discrete peak force values corresponding to the moments of filopodia growth

cessation (seen in the middle panel) as marked with dotted lines. Inset: The distribution of peak force values, based on the pooled measurements of 21 peaks from 6 beads.

Figure 2

Fig. 2. Localization of myosin II and myosin X constructs in filopodia of HeLa-JW and Cos-7 cells, and their effects on filopodia lifetime

(A) Visualization of myosin X (mApple-myosin X, shown in red), myosin II (RLC-GFP, green) and actin (mTagBFP-Lifeact, blue) in HeLa-JW cell. **(B)** Zoomed images of bipolar myosin IIA filaments at the bases of filopodia. Top panel: myosin IIA and myosin X labeled as indicated in (A). Bottom panel: Cos-7 cell expressing mApple-myosin X (red) and GFP-myosin IIA heavy chain (green). Arrows indicate the doublets of fluorescent spots corresponding to heads of myosin II mini-filaments. **(C-H)** Images of Cos-7 cells expressing various constructs of myosin X (red) and myosin II (green): **(C)** GFP-myosin X, **(D)** mApple-myosin X and GFP-myosin IIA heavy chain, **(E)** GFP-myosin X Δ FERM, **(F)** GFP-myosin X Δ FERM and mCherry-myosin IIA heavy chain, **(G)** mApple-myosin X and GFP-myosin IIB heavy chain, **(H)** mApple-myosin X and GFP-myosin IIA N93K. Note that both myosin IIA wild type- and N93K-containing filaments, but not myosin IIB, can be seen at the bases of filopodia. See also movies S8A-G, which correspond to images A, C-H, respectively. Scale bars, 2 μ m. **(I)** Dynamics of survival fractions of the filopodia cohorts in Cos-7 cells transfected with the indicated constructs. **(J)** Lifetimes of filopodia calculated from the graphs shown in (I) by fitting them to exponential decay function: survival fraction = $e^{-t/\lambda}$, where t is time and λ is the mean filopodia lifetime. Standard deviations (SD) of λ were estimated as fitting errors according to Levenberg-Marquardt algorithm (“Origin” software package). Calculated lifetime values (seconds) indicated in the graph (from left to right) were (mean \pm SD): 151.0 \pm 5.1 (n=72, 3 cells), 537.2 \pm 19.8 (n=30, 3 cells), 152.4 \pm 1.7 (n=81, 3 cells), 513.0 \pm 16.9 (n=29, 3 cells), 194.7 \pm 2.6 (n=53, 3 cells), and 229.8 \pm 4.7 (n=70, 4 cells). All images

and data for analysis were collected using structural illumination microscopy (SIM) except (C), which was obtained by spinning disk confocal microscopy (SDCM).

Figure 3

Fig. 3. Inhibition of myosin II or formin reduces filopodia adhesion

(A) Filopodium growth upon application of pulling force in HeLa-JW cells. The deflection of the bead from its initial position at the center of the laser trap (dashed line) is proportional to the forces exerted by the filopodium (see Fig. 1, movie S9A). At 16:50 min the filopodium retracted and pulled the bead out of the trap. (B-D) Filopodia in cells with suppressed myosin II or formin activity cannot maintain sustained adhesion to the bead and do not produce forces sufficient for noticeable bead deflection during the stage movement. Cells treated with 20 μ M of S-nitro-blebbistatin for 10-20 min (B), transfected with myosin IIA siRNA (C), or treated for with 40 μ M of the formin inhibitor SMIFH2 for 1 hour (D) are shown. GFP-myosin X and tdTomato-Ftractin are shown in green and red, respectively. See also movies S9B-D. **The adhesions of the beads to filopodia were broken at 6 min 45 s, 3 min 50s**

and 2 min after starting the stage movement for S-nitro-blebbistatin-treated, myosin IIA knocked down, and SMIFH2-treated cells, respectively. Scale bars, 2 μ m. (E)

Peak values of the forces exerted by filopodia on the beads during the stage movement in control cells (no treatment) and in cells transfected with myosin IIA siRNA, or treated with S-nitro-blebbistatin, Y27632 (30 μ M, 10-20 min), or SMIFH2. Mean values (horizontal lines) and SEMs (error bars) are indicated. The mean \pm SEM of the maximal forces exerted by control filopodia (5.1 \pm 0.4pN, n = 13) was significantly higher than those in myosin IIA knockdown, as well as S-nitro-blebbistatin-, Y27632-, and SMIFH2-treated cells (1.2 \pm 0.3, n = 16; 1.0 \pm 0.2, n = 15; 0.4 \pm 0.2, n = 22; and 1.5 \pm 0.6pN, n = 14, respectively) (p<0.0001 for control vs all treatment cases).

Figure 4

Fig. 4. Effect of formin inhibition on filopodia growth and centripetal movement of myosin X patches

(A) The average length of unconstrained filopodia (left) in control HeLa-JW cells expressing GFP-myosin X (mean±SEM) was $4.1 \pm 0.1 \mu\text{m}$ ($n = 1710$ in 34 cells), which exceeded that of SMIFH2 treated cells $3.2 \pm 0.1 \mu\text{m}$ ($n = 1645$ in 31 cells), while the numbers of filopodia (right) per micron of cell boundary did not differ significantly (mean±SEM): 0.36 ± 0.01 ($n = 34$ cells) and 0.39 ± 0.02 ($n = 31$ cells), respectively. The mean values are indicated by horizontal black lines; the error bars correspond to SEMs. (B) Top panel: Disintegration of the myosin X comet tail following a 2 hours exposure to $20 \mu\text{M}$ SMIFH2. Numerous myosin X patches are seen in the filopodia shaft. Bottom panel: A kymograph showing fast centripetal movement of the patches boxed in the top panel towards the cell body (red arrowheads, see also movie S11).

Intervals of slow centripetal movements are indicated by yellow arrowheads. (C)
Addition of Y27632 treatment stops the movement of myosin X patches in SMIFH2 treated cells. The same filopodium is shown before SMIFH2 treatment (top panel), 15 min after the addition of 20 μ M SMIFH2 (middle panel) and 15 min after subsequent addition of 30 μ M Y27632 (bottom panel). Myosin X patches are shown in the left images (see also movies S12A-C), and kymographs representing the movement of the patches in the boxed area - in the images on the right. All images and data for analysis were obtained with SDCM. Scale bars, 5 μ m.

Figure 5

Fig. 5. Inhibition of myosin II or formins interfere with growth of unconstrained filopodia

A graph showing the distribution of growth/retraction velocities of unconstrained HeLa-JW cells filopodia for control, myosin II siRNA knockdown, S-nitro-blebbistatin, Y27632 and SMIFH2 treatment, observed in the same experiments as those assessing the fibronectin-coated bead attachment to filopodia. *n* represents the number of processed filopodia (with number of cells in parenthesis). (Inset) A cell expressing GFP-myosin X and tdTomato-F-actin, which is representative of those used in experiments assessing filopodia growth. The filopodium attached to the laser

trapped fibronectin-coated bead is indicated by the red arrowhead. Such filopodia were excluded from the score. Scale bar, 5 μ m.

REFERENCES

1. Jacquemet, G., Hamidi, H. & Ivaska, J. Filopodia in cell adhesion, 3D migration and cancer cell invasion. *Curr Opin Cell Biol* **36**, 23-31 (2015).
2. Mattila, P.K. & Lappalainen, P. Filopodia: molecular architecture and cellular functions. *Nat Rev Mol Cell Biol* **9**, 446-454 (2008).
3. Heckman, C.A. & Plummer, H.K., 3rd Filopodia as sensors. *Cell Signal* **25**, 2298-2311 (2013).
4. Bryant, P.J. Filopodia: fickle fingers of cell fate? *Curr Biol* **9**, R655-657 (1999).
5. Prols, F., Sagar & Scaal, M. Signaling filopodia in vertebrate embryonic development. *Cell Mol Life Sci* **73**, 961-974 (2016).
6. Romero, S. *et al.* Filopodium retraction is controlled by adhesion to its tip. *J Cell Sci* **125**, 4999-5004 (2012).
7. Vonna, L., Wiedemann, A., Aepfelbacher, M. & Sackmann, E. Micromechanics of filopodia mediated capture of pathogens by macrophages. *Eur Biophys J* **36**, 145-151 (2007).
8. Moller, J., Luhmann, T., Chabria, M., Hall, H. & Vogel, V. Macrophages lift off surface-bound bacteria using a filopodium-lamellipodium hook-and-shovel mechanism. *Sci Rep* **3**, 2884 (2013).
9. Bornschlogl, T. *et al.* Filopodial retraction force is generated by cortical actin dynamics and controlled by reversible tethering at the tip. *Proc Natl Acad Sci U S A* **110**, 18928-18933 (2013).
10. Faix, J., Breitsprecher, D., Stradal, T.E. & Rottner, K. Filopodia: Complex models for simple rods. *Int J Biochem Cell Biol* **41**, 1656-1664 (2009).
11. Jaiswal, R. *et al.* The formin Daam1 and fascin directly collaborate to promote filopodia formation. *Curr Biol* **23**, 1373-1379 (2013).
12. Vignjevic, D. *et al.* Role of fascin in filopodial protrusion. *J Cell Biol* **174**, 863-875 (2006).
13. Delanote, V. *et al.* An alpaca single-domain antibody blocks filopodia formation by obstructing L-plastin-mediated F-actin bundling. *FASEB J* **24**, 105-118 (2010).
14. Van Audenhove, I. *et al.* Fascin Rigidity and L-plastin Flexibility Cooperate in Cancer Cell Invadopodia and Filopodia. *J Biol Chem* **291**, 9148-9160 (2016).
15. Barzik, M., McClain, L.M., Gupton, S.L. & Gertler, F.B. Ena/VASP regulates mDia2-initiated filopodial length, dynamics, and function. *Mol Biol Cell* **25**, 2604-2619 (2014).
16. Block, J. *et al.* Filopodia formation induced by active mDia2/Drf3. *J Microsc* **231**, 506-517 (2008).
17. Pellegrin, S. & Mellor, H. The Rho family GTPase Rif induces filopodia through mDia2. *Curr Biol* **15**, 129-133 (2005).
18. Schirenbeck, A., Bretschneider, T., Arasada, R., Schleicher, M. & Faix, J. The Diaphanous-related formin dDia2 is required for the formation and maintenance of filopodia. *Nat Cell Biol* **7**, 619-625 (2005).
19. Block, J. *et al.* FMNL2 drives actin-based protrusion and migration downstream of Cdc42. *Curr Biol* **22**, 1005-1012 (2012).
20. Harris, E.S., Gauvin, T.J., Heimsath, E.G. & Higgs, H.N. Assembly of filopodia by the formin FRL2 (FMNL3). *Cytoskeleton (Hoboken)* **67**, 755-772 (2010).
21. Young, L.E., Heimsath, E.G. & Higgs, H.N. Cell type-dependent mechanisms for formin-mediated assembly of filopodia. *Mol Biol Cell* **26**, 4646-4659 (2015).
22. Applewhite, D.A. *et al.* Ena/VASP proteins have an anti-capping independent function in filopodia formation. *Mol Biol Cell* **18**, 2579-2591 (2007).
23. Schirenbeck, A., Arasada, R., Bretschneider, T., Schleicher, M. & Faix, J. Formins and VASPs may co-operate in the formation of filopodia. *Biochem Soc Trans* **33**, 1256-1259 (2005).
24. Bear, J.E. & Gertler, F.B. Ena/VASP: towards resolving a pointed controversy at the barbed end. *J Cell Sci* **122**, 1947-1953 (2009).
25. Zhang, H. *et al.* Myosin-X provides a motor-based link between integrins and the cytoskeleton. *Nat Cell Biol* **6**, 523-531 (2004).
26. Kerber, M.L. & Cheney, R.E. Myosin-X: a MyTH-FERM myosin at the tips of filopodia. *J Cell Sci* **124**, 3733-3741 (2011).
27. Sousa, A.D. & Cheney, R.E. Myosin-X: a molecular motor at the cell's fingertips. *Trends Cell Biol* **15**, 533-539 (2005).
28. Hoffmann, B. & Schafer, C. Filopodial focal complexes direct adhesion and force generation towards filopodia outgrowth. *Cell Adh Migr* **4**, 190-193 (2010).

29. Sydor, A.M., Su, A.L., Wang, F.S., Xu, A. & Jay, D.G. Talin and vinculin play distinct roles in filopodial motility in the neuronal growth cone. *J Cell Biol* **134**, 1197-1207 (1996).
30. Lagarrigue, F. *et al.* A RIAM/lamellipodin-talin-integrin complex forms the tip of sticky fingers that guide cell migration. *Nat Commun* **6**, 8492 (2015).
31. Schafer, C. *et al.* One step ahead: role of filopodia in adhesion formation during cell migration of keratinocytes. *Exp Cell Res* **315**, 1212-1224 (2009).
32. Schafer, C. *et al.* The key feature for early migratory processes: Dependence of adhesion, actin bundles, force generation and transmission on filopodia. *Cell Adh Migr* **4**, 215-225 (2010).
33. Geiger, B., Spatz, J.P. & Bershadsky, A.D. Environmental sensing through focal adhesions. *Nat Rev Mol Cell Biol* **10**, 21-33 (2009).
34. Sun, Z., Guo, S.S. & Fassler, R. Integrin-mediated mechanotransduction. *J Cell Biol* **215**, 445-456 (2016).
35. Rivelino, D. *et al.* Focal contacts as mechanosensors: externally applied local mechanical force induces growth of focal contacts by an mDial1-dependent and ROCK-independent mechanism. *J Cell Biol* **153**, 1175-1186 (2001).
36. Goffin, J.M. *et al.* Focal adhesion size controls tension-dependent recruitment of alpha-smooth muscle actin to stress fibers. *J Cell Biol* **172**, 259-268 (2006).
37. Prager-Khoutorsky, M. *et al.* Fibroblast polarization is a matrix-rigidity-dependent process controlled by focal adhesion mechanosensing. *Nat Cell Biol* **13**, 1457-1465 (2011).
38. Trichet, L. *et al.* Evidence of a large-scale mechanosensing mechanism for cellular adaptation to substrate stiffness. *Proc Natl Acad Sci U S A* **109**, 6933-6938 (2012).
39. Wong, S., Guo, W.H. & Wang, Y.L. Fibroblasts probe substrate rigidity with filopodia extensions before occupying an area. *Proc Natl Acad Sci U S A* **111**, 17176-17181 (2014).
40. Berg, J.S. & Cheney, R.E. Myosin-X is an unconventional myosin that undergoes intrafilopodial motility. *Nat Cell Biol* **4**, 246-250 (2002).
41. Watanabe, T.M., Tokuo, H., Gonda, K., Higuchi, H. & Ikebe, M. Myosin-X induces filopodia by multiple elongation mechanism. *J Biol Chem* **285**, 19605-19614 (2010).
42. Tullio, A.N. *et al.* Nonmuscle myosin II-B is required for normal development of the mouse heart. *Proc Natl Acad Sci U S A* **94**, 12407-12412 (1997).
43. Even-Ram, S. *et al.* Myosin IIA regulates cell motility and actomyosin-microtubule crosstalk. *Nat Cell Biol* **9**, 299-309 (2007).
44. Bohil, A.B., Robertson, B.W. & Cheney, R.E. Myosin-X is a molecular motor that functions in filopodia formation. *Proc Natl Acad Sci U S A* **103**, 12411-12416 (2006).
45. Choi, C.K. *et al.* Actin and alpha-actinin orchestrate the assembly and maturation of nascent adhesions in a myosin II motor-independent manner. *Nat Cell Biol* **10**, 1039-1050 (2008).
46. Vicente-Manzanares, M., Zareno, J., Whitmore, L., Choi, C.K. & Horwitz, A.F. Regulation of protrusion, adhesion dynamics, and polarity by myosins IIA and IIB in migrating cells. *J Cell Biol* **176**, 573-580 (2007).
47. Smith, R.C. *et al.* Regulation of myosin filament assembly by light-chain phosphorylation. *Philos Trans R Soc Lond B Biol Sci* **302**, 73-82 (1983).
48. Betapudi, V. Life without double-headed non-muscle myosin II motor proteins. *Front Chem* **2**, 45 (2014).
49. Vicente-Manzanares, M., Ma, X., Adelstein, R.S. & Horwitz, A.R. Non-muscle myosin II takes centre stage in cell adhesion and migration. *Nat Rev Mol Cell Biol* **10**, 778-790 (2009).
50. Hu, S. *et al.* Long-range self-organization of cytoskeletal myosin II filament stacks. *Nat Cell Biol* **19**, 133-141 (2017).
51. Rizvi, S.A. *et al.* Identification and characterization of a small molecule inhibitor of formin-mediated actin assembly. *Chem Biol* **16**, 1158-1168 (2009).
52. Tokuo, H. & Ikebe, M. Myosin X transports Mena/VASP to the tip of filopodia. *Biochem Biophys Res Commun* **319**, 214-220 (2004).
53. Yu, C.H., Law, J.B., Suryana, M., Low, H.Y. & Sheetz, M.P. Early integrin binding to Arg-Gly-Asp peptide activates actin polymerization and. *Proc Natl Acad Sci U S A* **108**, 20585-20590 (2011).
54. Jacquemet, G. *et al.* L-type calcium channels regulate filopodia stability and cancer cell invasion downstream of integrin signalling. *Nat Commun* **7**, 13297 (2016).
55. Billington, N., Wang, A., Mao, J., Adelstein, R.S. & Sellers, J.R. Characterization of three full-length human nonmuscle myosin II paralogs. *J Biol Chem* **288**, 33398-33410 (2013).
56. Hundt, N., Steffen, W., Pathan-Chhatbar, S., Taft, M.H. & Manstein, D.J. Load-dependent modulation of non-muscle myosin-2A function by tropomyosin 4.2. *Sci Rep* **6**, 20554 (2016).

57. Kovacs, M., Wang, F., Hu, A., Zhang, Y. & Sellers, J.R. Functional divergence of human cytoplasmic myosin II: kinetic characterization of the non-muscle IIA isoform. *J Biol Chem* **278**, 38132-38140 (2003).
58. Yan, J., Yao, M., Goult, B.T. & Sheetz, M.P. Talin Dependent Mechanosensitivity of Cell Focal Adhesions. *Cell Mol Bioeng* **8**, 151-159 (2015).
59. Atherton, P. *et al.* Vinculin controls talin engagement with the actomyosin machinery. *Nat Commun* **6**, 10038 (2015).
60. Hu, X. *et al.* Cooperative Vinculin Binding to Talin Mapped by Time-Resolved Super Resolution Microscopy. *Nano Lett* **16**, 4062-4068 (2016).
61. Gauthier, N.C. & Roca-Cusachs, P. Mechanosensing at integrin-mediated cell-matrix adhesions: from molecular to integrated mechanisms. *Curr Opin Cell Biol* **50**, 20-26 (2018).
62. Hu, W., Wehrle-Haller, B. & Vogel, V. Maturation of filopodia shaft adhesions is upregulated by local cycles of. *PLoS One* **9**, 0107097 (2014).
63. Goult, B.T. *et al.* RIAM and vinculin binding to talin are mutually exclusive and regulate adhesion. *J Biol Chem* **288**, 8238-8249 (2013).
64. Lafuente, E.M. *et al.* RIAM, an Ena/VASP and Profilin ligand, interacts with Rap1-GTP and mediates Rap1-induced adhesion. *Dev Cell* **7**, 585-595 (2004).
65. Courtemanche, N., Lee, J.Y., Pollard, T.D. & Greene, E.C. Tension modulates actin filament polymerization mediated by formin and profilin. *Proc Natl Acad Sci U S A* **110**, 9752-9757 (2013).
66. Jegou, A., Carlier, M.F. & Romet-Lemonne, G. Formin mDia1 senses and generates mechanical forces on actin filaments. *Nat Commun* **4**, 1883 (2013).
67. Kozlov, M.M. & Bershadsky, A.D. Processive capping by formin suggests a force-driven mechanism of actin polymerization. *J Cell Biol* **167**, 1011-1017 (2004).
68. Yu, M. *et al.* mDia1 senses both force and torque during F-actin filament polymerization. *Nat Commun* **8**, 1650 (2017).
69. Galbraith, C.G., Yamada, K.M. & Galbraith, J.A. Polymerizing actin fibers position integrins primed to probe for adhesion sites. *Science* **315**, 992-995 (2007).
70. Guillou, H. *et al.* Lamellipodia nucleation by filopodia depends on integrin occupancy and downstream Rac1 signaling. *Exp Cell Res* **314**, 478-488 (2008).
71. Chan, C.E. & Odde, D.J. Traction dynamics of filopodia on compliant substrates. *Science* **322**, 1687-1691 (2008).
72. Lo, C.M., Wang, H.B., Dembo, M. & Wang, Y.L. Cell movement is guided by the rigidity of the substrate. *Biophys J* **79**, 144-152 (2000).
73. Heidemann, S.R. & Bray, D. Tension-driven axon assembly: a possible mechanism. *Front Cell Neurosci* **9**, 316 (2015).
74. Kerstein, P.C., Nichol, R.I. & Gomez, T.M. Mechanochemical regulation of growth cone motility. *Front Cell Neurosci* **9**, 244 (2015).
75. Bridgman, P.C., Dave, S., Asnes, C.F., Tullio, A.N. & Adelstein, R.S. Myosin IIB is required for growth cone motility. *J Neurosci* **21**, 6159-6169 (2001).
76. Bray, D. Mechanical tension produced by nerve cells in tissue culture. *J Cell Sci* **37**, 391-410 (1979).
77. Bai, M., Harfe, B. & Freimuth, P. Mutations that alter an Arg-Gly-Asp (RGD) sequence in the adenovirus type 2 penton base protein abolish its cell-rounding activity and delay virus reproduction in flat cells. *J Virol* **67**, 5198-5205 (1993).
78. Paran, Y. *et al.* Development and application of automatic high-resolution light microscopy for cell-based screens. *Methods Enzymol* **414**, 228-247 (2006).
79. Schell, M.J., Erneux, C. & Irvine, R.F. Inositol 1,4,5-trisphosphate 3-kinase A associates with F-actin and dendritic spines via its N terminus. *J Biol Chem* **276**, 37537-37546 (2001).
80. Lin, A.Y., Prochniewicz, E., James, Z.M., Svensson, B. & Thomas, D.D. Large-scale opening of utrophin's tandem calponin homology (CH) domains upon actin binding by an induced-fit mechanism. *Proc Natl Acad Sci U S A* **108**, 12729-12733 (2011).
81. Winder, S.J. *et al.* Calmodulin regulation of utrophin actin binding. *Biochem Soc Trans* **23**, 397S (1995).
82. Kengyel, A., Wolf, W.A., Chisholm, R.L. & Sellers, J.R. Nonmuscle myosin IIA with a GFP fused to the N-terminus of the regulatory light chain is regulated normally. *J Muscle Res Cell Motil* **31**, 163-170 (2010).
83. Watanabe, S. *et al.* mDia2 induces the actin scaffold for the contractile ring and stabilizes its position during cytokinesis in NIH 3T3 cells. *Mol Biol Cell* **19**, 2328-2338 (2008).

84. Brock, R. & Jovin, T.M. Heterogeneity of signal transduction at the subcellular level: microsphere-based focal EGF receptor activation and stimulation of Shc translocation. *J Cell Sci* **114**, 2437-2447 (2001).
85. Neuman, K.C. & Block, S.M. Optical trapping. *Rev Sci Instrum* **75**, 2787-2809 (2004).
86. Toth, M.A. *et al.* Biochemical Activities of the Wiskott-Aldrich Syndrome Homology Region 2 Domains of Sarcomere Length Short (SALS) Protein. *J Biol Chem* **291**, 667-680 (2016).
87. Spudich, J.A. & Watt, S. The regulation of rabbit skeletal muscle contraction. I. Biochemical studies of the interaction of the tropomyosin-troponin complex with actin and the proteolytic fragments of myosin. *J Biol Chem* **246**, 4866-4871 (1971).
88. Romero, S. *et al.* Formin is a processive motor that requires profilin to accelerate actin assembly and associated ATP hydrolysis. *Cell* **119**, 419-429 (2004).
89. Tsygankov, D. *et al.* CellGeo: a computational platform for the analysis of shape changes in cells with complex geometries. *J Cell Biol* **204**, 443-460 (2014).
90. Lin, W.C., Yu, C.H., Triffo, S. & Groves, J.T. Supported membrane formation, characterization, functionalization, and patterning. *Curr Protoc Chem Biol* **2**, 235-269 (2010).
91. Gelles, J., Schnapp, B.J. & Sheetz, M.P. Tracking kinesin-driven movements with nanometre-scale precision. *Nature* **331**, 450-453 (1988).

Acknowledgments: Encouraging and stimulating discussions with Drs. D. Bray (University of Cambridge, UK), M.M. Kozlov (Tel Aviv University, Israel) are much appreciated. We are grateful to Dr. T. Kachanawong (MBI, Singapore) for providing genetic constructs, Dr. D. Kovar (University of Chicago, IL, USA) for a sample of SMIFH2 inhibitor, and Dr. Tsygankov (Georgia Tech, USA) for providing code for filopodia length computation. We thank the Protein Cloning and Expression Core facility of the MBI for help with sub-cloning of mCherry-mDia2 and mApple-myosin X. We also thank Dr. F. Margadant and Lau Wai Han (MBI Microscopy Core facility) and Dr. V. Vyasoff (MBI, Singapore) for their kind help with the optical tweezers setup. This research has been supported by the National Research Foundation Singapore, Ministry of Education of Singapore, Grant R714006006271 & R714019006271 (awarded to A.D.B.), Grant MOE2012T31001 (awarded to Y.J.) and BMRC Grant A*Star-JST 1514324022 (awarded to A.D.B.). As well we thank S. Wolf and A. Wang (Science Communication MBI, Singapore) for excellent editorial help.

Supplementary Materials for

Force dependence of filopodia adhesion: involvement of myosin II and formins

N.O. Alieva^{1†}, A.K. Efremov^{1,2†}, S. Hu¹, D. Oh¹, Z. Chen¹, M. Natarajan¹,
H.T. Ong¹, A. Jégou³, G. Romet-Lemonne³, J.T. Groves^{1,4}, M.P. Sheetz^{1,5}, J. Yan^{1,2,6},

A.D. Bershadsky^{1,7*}

correspondence to: alexander.bershadsky@weizmann.ac.il

This PDF file includes:

Figs. S1 to S9

Captions for Movies S1 to S14

Other Supplementary Materials for this manuscript includes the following:

Movies S1 to S14

Figure S1

Fig. S1

Dynamics and composition of filopodia induced by myosin X expression

(A) Left: Filopodia from HeLa-JW cells expressing GFP-myosin X. Myosin X positive “comet tails” are seen at the tips of filopodia. Right: A kymograph showing the growth of a GFP-myosin X labeled filopodium indicated by dashed box in the left image. Note that periods of fast growth alternate with periods of slow shrinking. See also movies S1. (B) Filopodia tips labeled with myosin X are enriched with mDia2, VASP and talin. Left: Images of filopodia in cells co-expressing GFP-myosin X with mCherry fusion constructs of mDia2, VASP and talin respectively. Right: Line scans of the fluorescence intensities through the filopodia indicated by dashed boxes in the left images. Intensities of the myosin X, mDia2, VASP and formin were normalized to their maximal values at the filopodia tips. Scale bars, 5 μ m. (C) Kymograph analysis of the retrograde movement of myosin X and photoactivated actin in the same filopodia. Left panel corresponds to GFP-myosin X and shows immobile tip of the filopodium (upper line) and retrograde movement of myosin X (tilted line). Right panel represents the retrograde movement of photoactivated PAmCherry-b-actin (tilted line) in the same filopodia. Note that the rate of retrograde movements of myosin X and actin are similar and were equal to 37nm/s. See also movie S7. (D) A cartoon depicting the protein composition at the tip of filopodium. Actin filaments are connected with integrin receptors via talin and polymerized by formin family. VASP proteins are located at the tip. Myosin X is interacting with actin filaments, integrins and VASP. Images show in panels (A) and (B) were obtained with SDCM; (C) was obtained with SIM.

Figure S2

Fig. S2

Length of myosin X-induced filopodia in HeLa-JW and Cos-7 cells

From left to right: Scattered dot plots representing the lengths of filopodia in HeLa-JW cells expressing GFP-myosin X untreated (control) and myosin IIA-depleted cells, as well as treated with S-nitro blebbistatin (20μM, 1 hour) and with Y27632 (30μM, 1 hour). The last two plots represent control myosin X-expressing Cos-7 cells and Cos-7 cells co-transfected with myosin X and myosin IIA. The symbols correspond to individual filopodia. The mean values are indicated by horizontal black lines; the error bars correspond to SEMs. The mean lengths of control myosin X-induced filopodia, and filopodia from myosin IIA siRNA-, S-nitro-blebbistatin-, and Y27632-treated cells, were (mean±SEM) 4.6±0.1μm (n = 922, 38 cells), 3.8±0.2μm (n = 160, 18 cells), 4.5±0.1μm (n = 299, 24 cells), 4.4±0.1μm (n = 656, 27 cells), respectively.

The mean lengths of filopodia in myosin X-transfected control Cos-7 cells and Cos-7 cells co-expressing myosin X and myosin IIA were: $3.7 \pm 0.1 \mu\text{m}$ (n = 269, 11 cells) and $4.4 \pm 0.1 \mu\text{m}$ (n = 950, 32 cells) respectively.

Figure S3

Fig. S3

siRNA knockdown of myosinIIA

(A) Immunoblots of myosin IIA in non-targeted and knockdown HeLa-JW cells. α -tubulin was used as loading control. (B) Myosin knockdown cells did not contain either myosin IIA (left bottom) or prominent actin stress fibers (right bottom), but still contained filopodia. The filopodium attached to the bead, which was stretched in one of optical trap experiments, is indicated by asterisk. Images were obtained with SDCM. Scale bar, 10 μ m.

Figure S4

Fig. S4

Immediate effects of ROCK inhibitor, Y27632, and **formin inhibitor, SMIFH2**, on force-induced filopodia growth and adhesion

Fibronectin-coated beads trapped by laser tweezers were attached to the filopodia tips of HeLa-JW cells. Filopodia growth was then induced through the generation of pulling force, which resulted from the movement of the microscope stage, as shown in Fig. 1. 30 μ M of Y27632 (A-B) or 40 μ M of SMIFH2 (C-D) were added at about 4 and 11 min following the start of the stage movement, respectively. The positions of the trapped beads during the filopodia growth are shown in (A) and (C). The intensity of actin labeling in (A) was relatively low and apparent “disappearance” of actin in the late frames was a result of photobleaching. Scale bars, 2 μ m. See also movies S10 and S13. (B and D) Dynamics of the forces exerted by the filopodia on the beads after addition of Y27632 (B) or SMIFH2 (D). Moments of drugs addition are indicated with green and purple arrows, respectively. Detachment of the bead in (B) occurred at 13 min following the drug addition (black arrow, see also movie S10). Moments of switching the trap off in (D) are indicated by red arrows. In this experiment, the bead remained associated with the filopodium via the membrane tether (see also movie S13).

Figure S5

Fig. S5

Effect of formin inhibitor SMIFH2 on myosin X induced filopodia

Image of HeLa-JW cell, labeled with GFP-myosin X and Cherry-Utrophin, that was treated with 20 μ M SMIFH2 formin inhibitor for 2 hours. Note that the majority of the filopodia do not contain myosin X. The image was obtained using SDCM. Scale bar, 10 μ m.

Figure S6

Fig. S6

Localization of VASP at myosin X patches in cells treated with formin inhibitor

Distribution of GFP-myosin X (green, left panel) and Cherry-VASP (red, central panel) in the same HeLa-JW cell 1.5h after addition of $20\mu\text{M}$ SMIFH2. Right panel: the line scans through the boxed area, showing the co-distribution of myosin X and VASP. Fluorescent intensities are normalized to their maximal values. Scale bar, $5\mu\text{m}$. Images were obtained with SDCM.

Figure S7

Fig. S7

SMIFH2 enhances the detachment of formins from actin filaments in vitro

The constitutively active mDia1 formin construct (FH1FH2DAD) was anchored to the glass surface of a microfluidic chamber by one of their FH2 domains using an anti-His antibody (as in Jegou et al. 2013, see Materials and Methods). Actin filaments were grown first in the presence of Alexa488-labelled actin to form fluorescent segments at the tips of actin filaments. Once the formin had nucleated a filament, it was exposed, from time zero onward, to 1 μM unlabeled actin and 4 μM profilin, in absence or presence of 100 μM SMIFH2. Panel (A) shows frames corresponding to the beginning (time zero) and 300 sec following addition of unlabeled actin into the flow chamber in the presence or absence of SMIFH2 inhibitor

in solution. The epifluorescence images of the labeled segments of actin filaments are seen. The number of filaments decreased with time due to their detachment from immobilized formin molecules. The time at which each filament detached from its formin was recorded and the survival fraction of the filaments at each time point was calculated. (B) SMIFH2 enhanced the filament detachment rate by an order of

magnitude without affecting the formin mediated elongation rates of the filaments (35 ± 6 vs. 32 ± 8 subunits/s, $n = 40$ filaments without and with SMIFH2, respectively).

Scale bar, $10\mu\text{m}$.

Figure S8

Fig. S8

(A) A fluorescence image of a hybrid substrate containing a RGD ligand which is tethered to PLL-PEG polymer (background) and a supported lipid bilayer (SLB) surface (circles) via a biotin- Dylight-405 NeutrAvidin conjugation. **(B)** RGD intensity profile along the white line shown in (A). The red and black lines at the top of the curve mark the SLB and PLL-PEG regions, respectively, showing a similar concentration of RGD ligands in both regions.

Figure S9

Fig. S9

Attachment of filopodia to RGD-coated rigid and fluid substrate

(A) HeLa-JW cells expressing GFP-myosin X were plated on micropatterned coverslips covered with circular islands ($D = 3\mu\text{m}$) of supported lipid bilayer (SLB) conjugated to RGD (orange circles), organized into a square lattice. The glass between the islands was covered with poly-L-lysine-PEG conjugated to RGD at the same density (fig. S8). Trajectories of GFP-positive filopodia tips acquired during a 14-36 min time interval are shown. The cell border is shown by a yellow dashed line. For comparison of the trajectories on rigid and fluid substrates, the circles of similar diameter were drawn by computer in the centers of the square lattice formed by SLB islands (outlined by gray contours). The segments of the trajectories located inside either the SLB islands or the drawn circles on the rigid substrate are shown in red, and the remaining parts of the trajectories are shown in white. See also movie S14. Scale

bar, 5 μ m. **(B-C)** Quantification of the trajectories of filopodia tips inside rigid and fluid circular islands for five cells (at least 200 individual trajectories per cell were scored). **(B)** The bars represent the average dwelling time that filopodia tips spent inside rigid (turquoise) or fluid (red) circles defined above. **(C)** Fraction of filopodia tip trajectories remaining inside rigid circles (green bar) and fluid circles (orange bar) relatively to the total number of trajectories in the circles during the period of observation. Error bars correspond to the SEM.

Movie S1

Dynamics of filopodia induced by myosin X expression

Growing, pausing and retracting filopodia in HeLa-JW cells overexpressing GFP-myosin X. The myosin X is localized to filopodia tips where it appears as characteristic “comet tails”. The duration of the entire movie is 3 min, the movie images were recorded at 4.31 frames per second (fps) and display rate is 50 fps.

Movie S2

3D view of the fixed myosin X-induced filopodium attached to a 2 μ m

fibronectin-coated bead

HeLa-JW cell was transfected with GFP-myosin X and tdTomato-Ftractin. Single filopodium was attached to optically trapped bead. After pulling of the filopodium with the optical trap for several minutes, the cell was fixed by addition of 20% PFA (Tousimis) directly to the chamber up to final concentration of 4%. Image and position of the bead were reconstructed from bright field image. Note, clearly visible integrity of actin core (red).

Movie S3

Dynamics of force-induced growth of GFP-myosin X-induced filopodia

Sustained filopodial growth induced by pulling force generated by the optically trapped microbead coated with fibronectin (see details in SI and Fig. 1A). The frames from this movie, and the kymograph based on it, are shown in Fig. 1B and 1C respectively. The duration of the entire movie is 26 min 52 s. Movie images were recorded at 0.5 fps and displayed at 25 fps.

Movie S4

Pulling of GFP-myosin X-induced filopodia using concanavalin A-coated beads

Pulling force was applied to a filopodium of HeLa-JW cell via a concanavalin-A-coated laser-trapped bead by moving the microscope stage. This force was applied to induce filopodia growth, but in contrast with the results obtained from experiments using fibronectin-coated beads (movie S3), under these conditions, filopodium did not grow and formation of membrane tethers occurred instead. In the movie, the tether formation can be inferred from the rapid return of the bead to the myosin X-containing tip of the filopodium after switching off the laser trap. Note, that the bead continued to be attached to the filopodium by thin membrane tether and stayed in the field of observation during last several seconds of the movie after the bead was released from the trap. The duration of the movie is 4 min and the movie images were recorded at 0.5 fps with a display rate of 15 fps.

Movie S5

Pulling of Cdc42 Q61L-induced filopodia

Filopodia of a HeLa-JW cell transfected with constitutively active Cdc42 mutant, GFP-Cdc42 Q61L, and tdTomato-Ftractin were pulled using the optical trap. Arrow indicates the center of the trap. Note, that Cdc42-induced filopodium attached to laser-trapped fibronectin-coated beads did not grow upon stage movement and eventually pulled the beads out of the trap. The duration of the movie is 6 min 36 s and the movie images were recorded at 1 fps with a display rate of 15 fps.

Movie S6

Centripetal co-movement of VASP and myosin X patches in cells treated with formin inhibitor

Co-localization of mApple-myosin X (green) and GFP-VASP (red) in the same filopodia of HeLa-JW cell 90 min after the addition of 20 μ M SMIFH2. The duration of the movie is 83 min 10 s and the movie images were recorded at 0.1 fps with a display rate of 50 fps. Movie was obtained by SDCM.

Movie S7

Visualization of photoactivated actin in filopodia expressing myosin X

This movie corresponds to the fig. S1C.

The left panel corresponds to GFP-myosin X. The right panel represents photoactivated PAmCherry-b-actin in the same filopodium of HeLa-JW cell. The site of photoactivation is indicated by the pink line and was performed at 6 s from the start of the movie. Note the retrograde movements of myosin X and actin inside the filopodium. The kymograph of the line drawn along the length of the filopodium, in which the photoactivation assay was performed, is shown in fig. S1C. The duration of the movie is 1 min 33 s and the movie images were recorded at 0.6 fps with a display rate of 7 fps.

Movie S8A-G

Myosin II filaments in filopodia of HeLa-JW and Cos-7 cells

These movies correspond to the Fig. 2A, C-H. (A) mApple-myosin X (red), RLC-GFP (green) and mTagBFP-Lifeact (blue) in HeLa-JW cell. (B-G) Cos-7 cells expressing myosin X (red) and myosin IIA or B (green) and their mutants: (B) GFP-myosin X. (C) mApple-myosin X and GFP-myosin IIA heavy chain. (D) GFP-myosin

X Δ FERM. (E) GFP-myosin X Δ FERM and mCherry-myosin IIA heavy chain. (F) mApple-myosin X and GFP-myosin IIB heavy chain. (G) mApple-myosin X and GFP-myosin IIA N93K. Note the presence of myosin IIA filaments in movies A, C, E and G and the absence of myosin IIB in movie F at the filopodia bases. The duration of the movies were 15 min 50 s, 4 min 8s, 4 min 5 s, 4 min 5 s, 4 min 5s, 3 min 15 s and 4 min 5s respectively, the movie images were recorded and displayed at 0.003 and 7 for (A), 0.5 and 18 for (B), 0.2 and 7.5 fps for (C-G).

Movies S9A-D

Inhibition of myosin II or formin suppresses filopodia adhesion and growth (see also Fig. 3 A-D)

(A) Movie showing sustained growth of a filopodium induced by pulling force in a control HeLa-JW cell transfected with GFP-myosin X construct. The experiment was analogous to that described in the legend to Fig. 1A. The cell was used as a control for experiments with inhibitors presented in Fig. 3. The duration of the movie is 17 min 8 s. (B) Effect of light-insensitive blebbistatin. The cell was pretreated with 20 μ M S-nitro-blebbistatin for 10-20 min prior to placing the laser trapped fibronectin-coated bead onto the filopodium tip. 30 s later, stage movement commenced simultaneously with filming. Note that adhesion of the bead to filopodium was broken 6 min 45 s after starting the stage movement. The duration of the movie is 13 min 12 s. (C) Effect of myosin IIA knockdown. The detachment of the filopodium from the bead occurred 3 min 50 s after the stage movement was initiated. **The velocity of the retrograde movement of the myosin X patch in this experiment was about 10nm/s, significantly lower than the velocity of myosin X patches driven by myosin II (Fig. 4).** The duration of the movie is 4 min 38 s. (D) Effect of formin inhibition by

SMIFH2. The cell was pretreated with 40 μ M SMIFH2 for 1 hour prior the bead being placed onto the filopodium tip. To check whether the bead had attached to the filopodium, the laser trap was switched off for several seconds, during which time unattached beads typically disappeared from the field of view. The detachment of the filopodium from the bead occurred about 2 min after the stage movement was initiated. The duration of the entire movie is 4 min 56 s. All the movies in this figure were recorded at 0.5 fps with a display rate of 15 fps.

Movie S10

Immediate effect of ROCK inhibitor Y27632 on force-induced filopodia growth and adhesion

This movie corresponds to the frames shown in fig. S4A. HeLa-JW cell filopodium growth was induced by applying a pulling force, generated as a result of microscope stage movement, as in Fig. 1. At about 4 min after stage movement commenced, 50 μ M of Y27632 was added, which eventually resulted in bead detachment at 17 min.

The intensity of actin labeling was relatively low and apparent “disappearance” of actin in the second half of the movie was a result of photobleaching. The duration of the entire movie is 19 min 22 s. The movie was recorded at 1 fps and displayed at 25 fps.

Movie S11

Effect of formin inhibition on dynamics of myosin X in filopodia

This movie corresponds to the Fig. 4B. The filopodia were induced by HeLa-JW cell transfection with a construct encoding GFP-myosin X. Before imaging, the cell was treated with 20 μ M of SMIFH2 for 2 hours. Note the apparent disintegration of the

myosin X containing patch at the tip of the filopodium into numerous small patches, the majority of which rapidly moved centripetally from the filopodium tip to the cell body, with retrograde movement of myosin X patches sometimes interrupted by short periods of anterograde sliding towards the filopodium tip. The duration of the entire movie is 5 min; the movie was recorded at 4 fps and displayed at 100 fps. The movie was obtained with SDCM.

Movie S12A-C

Effect of myosin II inhibition on fast centripetal movement of myosin X patches in filopodia of SMIFH2 treated cells

These movies correspond to Fig.4C. A HeLa-JW cell transfected with GFP-myosin X was filmed before treatment (A), 15 min after the addition of 20 μ M SMIFH2 (B) and 20 min after the subsequent addition of 30 μ M Y27632 (C). Note that after SMIFH2 was added, myosin X comet tails underwent rapid disintegration into small patches, which moved centripetally towards the cell body (B). Addition of Y27632 resulted in cessation of this movement (C). The duration of the movies A-C were 6 min 41 s; the movie images were recorded at 0.1 fps and displayed at 7 fps. The movies were obtained with SDCM.

Movie S13

Immediate effect of formin inhibitor SMIFH2 on force-induced filopodia growth and adhesion

40 μ M of SMIFH2 was added to HeLa-JW cell expressing GFP-myosin X (green) and tdTomato-Fractin (red) after establishment of sustained filopodia growth (at 11th min). About 3 minutes later, cessation of growth of the actin core and the drop in the

pulling force generated by the filopodium were observed but the fibronectin-coated bead remained associated with the filopodium tip via a membrane tether. The duration of the entire movie is 20 min 39 s. The movie was recorded at 1 fps and displayed at 30 fps. (A sequence from this movie is shown also in fig. S4).

Movie S14

Filopodia distinguish between fluid and rigid substrates

GFP-myosin X transfected HeLa-JW cell spreading on the micropatterned substrate with 3 μ m islands covered with supported lipid bilayer (SLB). Both the islands and the rigid substrate between them were coated with fluorescent Dylight-405 RGD ligand shown in blue. The GFP-myosin X positive filopodia tips are shown in green. The movie started 30 min following the cell plating. Note that filopodia tips apparently avoid the SLB islands, see the analysis in fig. S9. The duration of the movies was 16 min 40 s; the movie images were recorded at 0.3 fps and displayed at 30 fps.

Reviewers' comments:

Reviewer #1 (Remarks to the Author):

In this study, authors show that under external force, myosin X-induced filopodia maintain adhesion to fibronectin-coated beads while their actin core keeps growing. These data indicate that the integrin-mediated adhesion of filopodia is mechanosensitive. Furthermore, authors show that these effects require myosin IIA and formin activity, suggesting that myosin IIA contributes endogenous force to strengthen adhesions, whereas formins function by attaching filopodial actin filaments to the adhesion site at the filopodial tip. These findings present an important contribution to our understanding of cell motility and adhesion. They complement and extend the existing knowledge that other cell adhesion types (focal adhesions, cell-cell adhesions, and lamellipodial adhesions) are also mechanosensitive.

The previous version of the manuscript suffered from multiple weaknesses in terms of data presentation, both technical and conceptual. In the revised manuscript, authors clarified some, but not all of these confusing aspects of their study.

Major points:

1. In the response letter, authors limit the scope of their study specifically to myosin X-induced filopodia, because Cdc42- and mDia-induced filopodia in their hands behaved differently. This decision is acceptable. However, because of this observation, they need to discuss (i) how general their findings are considering that other types of filopodia are expected to have endogenous myosin X at the tips, and (ii) what could be special about overexpressed myosin X that enables different behavior of filopodia.

2. One of the main conclusions of the study is that myosin IIA functions locally from the base of filopodia. Although this mechanism seems most natural, I do not find that it is sufficiently supported by the presented data. I am not satisfied with how quantification of myosin IIA localization to filopodia bases is done. First, only one movie was quantified. Second, authors quantified a fraction of filopodia that was ever visited by myosin IIA, which could give an impression of a greater presence of myosin II at filopodial bases than it really was. However, even this quantification gave only 70% of myosin IIA-positive filopodia. The authors need to (i) determine a fraction of time during which each filopodium had myosin II at the base; (ii) describe how the base was defined; (iii) use non-filopodial cell edges as controls; and (iv) provide statistics for these data. Similar quantification should be done for myosin IIB. Even in the absence of proper quantification, the provided data suggest that myosin IIA is only transiently present, not abundant and not ubiquitous at the filopodial bases. Therefore, the outcomes of myosin II inhibition could reflect global effects, such as changes in overall cell contractility, membrane tension, etc., rather than local force generated at the filopodial bases. This possibilities could be discriminated by correlating the force peaks with the presence of myosin II at the filopodial base, but authors declined doing this experiment, in which case they need to significantly tone down their claims about specific mechanism by which myosin II functions here.

3. I am still confused by the description of how forces and bead detachments were determined. In response to my previous comment, the authors provided the following description in Methods: "The moment of detachment of filopodia from the bead was detected by return of the bead to the center of optical trap". In Results, they say: "in the cells with impaired myosin II activity the filopodia detached earlier" (p. 9), suggesting that the bead returned to the trap sooner than in control. However, the figure 3B-D legend says: "Filopodia in cells with suppressed myosin II or formin activity ... do not produce forces sufficient for noticeable bead deflection during the stage movement", which is indeed obvious from the figure. How did they detect detachment in myosin-impaired cells if the bead did not

move out of the trap? How did they determine forces presented in Figure 3E without bead movement? In control cells, the bead periodically returns to the trap after high force episodes (Figure 1). Apparently, such cases were not considered to be detachments. Then, how were actual detachments discriminated objectively from the bead returns after high force episodes?

4. When discussing myosin II and formin inhibition, authors state that the duration of the contacts was “shorter”. However, the contact duration remains unquantified despite my request in the previous review. Instead, the authors responded that they showed specific detachment times for single examples in a couple of figures, which cannot be considered as quantification. They need a sufficient sample size and statistics to show the stated differences.

5. If authors agree that the filopodial length is not a useful parameter, why not to delete these data altogether? They are totally uninterpretable, but disrupt the flow and distract. For figure 4, I suggest to delete the graph in panel A that shows filopodial lengths and present instead quantification of the retrograde flow rates, which is the main point of this figure that validates the conclusion that formins are needed for actin attachment. Right now, flow rates are given in the text and less visible.

6. The speculation that forces accelerate the filopodial growth is too far stretched and poorly, if at all, supported by the data. If it was the case, the trace in Figure 1C (middle) would show peaks of filopodial growth after the peak force is over. Instead, the trace shows dips during a force peak indicating that filopodial growth is slowed down when the force is high, but the growth rate returns to approximately the same (not higher) value, as soon as the peak force is over.

7. Mechanosensitivity of other adhesion types (focal adhesions, cell-cell adhesions, and lamellipodial adhesions) is well recognized, although they rely on different molecular mechanisms. Therefore, regardless of whether or not the authors decide to define their results as “striking”, they need to put them on the background of this prior knowledge and compare and contrast various mechanosensitive (or not mechanosensitive, if they know such) adhesions.

Minor points:

1. Legend for Movie 4: The bead does not really return to the filopodial tip, as described in the legend, although it does stay in the field. Please, find more accurate wording to describe what happens in the movie and indicate the time of trap release, ideally in the movie itself.

2. The explanations given by authors about differences between the peak forces and the maximal forces, as well as about the apparent discrepancy between figures 3E and S4, did not appear in the revised manuscript and were only shown in the response letter.

3. p. 4 (bottom). “In addition to myosin X, the filopodia tips were also enriched in several other proteins such as mDia2, VASP and talin (fig. S1B and D).” This statement could be understood as if endogenous proteins were found at the tips in myosin X-induced filopodia. To avoid this false impression, indicate right here that these are ectopically expressed proteins. The references about similar localization of endogenous proteins would be more appropriate here than elsewhere, as in the current version of the manuscript.

4. p. 6 (about retrograde flow of myosin X and VASP): As before, a still image is cited to illustrate the point of co-migration (Figure S1B). In response to my previous comment, authors provided a reference to movie 6 in the letter (not in the manuscript), which actually shows dynamics of these proteins after SMIFH2 treatment, rather than in control conditions, and thus does not illustrate the point given on p. 6. I suggest authors to skip the point of myosin X/VASP co-migration here, because

it is not new anyway, and present the results about the retrograde flow only in the section about formin inhibition, where it is important to support the idea of actin detachment after inhibition of formins.

5. Figures 1 and S4: Express time in the same units (either min:sec or sec) for images (now, min;sec) and graphs (now, sec) to allow for easier comparison of the timing of detachment events and trap releases between both sets of data. These events should be also indicated on images, not only on graphs and in legends.

6. Figure S7 legend: "(B) SMIFH2 enhanced the filament detachment rate by an order of magnitude without affecting the formin mediated elongation rates of the filaments (35 ± 6 vs. 32 ± 8 subunits/s, $n = 40$ filaments without and with SMIFH2, respectively)." An order of magnitude difference in the detachment rates is not obvious from the graph, which shows survival fractions. I recommend to show data for the detachment rates with appropriate statistics in addition to the existing graph. On the other hand, the point of precise quantification of filament elongation rates is obscure, especially because the lack of a difference is counterintuitive considering a known role of formins in filament elongation. This quantification seems irrelevant in any case.

For future, I recommend authors to pay more attention to details during preparation of their manuscript, in terms of accuracy of statements, clarity of illustrations and their annotation, and completeness of technical information. It would not only make the reviewers' job easier, but would also allow them to focus on more conceptual issues and provide comments more to the point already in the first round of reviews.

Reviewer #2 (Remarks to the Author):

In the revised version of the manuscript "Force dependence of filopodia adhesion: involvement of myosin II and formins" the authors addressed most of the concerns that have been raised by both reviewers. The manuscript now visualizes data more clearly and contains additional experiments. These additional experiments lead to new insights into molecular regulation, mechanosensing and mechanoresponse of filopodia.

Ironically, the additional data even more clearly demonstrate the limitations of the experimental setting. Solving all issues, however, would be beyond the scope of this paper. Nevertheless, before acceptance of the final manuscript the limitations of their experimental settings, the relevance of the findings in a physiological context, and alternative mechanisms explaining the observed effects should be addressed more carefully and more precisely by the authors.

1.) This reviewer agrees with the authors that the experimental findings support the importance of myosin IIA (specifically the motor function) for the mechanoresponsive behavior of filopodia. But at the same time, the reviewer is still not convinced that the mechanism is as direct as the authors imply. The estimation of forces based on myosin numbers and duty ratio (second subchapter of discussion) implies a direct force transmission from actin + MyoIIA minifilament to formin at the filopodia tip without any friction on its way. Friction might be low or forces within the surrounding actomyosin network might add up to compensate for friction. That might be unlikely but yet conceivable. However, the localizations of MyoIIA minifilaments in Fig. 2D and movie S8C are still too far away from the filopodia base to support a direct and friction-less force transmission. (I admit that the MyoIIA localization in Fig. 2B supports a model of direct force transmission. But again, Fig. 2D and movie S8C do not.)

Even if the authors were able to correlate MyoIIA localization at the base of a filopodium with a 3 pN peak as seen in Fig 1C it would be surprising for a completely direct model that the growth rate of a

filopodium is not affected by a peak in the applied force.

Therefore, I would appreciate if the authors made readers more aware of the limitations of a direct force transmission model.

2.) Based on their data obtained with MyoX induced filopodia, the authors propose a hypothesis of formin-mediated filopodia growth in response to a pulling force. However, other publications document different filopodial behavior (not only Guillou et al, 2008, but also, for example, Johnson et al, 2015, JCB, or Congying Wu et al, 2012, Cell, as well as unpublished results of filopodia dynamics of this reviewer).

Most importantly, additional data now presented by the authors (Cdc42 Q61L induced filopodia) confirm that filopodia without MyoX overexpression have similar phenotypes as described in the publications above. These filopodia do not show growth after pulling and therefore contradict the hypothesis of this manuscript. Accordingly, it is of high relevance to present these differences and possible explanations in the discussion in a more detailed way as it is currently the case.

At the moment, the authors justify the overexpression of MyoX and the relevance of the observed effects with an overexpression of MyoX in certain cancer cells. This is an important point and does not diminish the relevance of these findings. In fact, a careful comparison of MyoX induced “cancer-like” filopodia and “normal/physiological” filopodia might even increase the interest of a broader readership in this manuscript.

3.) The authors agree that filopodia length is not an ideal measure for their claims. Based on their hypothesis that pulling forces on formins lead to filopodia growth a plot as shown in Fig1C middle panel appears best suited whenever possible. Accordingly, a similar analysis would be important whenever possible which should be the case for Fig3 and SupFig4. The perturbation of MyoIIA or formins in these experiments would make such an analysis very important to test the author’s hypothesis. Additionally, the correlation of force versus time and growth versus time could be discussed for different settings and would also address the concerns raised above (point 1).

Minor points:

- The introduction considers filopodia “as primary minimal cell-matrix adhesion structures”. But the authors claim in response to reviewer 1 that filopodia are not identical to either nascent or mature focal adhesions. They emphasize the lack of vinculin as a difference to classical adhesion structures. Nascent adhesions are regarded as primary minimal cell-matrix adhesion structures. The authors should clarify if they have a different opinion. Their reply to reviewer 1 on this subject could also be helpful for readers to understand the relevance of this paper and could, therefore, be used in the discussion.

- Fig 1C: The labeling of the y-axis is wrong. The figure shows length over time, i.e. velocity. A plot of length change (= velocity), dx as currently indicated, over time would indicate acceleration.

- Units should be consistently separated from numbers by a space character (this is at the moment only the case for units of time).

Reviewers' comments:

Reviewer #1 (Remarks to the Author):

In this study, authors show that under external force, myosin X-induced filopodia maintain adhesion to fibronectin-coated beads while their actin core keeps growing. These data indicate that the integrin-mediated adhesion of filopodia is mechanosensitive. Furthermore, authors show that these effects require myosin IIA and formin activity, suggesting that myosin IIA contributes endogenous force to strengthen adhesions, whereas formins function by attaching filopodial actin filaments to the adhesion site at the filopodial tip. These findings present an important contribution to our understanding of cell motility and adhesion. They complement and extend the existing knowledge that other cell adhesion types (focal adhesions, cell-cell adhesions, and lamellipodial adhesions) are also mechanosensitive.

We are grateful to this reviewer for qualifying our studies as an “important contribution to our understanding of cell motility and adhesion”.

The previous version of the manuscript suffered from multiple weaknesses in terms of data presentation, both technical and conceptual. In the revised manuscript, authors clarified some, but not all of these confusing aspects of their study.

We appreciate this reviewer's help in finding the weaknesses in our manuscript and made every effort to address her/his comments.

Major points:

1. In the response letter, authors limit the scope of their study specifically to myosin X-induced filopodia, because Cdc42- and mDia-induced filopodia in their hands behaved differently. This decision is acceptable. However, because of this observation, they need to discuss (i) how general their findings are considering that other types of filopodia are expected to have endogenous myosin X at the tips, and (ii) what could be special about overexpressed myosin X that enables different behavior of filopodia.

During preparation of the revised version of the manuscript we conducted several new experiments to elucidate the differences between force- and myosin IIA-dependence of different types of filopodia. In particular, we determined the lifetimes of filopodia induced in Cos-7 cells by myosin X, constitutively active mutant of mDia2 (DIAPH3) formin (mDia2 Δ DAD), and constitutively active mutant of Cdc42 (Cdc42 Q61L), and checked whether these lifetimes are affected by the expression of myosin IIA. In addition, we included experiments in which we compared the effects of pulling forces on the growth of filopodia induced by expression of these proteins. We have now presented and discussed

these data in the corresponding sections of the manuscript. In addition we extended the discussion of the mechanisms of myosin X function.

2. One of the main conclusions of the study is that myosin IIA functions locally from the base of filopodia. Although this mechanism seems most natural, I do not find that it is sufficiently supported by the presented data. I am not satisfied with how quantification of myosin IIA localization to filopodia bases is done. First, only one movie was quantified. Second, authors quantified a fraction of filopodia that was ever visited by myosin IIA, which could give an impression of a greater presence of myosin II at filopodial bases than it really was. However, even this quantification gave only 70% of myosin IIA-positive filopodia. The authors need to (i) determine a fraction of time during which each filopodium had myosin II at the base; (ii) describe how the base was defined; (iii) use non-filopodial cell edges as controls; and (iv) provide statistics for these data. Similar quantification should be done for myosin IIB. Even in the absence of proper quantification, the provided data suggest that myosin IIA is only transiently present, not abundant and not ubiquitous at the filopodial bases. Therefore, the outcomes of myosin II inhibition could reflect global effects, such as changes in overall cell contractility, membrane tension, etc., rather than local force generated at the filopodial bases. This possibilities could be discriminated by correlating the force peaks with the presence of myosin II at the filopodial base, but authors declined doing this experiment, in which case they need to significantly tone down their claims about specific mechanism by which myosin II functions here.

In the revised version of our paper we significantly strengthen the evidence in favor of local function of the myosin IIA filaments at the base of filopodia. Briefly, we formalized the assessment of myosin II-associated filopodia by defining the base of filopodium as the space occupied by a circle with a diameter of 1.5 μm (slightly larger than diameter of filopodia), which was internally tangent to the plasma membrane at the area of negative curvature manifesting the filopodia orifice. We also taken into account the myosin IIA filaments located above such a circle inside the filopodia. We classified all the filopodia into three groups: filopodia not associated with myosin IIA filaments; filopodia associated with myosin IIA filaments during the entire period of observation (240 s); and filopodia associated with myosin IIA during a part of the observation period. These experiments demonstrated the presence of myosin IIA filaments at the bases of 53% of filopodia during entire period of observation, while myosin IIB filaments were detected in association with only 21% of filopodia. We have than measured the duration of lifetime in 102 filopodia of three independent cells and found that association with myosin IIA filaments makes the filopodia lifetime significantly longer according to a nonparametric Wilcoxon rank-sum test. These data strengthen our hypothesis on the local function of myosin IIA filaments in filopodia adhesion and growth.

3. I am still confused by the description of how forces and bead detachments were determined. In response to my previous comment, the authors provided the

following description in Methods: “The moment of detachment of filopodia from the bead was detected by return of the bead to the center of optical trap”. In Results, they say: “in the cells with impaired myosin II activity the filopodia detached earlier” (p. 9), suggesting that the bead returned to the trap sooner than in control. However, the figure 3B-D legend says: “Filopodia in cells with suppressed myosin II or formin activity ... do not produce forces sufficient for noticeable bead deflection during the stage movement”, which is indeed obvious from the figure. How did they detect detachment in myosin-impaired cells if the bead did not move out of the trap? How did they determine forces presented in Figure 3E without bead movement? In control cells, the bead periodically returns to the trap after high force episodes (Figure 1). Apparently, such cases were not considered to be detachments. Then, how were actual detachments discriminated objectively from the bead returns after high force episodes?

We are grateful to this reviewer for attracting attention to the necessity of rigorous definition of the bead detachment. We agree that this issue should be better clarified. In fact, there is no contradiction between our definition of detachment as returning of the bead to the center of optical trap and our statement that filopodia in cells with impaired function of myosin II or formin have decreased adhesion to fibronectin-coated beads. Filopodia from such cells were able to attach to the bead, which was proven in experiments where the laser trap was switched off, upon which beads still remained attached to filopodia. In the course of pulling, the position of the beads attached to the filopodia in cells with deficient myosin II/formin function relative to the center of the trap still underwent some slight changes, which reflected the fact that the forces applied by filopodia to such beads were still developed even though were significantly weaker than in control cells. As indicated in Methods, the optical trap technique permits detection of the slightest deflection of the bead from the center of the trap (with a resolution of ~ 5 nm, corresponding to the smallest force measured of ~ 0.04 pN), which are not obvious in the images shown in Fig. 3B-D (now 4B-D) mentioned by the reviewer. After some period of time, however, the bead returned to the center of the trap and did not demonstrate any detectable deflections for 1-2 minutes. Switching off the trap in such cases resulted in physical release of the bead and its disappearance from the field of view. Therefore, we qualified the last detectable drop of the force as “detachment” of bead from filopodium.

Such detachment events were sometimes observed in control cells but they usually occurred after longer period of pulling. However, the typical response of control cells was not detachment of the bead but its withdrawal from the trap demonstrating the “reinforcement” of the adhesion between the bead and filopodium. The bead withdrawn by filopodia from the trap did not detach from filopodia upon the switching off the trap, which permits us to easily detect this type of outcome. Withdrawal from the trap was rarely observed in the cells with impaired myosin II or formin function. The third type of outcome of pulling experiments was the formation of membrane tether, which did not contain actin

core. The tether formation was never observed in control filopodia attached to fibronectin-coated beads, but were evident in filopodia of cells with impaired myosin IIA or formin function and especially in filopodia attached to concanavalin A-coated beads. In the revised version of the paper we added a new figure (Fig. 4E), in which both fractions and numbers of cases for each type of outcome of pulling experiments are indicated for each type of cell treatment. We believe that this new information will be sufficient to address the reviewer's concerns. We have also corrected some statements in the description of the results, which could create confusion.

4. When discussing myosin II and formin inhibition, authors state that the duration of the contacts was "shorter". However, the contact duration remains unquantified despite my request in the previous review. Instead, the authors responded that they showed specific detachment times for single examples in a couple of figures, which cannot be considered as quantification. They need a sufficient sample size and statistics to show the stated differences.

As we mentioned in the answer to the previous comment, we have now presented complete data summarizing in full details the results of all our pulling experiments (new Fig. 4E). For each such experiment we have measured the duration of the time interval between the start of pulling and one of the three possible final outcomes: (i) withdrawal of the bead from trap by filopodium, (ii) detachment of filopodium from the bead detected by returning of the bead to the center of the trap, and (iii) formation of membrane tether between the bead and filopodium tip. For each type of treatment, average duration of the periods between the start of pulling and the respective outcome was indicated (mean \pm SEM). This information, together with maximal pulling force measurements shown in Fig. 4F provides complete quantitative description of our results.

5. If authors agree that the filopodial length is not a useful parameter, why not to delete these data altogether? They are totally uninterpretable, but disrupt the flow and distract. For figure 4, I suggest to delete the graph in panel A that shows filopodial lengths and present instead quantification of the retrograde flow rates, which is the main point of this figure that validates the conclusion that formins are needed for actin attachment. Right now, flow rates are given in the text and less visible.

We agree with this suggestion and modified the manuscript accordingly. Following the recommendations of the reviewer, in the new Fig. 5 (corresponding to the previous Fig. 4) we removed the data on filopodia length and introduced the complete set of measurement of myosin X retrograde flow (Fig. 5B).

6. The speculation that forces accelerate the filopodial growth is too far stretched and poorly, if at all, supported by the data. If it was the case, the trace in Figure 1C (middle) would show peaks of filopodial growth after the peak force is over. Instead, the trace shows dips during a force peak indicating that filopodial growth

is slowed down when the force is high, but the growth rate returns to approximately the same (not higher) value, as soon as the peak force is over.

In fact, we never claimed that the data presented in this manuscript revealed a correlation between filopodia growth velocities and the values of applied forces. In the revised version, we tried to avoid any statement that could be understood as a conjecture assuming such correlation. We have, however, referred to our previous publication, in which we demonstrated that the rate of formin-driven actin polymerization *in vitro* depends on the pulling force (Yu et al. 2017). This could create a wrong impression that we think that such dependence is valid for the *in vivo* growth of filopodia. In the revised version of the paper we introduced several editorial changes that made our statements clearer and reduced the probability of misunderstanding on this issue.

7. Mechanosensitivity of other adhesion types (focal adhesions, cell-cell adhesions, and lamellipodial adhesions) is well recognized, although they rely on different molecular mechanisms. Therefore, regardless of whether or not the authors decide to define their results as “striking”, they need to put them on the background of this prior knowledge and compare and contrast various mechanosensitive (or not mechanosensitive, if they know such) adhesions.

We agree that all well studied cell-matrix and cell-cell adhesions demonstrated one or another type of mechanosensitivity. The concept of adhesion-mediated mechanosensing appeared to be broader than was originally thought. At the same time, our knowledge of specific mechanisms of adhesion-dependent mechanosensitivity is very limited and extensive further studies are needed to understand these mechanisms and to reveal the differences between different types of adhesion in this context. We believe that our discovery of mechanosensitivity of filopodia adhesions is important, first of all, because filopodia in general and filopodia induced by myosin X in particular are in fact ubiquitous cellular features formed both under 2-D and 3-D conditions and responsible for basic cellular reactions such as, for example, durotaxis. More specifically, filopodia are interesting for the studies of force dependence of adhesion since they may represent the most elementary adhesion structure. Indeed, the filopodium is a relatively simple device in which small cluster of adhesion receptors at the tip is connected with force generating individual myosin filaments at the base by the actin bundle consisting of only about 20 individual actin filaments. The organization of focal adhesions and cadherin-mediated cell-cell adhesions is by far more complex. We believe that our finding of mechanosensitivity of filopodia adhesions will initiate a series of future studies that will significantly improve our understanding of the basic mechanisms of adhesion-dependent mechanosensitivity. We have now included some of the considerations outlined above into the new version of Discussion. The adjective “striking” is removed from the text of the paper.

Minor points:

1. Legend for Movie 4: The bead does not really return to the filopodial tip, as described in the legend, although it does stay in the field. Please, find more accurate wording to describe what happens in the movie and indicate the time of trap release, ideally in the movie itself.

After switching off the trap at the time point 3:30 the bead immediately returned to filopodium tip as seen in the next frame (3:32) and then randomly moved in the proximity of the filopodium tip (3:32-3:54). Thus, the bead remained associated with filopodium even though the link connecting the bead with the filopodium tip became hardly visible because of disappearance of actin. This indicates that the bead and filopodium tip remain connected via the membrane tether. All this information is now included into the legend to the movie (new Supplementary movie 12) and the moment of switching off the trap is indicated in the movie. Further observations showed that at 3:56 the bead moved away from the filopodium for about 4 μm while remaining apparently connected with the tip. Since this motion is not relevant to the main message illustrated by this movie, we decided to shorten the movie attached to the revised version of the paper by stopping it at 3:54. We are grateful to this reviewer for her/his comment that helped us to clarify the message presented in this movie.

2. The explanations given by authors about differences between the peak forces and the maximal forces, as well as about the apparent discrepancy between figures 3E and S4, did not appear in the revised manuscript and were only shown in the response letter.

We have now added the explanation of the differences between data presented in Fig. 3E and S4 (new Figs. 4F and Supplementary Fig. 2B) to the legend to Supplementary Fig. 2B.

3. p. 4 (bottom). "In addition to myosin X, the filopodia tips were also enriched in several other proteins such as mDia2, VASP and talin (fig. S1B and D)." This statement could be understood as if endogenous proteins were found at the tips in myosin X-induced filopodia. To avoid this false impression, indicate right here that these are ectopically expressed proteins. The references about similar localization of endogenous proteins would be more appropriate here than elsewhere, as in the current version of the manuscript.

We have now changed the text according to the reviewer's suggestion. In description of our results concerning localization of mDia2, VASP and talin to the filopodia tips, the usage of exogenous fusion constructs of these proteins with mCherry is clearly indicated in the text and in legend to new Fig. 1A. The references to the papers demonstrating localization of corresponding endogenous proteins by immunofluorescence staining are given in the same paragraph of the Results.

4. p. 6 (about retrograde flow of myosin X and VASP): As before, a still image is cited to illustrate the point of co-migration (Figure S1B). In response to my previous comment, authors provided a reference to movie 6 in the letter (not in the manuscript), which actually shows dynamics of these proteins after SMIFH2 treatment, rather than in control conditions, and thus does not illustrate the point given on p. 6. I suggest authors to skip the point of myosin X/VASP co-migration here, because it is not new anyway, and present the results about the retrograde flow only in the section about formin inhibition, where it is important to support the idea of actin detachment after inhibition of formins.

We are grateful to the reviewer for this suggestion. We have now moved the data on co-migration of myosin X and VASP to the section devoted to the dynamics of filopodia in SMIFH2-treated cells.

5. Figures 1 and S4: Express time in the same units (either min:sec or sec) for images (now, min;sec) and graphs (now, sec) to allow for easier comparison of the timing of detachment events and trap releases between both sets of data. These events should be also indicated on images, not only on graphs and in legends.

Thanks for these suggestions. In the revised version of the manuscript we unified the time scales in images and graphs indicating everywhere the time in seconds. The moments of particular events (detachment filopodium from the bead, withdrawal the bead from the trap, switching off etc.) are now indicated not only in the legend and but also on images as requested by the reviewer. At the same time for technical reasons we prefer to indicate time in the movies and legends to the movies in min:sec format. We believe it will not be difficult for the readers to convert one form of time presentation to another.

6. Figure S7 legend: "(B) SMIFH2 enhanced the filament detachment rate by an order of magnitude without affecting the formin mediated elongation rates of the filaments (35 ± 6 vs. 32 ± 8 subunits/s, $n = 40$ filaments without and with SMIFH2, respectively)." An order of magnitude difference in the detachment rates is not obvious from the graph, which shows survival fractions. I recommend to show data for the detachment rates with appropriate statistics in addition to the existing graph. On the other hand, the point of precise quantification of filament elongation rates is obscure, especially because the lack of a difference is counterintuitive considering a known role of formins in filament elongation. This quantification seems irrelevant in any case.

We have now included in the text of the results the values of k_{off} calculated by exponential decay fitting. In addition we used nonparametric logrank test to estimate the significance of the difference between the survival fraction curves as recommended in contemporary statistical instructions. This test revealed high significance of the difference between control and SMIFH2 treatment ($p = 0.0009$). Concerning the measurements of the rate of actin filament elongation,

our data are in complete agreement with the original data by Rizvi et al (2009), who demonstrated that SMIFH2 inhibits formin-dependent filament nucleation and decreases formin's affinity for the barbed end, but not the rate of elongation of actin filaments growing from formin-cated beads (see Fig. 3 in Rizvi et al, (2009)). The data on the rate of formin-induced actin elongation are relevant in the context of our study, since it was recently shown that elongation rate might also have an impact on the formin detachment rate (Cao et al, eLife, 2018, doi: 10.7554/eLife.34176), and we prefer to preserve them.

For future, I recommend authors to pay more attention to details during preparation of their manuscript, in terms of accuracy of statements, clarity of illustrations and their annotation, and completeness of technical information. It would not only make the reviewers' job easier, but would also allow them to focus on more conceptual issues and provide comments more to the point already in the first round of reviews.

We apologize for the defects in technical details of the manuscript and express our gratitude to this reviewer for her/his thorough examination of our manuscript and constructive criticism.

Reviewer #2 (Remarks to the Author):

In the revised version of the manuscript "Force dependence of filopodia adhesion: involvement of myosin II and formins" the authors addressed most of the concerns that have been raised by both reviewers. The manuscript now visualizes data more clearly and contains additional experiments. These additional experiments lead to new insights into molecular regulation, mechanosensing and mechanoreponse of filopodia.

We are grateful for this reviewer for appreciation of our efforts.

Ironically, the additional data even more clearly demonstrate the limitations of the experimental setting. Solving all issues, however, would be beyond the scope of this paper. Nevertheless, before acceptance of the final manuscript the limitations of their experimental settings, the relevance of the findings in a physiological context, and alternative mechanisms explaining the observed effects should be addressed more carefully and more precisely by the authors.

1.) This reviewer agrees with the authors that the experimental findings support the importance of myosin IIA (specifically the motor function) for the mechanoreponsive behavior of filopodia. But at the same time, the reviewer is still not convinced that the mechanism is as direct as the authors imply. The estimation of forces based on myosin numbers and duty ratio (second subchapter of discussion) implies a direct force transmission from actin + MyoIIA minifilament to formin at the filopodia tip without any friction on its way. Friction might be low or forces within the surrounding actomyosin network might add up to compensate for friction. That might be unlikely but yet conceivable. However,

the localizations of MyoIIA minifilaments in Fig. 2D and movie S8C are still too far away from the filopodia base to support a direct and friction-less force transmission. (I admit that the MyoIIA localization in Fig. 2B supports a model of direct force transmission. But again, Fig. 2D and movie S8C do not.)

Even if the authors were able to correlate MyoIIA localization at the base of a filopodium with a 3 pN peak as seen in Fig 1C it would be surprising for a completely direct model that the growth rate of a filopodium is not affected by a peak in the applied force.

Therefore, I would appreciate if the authors made readers more aware of the limitations of a direct force transmission model.

We understand the concerns of this reviewer and performed additional experiments to address the question about the effects of local forces generated by myosin filaments at the bases of filopodia on filopodia behavior. In these experiments performed on Cos-7 cells expressing myosin IIA-GFP, we compared the lifetimes of myosin X-induced filopodia associated and non-associated with the myosin IIA filaments in the same cell. This analysis showed that the presence of myosin IIA filaments at the filopodia bases positively correlates with lifetimes of the filopodia. The results of these experiments strengthen our idea that the function of myosin IIA in filopodia dynamics is local.

2.) Based on their data obtained with MyoX induced filopodia, the authors propose a hypothesis of formin-mediated filopodia growth in response to a pulling force. However, other publications document different filopodial behavior (not only Guillou et al, 2008, but also, for example, Johnson et al, 2015, JCB, or Congying Wu et al, 2012, Cell, as well as unpublished results of filopodia dynamics of this reviewer). Most importantly, additional data now presented by the authors (Cdc42 Q61L induced filopodia) confirm that filopodia without MyoX overexpression have similar phenotypes as described in the publications above. These filopodia do not show growth after pulling and therefore contradict the hypothesis of this manuscript. Accordingly, it is of high relevance to present these differences and possible explanations in the discussion in a more detailed way as it is currently the case. At the moment, the authors justify the overexpression of MyoX and the relevance of the observed effects with an overexpression of MyoX in certain cancer cells. This is an important point and does not diminish the relevance of these findings. In fact, a careful comparison of MyoX induced “cancer-like” filopodia and “normal/physiological” filopodia might even increase the interest of a broader readership in this manuscript.

We are grateful to this reviewer for these comments, which helped us to clarify our statements and make our Abstract and Discussion more focused. Briefly, we have introduced the following changes into the manuscript. First, we have extended experimental data related to induction of filopodia by the treatments other than myosin X overexpression (see new section “Note on other types of filopodia”). We presented preliminary data on filopodia induced by constitutively

active mDia2 formin (mDia2 Δ DAD) or Cdc42 (Cdc42 Q61L). In Cos-7 cells, lifetime of filopodia induced by mDia2 Δ DAD is increased in the presence on myosin IIA similarly to the situation with myosin X-induced filopodia. However, the lifetime of filopodia induced by Cdc42 Q61L did not respond on myosin IIA. Moreover, as we already mentioned in the previous version of the manuscript, filopodia induced by active mDia2 or Cdc42 in HeLa-JW cells did not grow upon application of pulling force through fibronectin-coated beads. Thus, force and myosin IIA dependence of filopodia induced by other treatments are not necessary the same as in filopodia induced by myosin IIA. We have now emphasized this statement in Discussion and, as this reviewer suggested, discussed possible mechanisms of myosin X function in filopodia more extensively.

Second, we clarified our statement concerning possible role of formins in mechanosensory response of filopodia. Our data indeed demonstrated that formin inhibitor SMIFH2 abolished the mechanosensory response of myosin X induced filopodia in HeLa-JW cells. We did not prove, however, that this effect is directly related to phenomenon of force-dependent augmentation of formin-driven actin polymerization, which we described in our other studies (Yu et al, 2017, 2018). Thus, we have now emphasized in the Discussion that explanation of mechanosensitivity of filopodia growth based on force dependence of actin polymerization remains still hypothetical. We have, however, strong evidence that SMIFH2 treatment leads to detachment of actin filaments from formins, which suggest that formins at the filopodia tips can participate in transmission of mechanical force generated by myosin IIA filaments at the filopodia bases.

Finally, we are grateful to this reviewer for attracting our attention to the excellent papers by Guillou et al (2008) and Johnson et al (2015), which we have now cited in the Discussion of the revised manuscript. The data presented in these papers do not contradict to our results and hypotheses we proposed. Indeed, the phenomenon of filopodia dependent stimulation of lamellipodia formation described by Guillou et al (2008) and Johnson et al (2015) can be explained by a signal generated at a site of filopodia adhesion to the substrate, which triggers formation of the lamellipodia in a Rac-dependent manner. We hypothesized that such signal can be force dependent. In our experimental system, filopodia-induced formation of lamellipodia was not very pronounced, but we agree that in other cellular systems this phenomenon could be important.

3.) The authors agree that filopodia length is not an ideal measure for their claims. Based on their hypothesis that pulling forces on formins lead to filopodia growth a plot as shown in Fig1C middle panel appears best suited whenever possible. Accordingly, a similar analysis would be important whenever possible which should be the case for Fig3 and SupFig4. The perturbation of MyoIIA or formins in these experiments would make such an analysis very important to test the author's hypothesis. Additionally, the correlation of force versus time and growth versus time could be discussed for different settings and would also

address the concerns raised above (point 1).

In the new version of the paper we performed more precise assessment of the adhesion and growth of filopodium attached to bead trapped by optical tweezers under condition of microscope stage movement. In addition to the measurement of the maximal force that filopodia exert on the beads during period of observation (shown in Fig. 3E), we have now presented several parameters characterizing the break of pulling-induced growth of filopodia at the last stage of each pulling experiment. Three possible outcomes of the pulling experiments are: (i) detachment of filopodia from bead that can be detected by returning of the bead to the center of the trap; (ii) withdrawal of the bead from the trap by filopodium; (iii) formation of membrane tether attached to bead at the tip of filopodium. Analysis of these data confirmed our conclusion that myosin IIA function is required for maintenance of filopodia-matrix adhesion. First, we have found that control HeLa-JW cells containing endogenous myosin IIA in the majority of the experiments withdrew the fibronectin coated bead from optical trap demonstrating “reinforcement” of filopodia adhesion to the bead. The cells lacking myosin IIA or treated with its inhibitors, as well as cells treated with formin inhibitor, demonstrate another type of outcome, namely, detachment of filopodia tips from the beads. Second, duration of the interval between start of stage movement and the filopodia detachment from the bead was in these cells shorter than in control cells. The forces developed at the interface of bead and filopodium were lower in these cells than in control ones as we described already in the previous version. Formation of membrane tethers at filopodia tips was never observed in control cells under the condition of stage movement, but sometimes occurred in cells with impaired function of myosin IIA or formin and especially in filopodia attached to concanavalin A-coated beads. Altogether, these data showed that without myosin IIA or formin cells couldn't reinforce or even maintain the adhesion of filopodia to the extracellular matrix. We have also preserved the data obtained earlier on cessation of filopodia growth under condition of suppression of myosin IIA or formin function. The data on filopodia length measurements did not add significant additional information and were therefore omitted from the revised version.

Minor points:

- The introduction considers filopodia “as primary minimal cell-matrix adhesion structures”. But the authors claim in response to reviewer 1 that filopodia are not identical to either nascent or mature focal adhesions. They emphasize the lack of vinculin as a difference to classical adhesion structures. Nascent adhesions are regarded as primary minimal cell-matrix adhesion structures. The authors should clarify if they have a different opinion. Their reply to reviewer 1 on this subject could also be helpful for readers to understand the relevance of this paper and could, therefore, be used in the discussion.

In the present version we omitted the phrase on filopodia “as primary minimal

cell-matrix adhesion structures”. Recent data from J. Ivaska laboratory (Jacquemet et al, 2018) demonstrated numerous differences between filopodia and focal adhesions, so this question requires more in depth discussion, which is beyond the scope of our paper.

- Fig 1C: The labeling of the y-axis is wrong. The figure shows length over time, i.e. velocity. A plot of length change (= velocity), dx as currently indicated, over time would indicate acceleration.

We apologize for not sufficiently clear labeling of the y-axis in the middle panel of old Fig. 1C. In the new version of the paper we changed this labeling (new Fig. 3C) from “Filopodia Dx , μm ” into “Filopodium length (Δx), μm ” and provided more detailed definition of the parameter Δx as well as method of its measurements. In fact, in this graph we plotted the length of new, pulling-induced, segment of filopodia versus time. This length was measured as coordinate of the bead in the system of coordinate associated with the moving microscope stage. The origin of the coordinate system corresponds to the bead position in the center of the laser trap at the initial time point. The net filopodium length (Δx) is a sum of the microscope piezo stage displacement from its initial position and deviation of the bead from the center of the optical trap.

- Units should be consistently separated from numbers by a space character (this is at the moment only the case for units of time).

We corrected the text accordingly.

REVIEWERS' COMMENTS:

Reviewer #1 (Remarks to the Author):

The manuscript is significantly improved. In general, I am satisfied with this revision except for a few minor comments:

1. p. 5, line 104: "One or few myosin II filaments were 104 usually located at the filopodium base (Fig. 1c and d)." Should it be Fig. 1d and e?
2. p. 8, line 189: definition of Δx here ("the bead displacement from the center of the trap") is different from that in the figure 3c legend ("The length of new pulling-induced segment of filopodia")
3. p. 9, line 194: Should this statement refer to Movie 10?
4. p. 26, lines 633-640: For quantification of myosin IIa at filopodial bases, there is no indication as for whether all filopodia were quantified or only those at the stable cell edges, as mentioned in Results. Since authors do show filopodia at dynamic edges in Movie 8, they should quantify all filopodia, especially because a motivation to consider only filopodia at the stable edges is unclear and may bias the quantification toward a greater presence of myosin IIa at filopodial bases.
5. Fig. 4d legend: Did the bead really disappear after SMIFH2 treatment? A part of it is visible. As I understand, it should not disappear, as it is supposed to be on a membrane tether.
6. Fig. 5a: In the control kymograph, myosin X either undergoes anterograde, not retrograde flow, or the image is incorrectly shown. If it is shown correctly, please, explain.

Reviewer #2 (Remarks to the Author):

The second revised version of this manuscript is now largely improved compared to the initial version submitted by the authors.

Overall, I still not completely convinced by the model of force-dependent, formin-mediated filopodia growth which is favored by the authors. The differences between MyoX induced filopodia and filopodia induced by constitutive active Cdc42 are substantial. As the authors correctly point out, this means that their findings are limited to MyoX induced filopodia. In addition, the function of MyoX in filopodia is still unclear (for example as a result of MyoX (Δ)FERM domain experiments). Another concern is that the effects of MyoIIA inhibition within the same cell are limited to filopodia attached to the substrate (line 261-263). If force-mediated filopodia growth is only caused by MyoIIA and formins at the tip, the differences between substrate attached and un-attached filopodia are hardly to explain by the author's model.

Nevertheless, I acknowledge the data gathered in this work and that addressing every possible question is beyond the scope of this work. The open questions might motivate and direct future work that will allow a better understanding of the topic.

Some minor points remain:

- Line 74: "...depends on myosin II activity" The authors should be specific throughout the manuscript about the myosin they are talking about (here MyoIIA; MyoIIB has not such an effect as the authors showed).

- Explaining the color code of Fig. 2d and Fig. 4e in the main text (in brackets for example) would help the reader to understand the figures.
- I don't agree in general with 3D figures as Fig. 4g. In any case, it seems important to me to know the degree of growing filopodia and not only the number of pausing filopodia.
- Line 497 seq: there is increasing evidence that the importance of RIAM might be overrated. Specifically, Klapproth et al. (Blood, 2015) showed that RIAM is needed for activation of beta2 integrins but not for alpha4beta1 integrin. I think it's important to mention these findings and to reduce claims about RIAM involvement to avoid entrenching a wrong idea in the community.
- Line 550 seq: The authors argue that their findings may explain filopodia dependent steering of growth cones. Ironically, reference 84 cited by the authors mentions in the title (!) that this process depends rather on Myo11B and not Myo11A which is, to my knowledge, an accepted finding among neurobiologists.

REVIEWERS' COMMENTS:

Reviewer #1 (Remarks to the Author):

The manuscript is significantly improved. In general, I am satisfied with this revision except for a few minor comments:

We are grateful to this reviewer for the appreciation of our efforts.

1. p. 5, line 104: "One or few myosin II filaments were usually located at the filopodium base (Fig. 1c and d)." Should it be Fig. 1d and e?

The Fig. 1c and d show filopodia of HeLa-JW cells, while Fig. 1e shows filopodium of Cos-7 cell (described in the next paragraph). Since Fig. 1c does not show myosin II filaments at high magnification, in revised version we now refer to Fig. 1d only (line 97).

2. p. 8, line 189: definition of Δx here ("the bead displacement from the center of the trap") is different from that in the figure 3c legend ("The length of new pulling-induced segment of filopodia")

We are very grateful to this reviewer for attracting our attention to this discrepancy. In fact, we are defining two different parameters: the bead displacement from the center of the trap (ΔX) and the net filopodium length change that we also mistakenly denoted as ΔX . In the revised version the net filopodium length change is denoted as ΔL and definition of this variable is included into the legend of Fig. 3 (lines 799-801, revised version).

3. p. 9, line 194: Should this statement refer to Movie 10?

We are grateful for this correction. Should be, indeed, Movie 10 (line 187, revised version).

4. p. 26, lines 633-640: For quantification of myosin IIa at filopodial bases, there is no indication as for whether all filopodia were quantified or only those at the stable cell edges, as mentioned in Results. Since authors do show filopodia at dynamic edges in Movie 8, they should quantify all filopodia, especially because a motivation to consider only filopodia at the stable edges is unclear and may bias the quantification toward a greater presence of myosin IIa at filopodial bases.

Quantification of myosin IIA at filopodia bases was performed in Cos-7 cells and based on assessment of all filopodia. This is now clearly indicated in Methods (line 529, revised version). The statement on restriction of our analysis to only filopodia growing from the stable edges (lines 167-169, revised version) is related to HeLa-JW cells where mainly filopodia of this type were suitable for pulling experiments. This is now indicated in the corresponding section of the Results.

5. Fig. 4d legend: Did the bead really disappear after SMIFH2 treatment? A

part of it is visible. As I understand, it should not disappear, as it is supposed to be on a membrane tether.

We agree. The sentence in the legend to Fig. 4d is now corrected (lines 818-820, revised version).

6. Fig. 5a: In the control kymograph, myosin X either undergoes anterograde, not retrograde flow, or the image is incorrectly shown. If it is shown correctly, please, explain.

Indeed, the myosin X patch in the untreated cell (Supplementary movie 17 and upper panel in new Fig. 6a) is slightly moving forward. The velocities of retrograde myosin X movements in non-treated cells shown in the new Fig. 6b were measured in filopodia pulled by the beads (as for example in Fig. 3c) as indicated in the text. In these cases, the anterograde movement of myosin X patches was not detected.

Reviewer #2 (Remarks to the Author):

The second revised version of this manuscript is now largely improved compared to the initial version submitted by the authors.

We are grateful to this reviewer for appreciation of our efforts.

Overall, I still not completely convinced by the model of force-dependent, formin-mediated filopodia growth which is favored by the authors. The differences between MyoX induced filopodia and filopodia induced by constitutive active Cdc42 are substantial. As the authors correctly point out, this means that their findings are limited to MyoX induced filopodia. In addition, the function of MyoX in filopodia is still unclear (for example as a result of MyoX (Δ)FERM domain experiments). Another concern is that the effects of MyoIIA inhibition within the same cell are limited to filopodia attached to the substrate (line 261-263). If force-mediated filopodia growth is only caused by MyoIIA and formins at the tip, the differences between substrate attached and un-attached filopodia are hardly to explain by the author's model.

According to our model, the force generated by myosin IIA is transmitted to the filopodia tips via actin core filaments attached to the formin molecules at the tips. This force can be transduced to the adhesive domain of filopodia (talin, integrin, etc.) only if filopodia is attached to the substrate. Therefore, we believe that only in attached filopodia myosin IIA and formin are involved in reinforcement of adhesion signaling, which is required for stabilization of filopodia adhesion. We have now highlighted this idea in the Discussion.

Nevertheless, I acknowledge the data gathered in this work and that addressing every possible question is beyond the scope of this work. The open questions might motivate and direct future work that will allow a better understanding of the topic.

We gratefully appreciate this statement of the reviewer.

Some minor points remain:

- Line 74: "...depends on myosin II activity" The authors should be specific throughout the manuscript about the myosin they are talking about (here MyoIIA; MyoIIB has not such an effect as the authors showed).

We agree, and corrected this (line 70, revised version).

- Explaining the color code of Fig. 2d and Fig. 4e in the main text (in brackets for example) would help the reader to understand the figures.

We have now explained the color codes on Fig. 2d and Fig. 4e in the corresponding sections of the results (lines 132-133 and 199-200, respectively, revised version).

- I don't agree in general with 3D figures as Fig. 4g. In any case, it seems important to me to know the degree of growing filopodia and not only the number of pausing filopodia.

We have substituted the 3D graph in former Fig. 4g by simple bar diagram in the new Fig. 5. This new diagram permits the reader easily estimate the fractions of pausing, protruding and retracting filopodia. Therefore, we have now withdrawn the data on the number of pausing filopodia from the text.

- Line 497 seq: there is increasing evidence that the importance of RIAM might be overrated. Specifically, Klapproth et al. (Blood, 2015) showed that RIAM is needed for activation of beta2 integrins but not for alpha4beta1 integrin. I think it's important to mention these findings and to reduce claims about RIAM involvement to avoid entrenching a wrong idea in the community.

We have now included the reference to Klapproth et al. (Blood, 2015) in our discussion (line 430, revised version) and shortened the section related to possible involvement of RIAM in mechanosensitivity of filopodia adhesion.

- Line 550 seq: The authors argue that their findings may explain filopodia dependent steering of growth cones. Ironically, reference 84 cited by the authors mentions in the title (!) that this process depends rather on MyoIIB and not MyoIIA which is, to my knowledge, an accepted finding among neurobiologists.

We believe that preferential use of one or another myosin II isoforms for filopodia mediated mechanosensing could be cell type specific. We have now indicated in the text that in growth cone myosin IIB can be responsible for the mechanosensitivity (lines 467-469, revised version).